# Age-related matrix stiffening epigenetically regulates α-Klotho expression and compromises chondrocyte integrity

Hirotaka Iijima [1,2,3] ✉, Gabrielle Gilmer [4,5,6,7,8], Kai Wang [1,7,8], Allison C. Bean[1,9], Yuchen He[10], Hang Lin [5,9,10], Wan-Yee Tang[11], Daniel Lamont [12], Chia Tai [13], Akira Ito[13], Jeffrey J. Jones [14], Christopher Evans [15] & Fabrisia Ambrosio [1,5,7,8,9,10,11] ✉

Extracellular matrix stiffening is a quintessential feature of cartilage aging, a leading cause of knee osteoarthritis. Yet, the downstream molecular and cellular consequences of age-related biophysical alterations are poorly understood. Here, we show that epigenetic regulation of α-Klotho represents a novel mechanosensitive mechanism by which the aged extracellular matrix influences chondrocyte physiology. Using mass spectrometry proteomics followed by a series of genetic and pharmacological manipulations, we discovered that increased matrix stiffness drove *Klotho* promoter methylation, downregulated *Klotho* gene expression, and accelerated chondrocyte senescence in vitro. In contrast, exposing aged chondrocytes to a soft matrix restored a more youthful phenotype in vitro and enhanced cartilage integrity in vivo. Our findings demonstrate that age-related alterations in extracellular matrix biophysical properties initiate pathogenic mechanotransductive signaling that promotes *Klotho* promoter methylation and compromises cellular health. These findings are likely to have broad implications even beyond cartilage for the field of aging research.

All cells in the human body are subject to mechanical influences. This is particularly true of articular cartilage, given that its primary role is to transmit forces to the underlying bone and decrease friction in the joint. However, these important functions are disrupted by the progressive cartilage deterioration that occurs with aging. In 1742, British anatomist William Hunter wrote that cartilage, "*when destroyed, it is never recovered.*"[1]. In most cases, cartilage damage eventually progresses to osteoarthritis (OA), which affects ~32.5 million Americans[2]. Although we have known of the poor healing capacity of cartilage for over 250 years, this limited capacity is both poorly understood and inadequately treated in the clinic.

The lack of successful non-surgical treatments for OA is attributed, in part, to: (1) gaps in our current understanding of whether pre-clinical OA models recapitulate human disease and (2) an incomplete knowledge

[1]Department of Physical Medicine and Rehabilitation, University of Pittsburgh, Pittsburgh, PA, USA. [2]Japan Society for the Promotion of Science, Tokyo, Japan. [3]Institute for Advanced Research, Nagoya University, Nagoya, Japan. [4]Medical Scientist Training Program, School of Medicine, University of Pittsburgh, Pittsburgh, PA, USA. [5]Department of Bioengineering, University of Pittsburgh, Pittsburgh, PA, USA. [6]Cellular and Molecular Pathology Graduate Program, University of Pittsburgh, Pittsburgh, PA, USA. [7]Discovery Center for Musculoskeletal Recovery, Schoen Adams Research Institute at Spaulding, Boston, MA, USA. [8]Department of Physical Medicine & Rehabilitation, Harvard Medical School, Boston, MA, USA. [9]McGowan Institute for Regenerative Medicine, University of Pittsburgh, Pittsburgh, PA, USA. [10]Department of Orthopaedic Surgery, University of Pittsburgh, Pittsburgh, PA, USA. [11]Department of Environmental and Occupational Health, University of Pittsburgh School of Public Health, Pittsburgh, PA, USA. [12]Petersen Institute of Nanoscience and Engineering, University of Pittsburgh, Pittsburgh, PA, USA. [13]Department of Motor Function Analysis, Human Health Sciences, Graduate School of Medicine, Kyoto University, Kyoto, Japan. [14]Proteome Exploration Laboratory, Beckman Institute, California Institute of Technology, Pasadena, CA, USA. [15]Department of Physical Medicine & Rehabilitation, Mayo Clinic, Rochester, MN, USA. ✉e-mail: iijima@met.nagoya-u.ac.jp; fambrosio@mgh.harvard.edu

of the molecular mechanisms driving disease development. This is particularly true for knee OA (KOA) since, in most cases, there is no clear inciting event, and the single greatest predictive factor for developing KOA is age[3]. To date, the majority of animal studies have utilized a post-traumatic model of OA (PTOA) in young male mice[4]. However, this post-traumatic model displays a distinct molecular signature when compared to age-associated KOA[5]. Thus, it is not surprising that the majority of pre-clinical findings in KOA have failed to translate to effective treatment for a population consisting largely of aged individuals with no joint trauma.

In this study, we first thoroughly characterized the trajectory of structural and proteomic changes associated with KOA in mice across the lifespan and according to sex. We directly compared these findings to our current understanding of KOA in humans. We next examined mechanistic drivers of KOA via a series of genetic and pharmacologic manipulations. Our findings identified the longevity protein, α-Klotho, as a regulator of chondrocyte health in both murine and human cartilage. Further interrogation using in vitro engineered models revealed that an age-related increase in matrix stiffness compromises chondrocyte integrity and declines α-Klotho expression. Notably, this inhibition of α-Klotho was attributed to *Klotho* promoter methylation that was concomitant with increased binding of DNA methyltransferase 1 (DNMT1). Finally, pharmacological reduction of cartilage stiffness in aged mice increased α-Klotho levels and restored cartilage integrity. Taken together, these findings suggest that age-related alterations in matrix stiffness initiate pathogenic mechanotransductive signaling cascades, leading to epigenetic repression of the *Klotho* promoter and compromised chondrocyte health.

## Results

### Aging-induced cartilage degeneration is accompanied by disruption of PI3K/Akt signaling in male, but not female, mice

We first thoroughly evaluated cartilage integrity in three age groups of male and female C57/BL6 mice: young (4–6 months), middle-aged (10–14 months), and aged (18–24 months). These age groups correspond to 20–30, 38–47, and 56–69 years of age in humans, respectively[6]. We focused on the medial tibial cartilage given this is the region most commonly affected by KOA in humans[7].

Histological observations confirmed progressive cartilage degeneration beginning at middle age (Fig. 1A). This is in line with clinical reports of structural abnormalities that manifest in the cartilage of 45–60-year-old individuals[8,9]. To further examine cartilage degeneration over time, we quantified cartilage surface roughness by assessing the deviation of the cartilage surface from a fitted curve (Fig. S1A). As expected, surface roughness significantly increased with aging (Fig. S1B, C). Interestingly, the magnitude of cartilage degeneration and surface roughness was greater in male mice than in female mice (Fig. 1A, Fig. S1B, C). It is unclear why female mice appear to display relative protection against KOA. However, it is worth noting that serum estrogen levels were unchanged across the three female age groups, indicating that aged female mice do not experience the same menopausal phenotype seen in humans (Fig. S1D). Previous studies have demonstrated that ~65% of aged female mice spontaneously transition to a polyfollicular anovulatory state, ultimately displaying pre-menopausal estrogen and progesterone profiles[10]. Given that post-menopausal women typically present with more severe KOA than men[11], our findings suggest that utilizing female mice without induction of menopause may not be reliable for modeling age-related KOA in human females.

We next performed mass spectrometry proteomics to identify potential signaling pathways associated with age- and sex-dependent cartilage degeneration. Articular cartilage was collected from young, middle-aged, and aged male and female mice ($n = 5$/age/sex; Fig. 1B). From these samples, we identified 44,689 peptides associated with 6694 unique proteins. Data are available via CalTech's Proteomics Repository (https://doi.org/10.22002/ee2yc-fg857; see Methods for details).

To gain a holistic understanding of the protein network-level changes in knee cartilage with aging and according to sex[12], we performed Kyoto Encyclopedia of Genes and Genomes (KEGG) enrichment analyses[13]. Similar to histological observations, male mice displayed more changes in individual protein expression over time when compared to female counterparts. Whereas only two pathways were significantly enriched between young and aged cartilage samples in female mice, eight pathways were significantly enriched between young and aged cartilage samples in male mice (Fig. 1C). Of these, PI3K/Akt signaling was the only pathway that significantly changed over time starting at middle age (Fig. 1D). Further, total normalized perturbation ($P$), a measure of how much a pathway deviates from physiologic conditions (young mice served as our standard of physiological condition; $P_{young} = 0$)[14], was elevated at middle-age (Fig. 1D). Taken together, these findings support previous reports highlighting disruption of PI3K/Akt signaling in human OA[15].

Examination of individual proteins associated with PI3K/Akt signaling enrichment demonstrated distinctly upregulated and downregulated clusters of biological processes associated with aging. Specifically, gene ontology (GO) enrichment analysis revealed that upregulated proteins were associated with Metabolic Stress, Mechanotransduction, and Apoptosis, whereas downregulated proteins were associated with Matrix Organization, Cell Proliferation, and Protein Metabolism (Fig. 1E).

To expand our systems biology approach to include transcript level changes, we accessed archived RNA-seq data from young (3 months old) and aged (24 months old) C57BL6 mice[16] and compared changes to those observed from mass spectrometry data. To do this, we first identified differentially expressed proteins between young and aged cartilage (yielding 1,057 proteins) and converted the identified proteins to their corresponding genes. When we compared the genes identified through mass spectrometry analysis with differentially expressed genes from young and aged transcriptomic data (yielding 3506 genes), we identified 125 overlapping genes (Fig. S2). KEGG enrichment analysis for the overlapping genes revealed that the PI3K/Akt signaling pathway was among the top five differential pathways.

### Global knockdown of α-Klotho drives cartilage degeneration in male mice

We next probed for candidate proteins that may regulate the observed proteomic changes.

Ingenuity pathway analysis[17] using mass spectrometry data predicted activated INS/INSR signaling as a significant regulator for age-related changes in proteins in male mice cartilage (Fig. S3). INS/INSR is upstream of PI3K/Akt signaling[13]. Therefore, we next sought to identify factors that inhibit the INS/INSR/PI3K axis. One key regulator of this pathway is the longevity protein, α-Klotho (Fig. 1F)[18]. A recent systematic review identified elevated inflammation, increased senescence, and impaired autophagy as common denominators associated with aging-induced KOA in mice[19], all of which are downstream of α-Klotho[20–24]. When we accessed archived microarray data (GSE80285)[25], we found that inhibition of α-Klotho protein significantly activated PI3K/Akt signaling in human primary chondrocytes in vitro (Fig. S4). While α-Klotho overexpression has been shown to delay cartilage degeneration in a PTOA model[25], whether α-Klotho plays a role in age-related KOA model has not been definitively established.

In both mouse and human cartilage, we found that α-Klotho was both decreased with increasing age and associated with more severe cartilage degeneration in males (Fig. 2A, B; Fig. S5A, B). To establish a more direct role of α-Klotho on cartilage health, we evaluated cartilage integrity in mice heterozygous for the *Klotho* gene (*Klotho*[+/−]). Young and middle-aged *Klotho*[+/−] male mice displayed significantly accelerated cartilage degradation (Fig. 2C), consistent with an aged phenotype. However, accelerated cartilage degradation was not observed in female young and middle-aged *Klotho*[+/−] mice (Fig. 2C). Sex differences in cartilage degeneration in *Klotho*[+/−] mice were further supported by

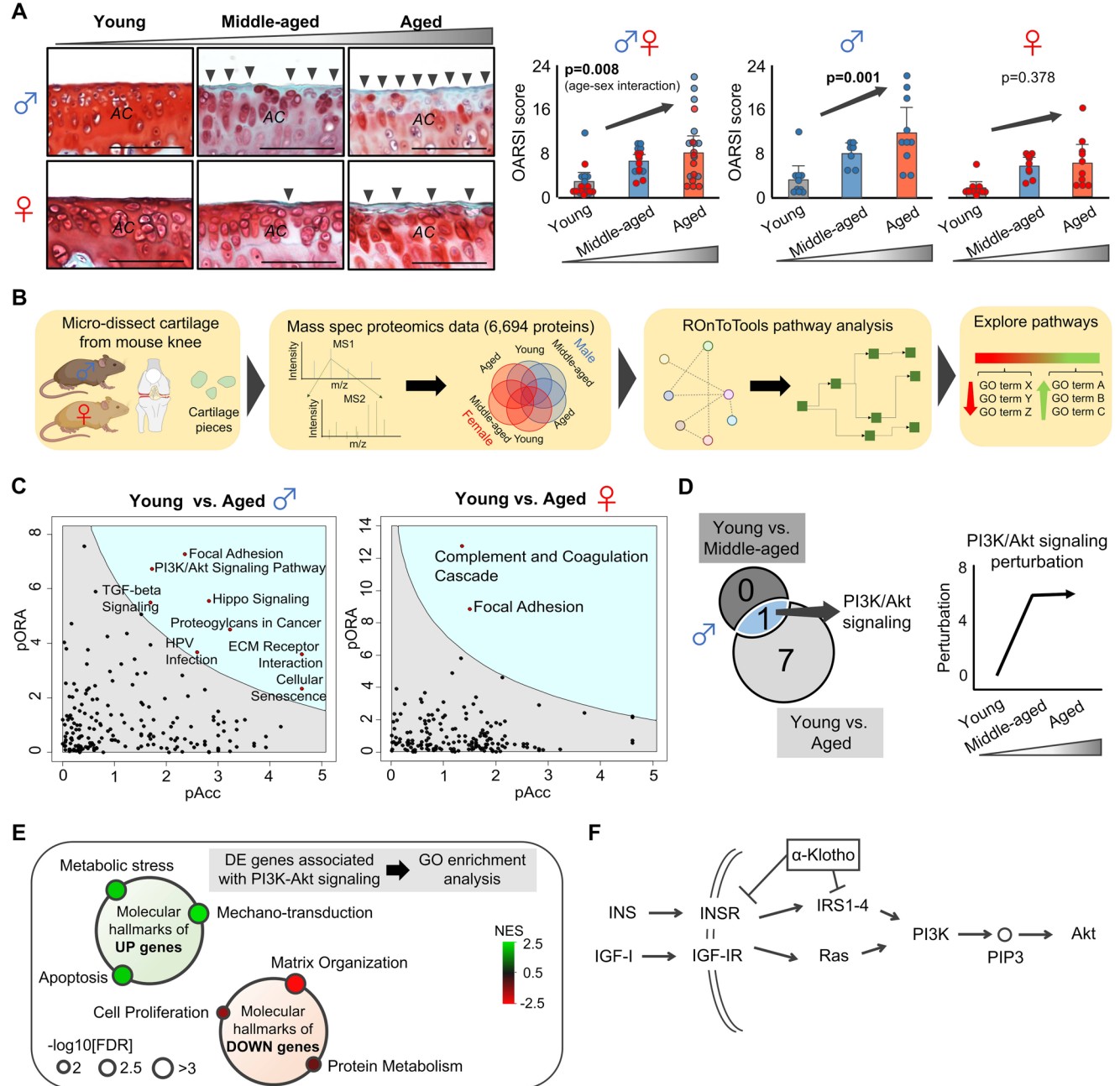

**Fig. 1 | Male, but not female, mice display age-related cartilage degeneration and disruption of the PI3K-Akt signaling pathway. A** Aging induced progressive cartilage degeneration in murine medial tibial plateaus in a sex-dependent manner. Representative histological sections stained with Safranin O/Fast Green are provided for each group. Black arrow heads indicate loss of cartilage matrix. The Osteoarthritis Research Society International (OARSI) score is provided (0–24 points; higher value indicates more severe cartilage degeneration), as assessed by a blinded assessor (male, $n = 10$ for young and aged, $n = 7$ for middle-aged; female, $n = 10$ for young and aged, $n = 9$ for middle-aged). Statistical analyses were performed using linear regression. Data are presented as means ± 95% confidence intervals. Scale bar: 50 µm. **B** Schematic showing the experimental protocol for

mass spectrometry. Knee cartilage was microdissected from young, middle-aged, and aged male and female mice ($n = 5$/sex/age). Individual proteins were identified and used for KEGG pathways analysis. Proteins from PI3K/Akt signaling were then grouped based on GO terms. **C** Pathway analysis for young vs. aged in male and female mice. pAcc: the Boolean value of total bootstrap permutations accumulation; pORA: the Boolean value of over-representation $p$ value. **D** Comparison between young vs. aged and young vs. middle-aged pathway analyses in male mice, and display of perturbation calculations. **E** Molecular hallmarks of upregulated and downregulated genes associated with PI3K/Akt signaling. **F** Schematic pathway of PI3K/Akt signaling and linking to α-Klotho. Portions of the figures were created with biorender.com. Source data are provided as a Source Data file.

computational histological image analysis demonstrating that male *Klotho*[+/−] mice displayed higher cartilage surface roughness (Fig. S6A, B), mimicking the phenotype of aged wild-type male mice. Female *Klotho*[+/−] mice did not display any change in cartilage integrity compared to sex- and age-matched counterparts (Fig. S6A, B). These findings suggest that loss of α-Klotho may be a driver of age-related KOA in a sex-dependent manner. Given the limited KOA phenotype

seen in aged female mice, only male mice were used for the subsequent experiments.

## Age-related declines in α-Klotho are associated with alterations in nuclear mechanics of mouse and human chondrocytes

*What drives a decline in chondrocyte α-Klotho expression over time?* Our proteomic data revealed that mechanotransductive signaling was

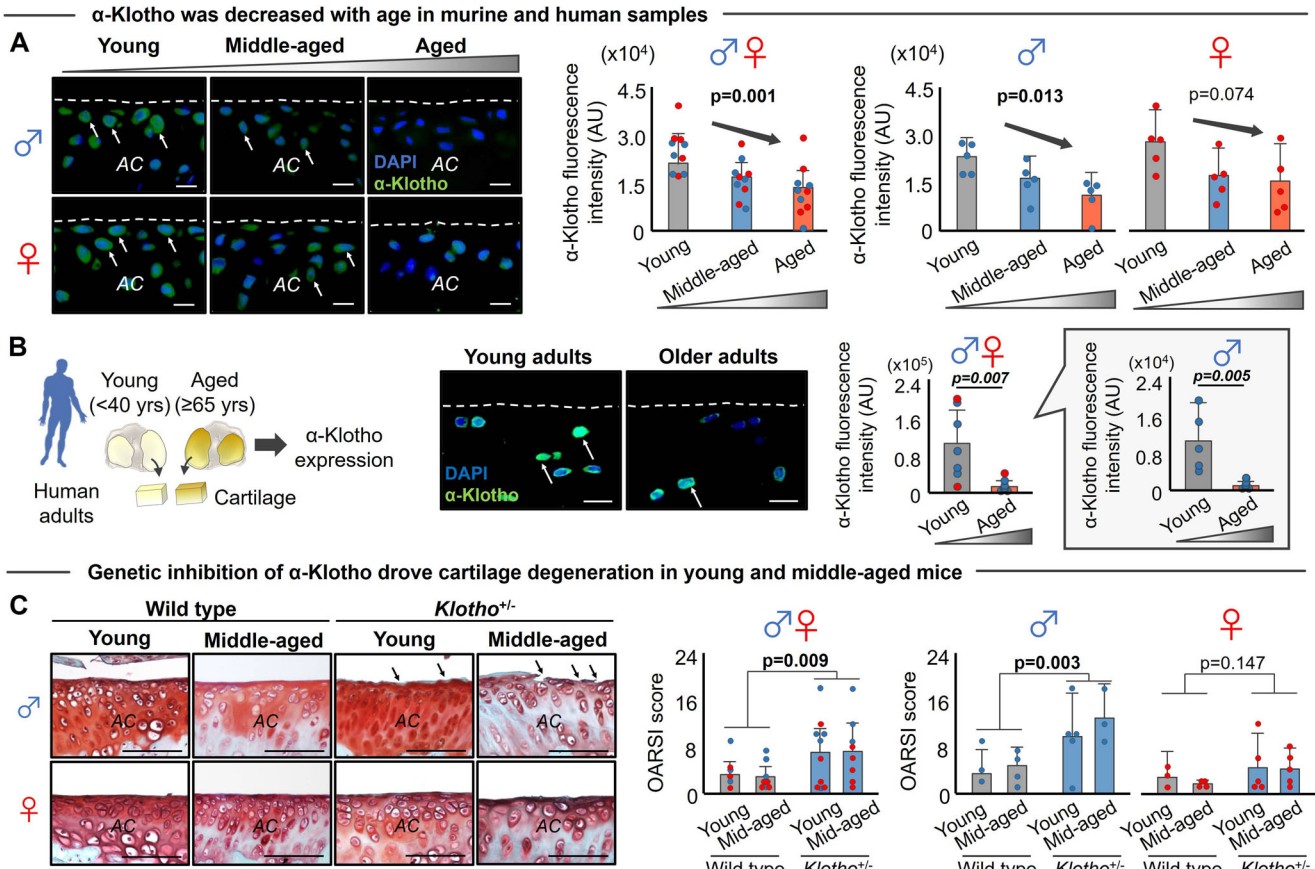

**Fig. 2 | Age-related declines in α-Klotho are associated with cartilage degeneration in male mice. A** Aging induced a progressive decline of α-Klotho in the murine medial tibia (*n* = 5/sex/age). White arrows indicate α-Klotho-positive chondrocytes. White dashed lines indicate cartilage surface. AC articular cartilage. Scale bar: 10 μm. α-Klotho expression per cell was quantified by immuno-fluorescence (50–100 cells per mouse). **B** Cartilage in older adults (≥65 years old; 71.9 ± 2.91 years; *n* = 7 [1 female]) displayed reduced α-Klotho expression compared to young adults (<40 years old; 27.6 ± 6.85 years; *n* = 7 [2 females]). Scale bar: 20 μm. α-Klotho expression per cell was quantified by immunofluorescence (30–70 cells per cartilage sample). **C** Loss-of function in *Klotho* (*Klotho*[+/−]) triggered murine cartilage degeneration in a sex-dependent manner (young, *n* = 7 for wild-type [3 females], *n* = 10 for *Klotho*[+/−] [5 females]; middle-aged, *n* = 10 for wild-type [5 females], *n* = 8 for *Klotho*[+/−] [5 females]). Black arrows indicate cartilage surface disruption. AC articular cartilage. Scale bar: 50 μm. Statistical analyses were performed using linear regression (**A**), two-way ANOVA (**C**), and a two-tailed Student *t* test (**B**). Age-sex interaction was not significant for α-Klotho expression in mice (**A**, *p* = 0.468). Data are presented as means ± 95% confidence intervals. Source data are provided as a Source Data file.

significantly disrupted in aged murine cartilage (Fig. 1C). This led us to question whether age-related declines in α-Klotho are a consequence of altered mechanical signals from the microenvironment. Aging is correlated with progressive alterations in nuclear mechanics and nuclear envelope dysfunction, driving chromatin remodeling and gene expression changes[26,27]. The nuclear envelope is primarily composed of a nuclear lamina (e.g., lamin A/C and lamin B) and a bilayer membrane through the linker of the nucleoskeleton and cytoskeleton complex. In particular, lamin A/C is a well-known regulator of nuclear integrity[28,29]. With this in mind, we revisited the mass spectrometry data to characterize age-related changes in nuclear envelope elements (Fig. 3A). We found that protein abundance of lamin A/C and lamin B2 was significantly increased in aged mice compared to young counterparts (Fig. 3B).

To further characterize age-associated alterations in the nuclear envelope, we quantified chondrocyte nuclear morphology in cartilage tissue using CellProfiler software[30] (Fig. 3C, D). CellProfiler measures the size, shape, intensity, and texture of a variety of nuclear/cell types in a high throughput manner. Our study focused on 53 morphological variables representing nuclear shape and geometry. Principal component analysis (PCA) for 53 morphological variables revealed clear segregation of nuclei from young and aged chondrocytes in mice (Fig. 3E). Of the top 10 variables contributing to the first principal component (Fig. 3F), nuclear eccentricity (or sphericity) was found to

be a highly sensitive nuclear morphological marker associated with aging in both murine and human samples (Fig. 3G). A detailed description for the top 10 variables is shown in Table S1. Notably, increased nuclear eccentricity (i.e., a less spherical nucleus) was also significantly associated with decreased α-Klotho protein levels (Fig. 3H). Further supporting the relationship between nuclear morphology and α-Klotho, the age-related increase in protein abundance of lamin A/C was not only associated with nuclear eccentricity (Fig. S7A) but was also inversely correlated with fluorescence intensity of α-Klotho (Fig. S7B). These results suggest that a decline in α-Klotho levels may be attributed to altered nuclear mechanics (Fig. 3I).

### Increased matrix stiffness induces a decline in α-Klotho and an aged phenotype in young chondrocytes

The nucleus is mechanically coupled to the microenvironment where cytoskeletal elements regulate nuclear shape according to extrinsic ECM stiffness cues[31,32]. Similar to other tissues throughout the body[33], cartilage stiffness in aged mice and humans is ~2–3 times higher than young counterparts[34]. Therefore, to evaluate the direct effect of increased ECM stiffness on chondrocyte phenotype, we seeded primary mouse chondrocytes onto polyacrylamide (pAAm) gels of different stiffnesses (5 kPa, 21 kPa, and 100 kPa; Fig. 4A)[35]. This stiffness range was selected to mimic physiologically-relevant ECM stiffnesses of young (1–30 kPa) and aged (50–100 kPa) murine and human knee cartilage[34].

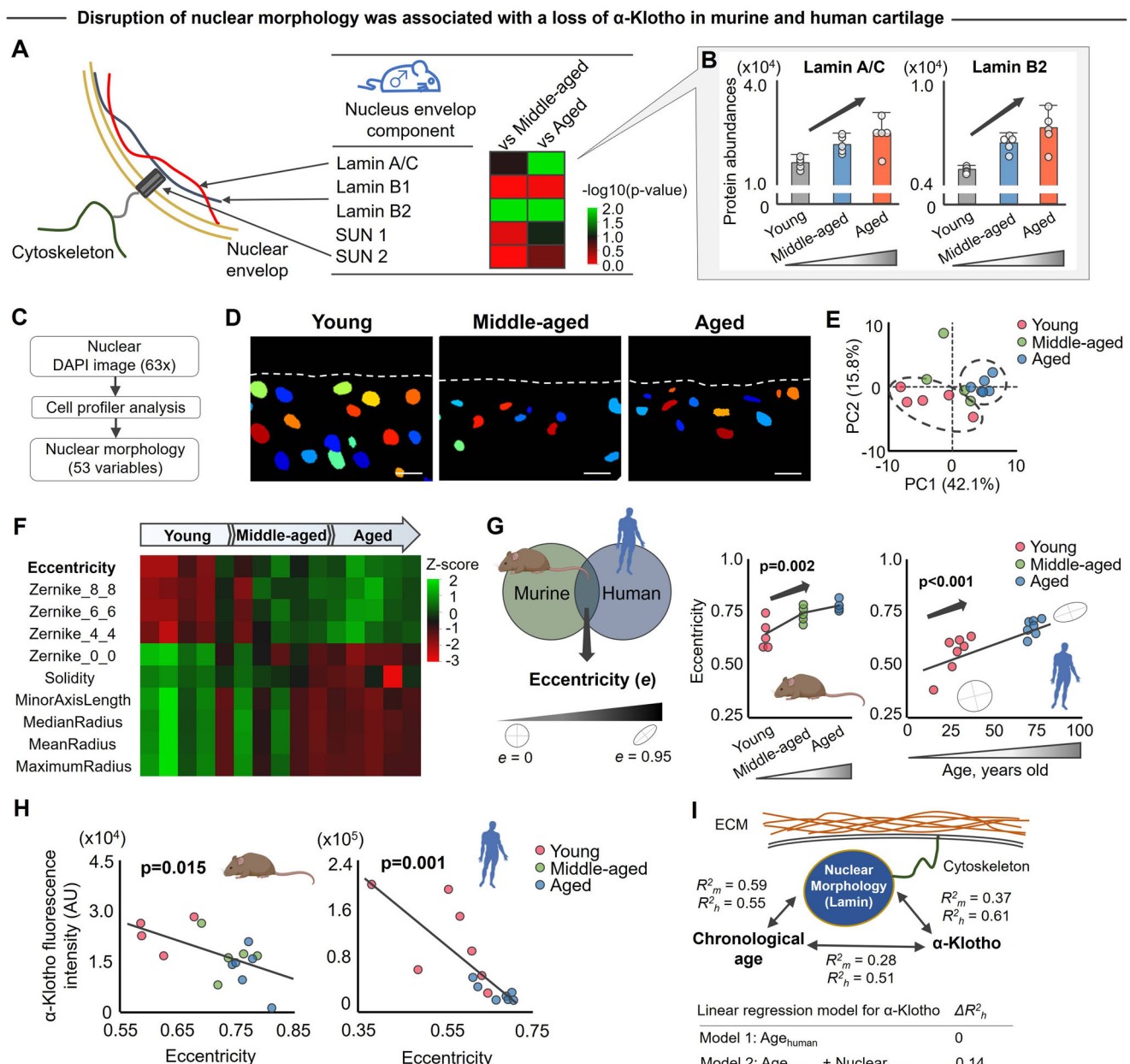

**Fig. 3 | Age-related α-Klotho decline is accompanied with nuclear morphological alterations. A** Age-related changes in nuclear envelope proteins in male mice quantified by mass spectrometry-based proteomics. The heat map represents log10 (p value) of each protein, with >1.3 considered significant. **B** Aging increased lamin A/C and lamin B2 protein expression (n = 5/age). **C** Analytical flow of the nuclear morphological analysis (53 variables) using CellProfiler software. **D** Representative nuclear images in murine medial tibia generated by CellProfiler, which was reproducible in another set of images. Dashed white lines indicate the cartilage surface. Scale bar: 10 µm. **E** Principal component analysis (PCA) of nuclear morphology characteristics showing separation of clusters between young and aged animals (n = 5/age; 30–60 nuclei per individual mice). **F** Heat map of the top 10 nuclear morphological variables contributing principal component 1. **G** Murine (n = 5/age) and human cartilage (n = 7/age) share changes in nuclear eccentricity, with higher age associated with increased nuclear eccentricity (i.e., less roundness). **H** Age-related increased nuclear eccentricity is associated with decreased α-Klotho expression in murine (n = 5/age) and human cartilage (n = 7/age). **I** Schematic showing the relationship between chronological age, α-Klotho expression, and nuclear morphology. Linear regression model shows the substantial and independent contribution of nuclear morphology (Nuclear_human) in the prediction of α-Klotho expression level in human cartilage beyond the effect of chronological age (Age_human). Coefficient of determinations are provided for mouse ($R^2_m$) and human cartilage ($R^2_h$). Statistical analysis was performed using linear regression (**B**, **G**, **H**, **I**). Data are presented as means ±95% confidence intervals. Portions of the figure were created with biorender.com. Source data are provided as a Source Data file.

Young chondrocytes cultured on stiff substrates developed an aged phenotype when compared to cells cultured on soft substrates, as evidenced by reduced type II collagen and aggrecan (Fig. 4A, B). These changes were accompanied by decreased α-Klotho protein levels (Fig. 4A, B). Conversely, aged chondrocytes cultured on soft substrates displayed increased type II collagen and α-Klotho compared to those cultured on stiff substrates (Fig. S8A, B). In these analyses, we confirmed that lamin A/C, but not lamin B2, was both highly sensitive to substrate stiffness (Fig. S9A, B) and positively associated with nuclear deformation (Fig. S9C, D); these results are consistent with previous reports[31,32,36,37]. Aged-like (stiff) substrates also increased stress fiber formation and altered chondrocyte cellular morphology (Fig. S10), thereby further implicating mechanotransductive signaling in the stiffness-dependent regulation of cellular phenotype. To directly

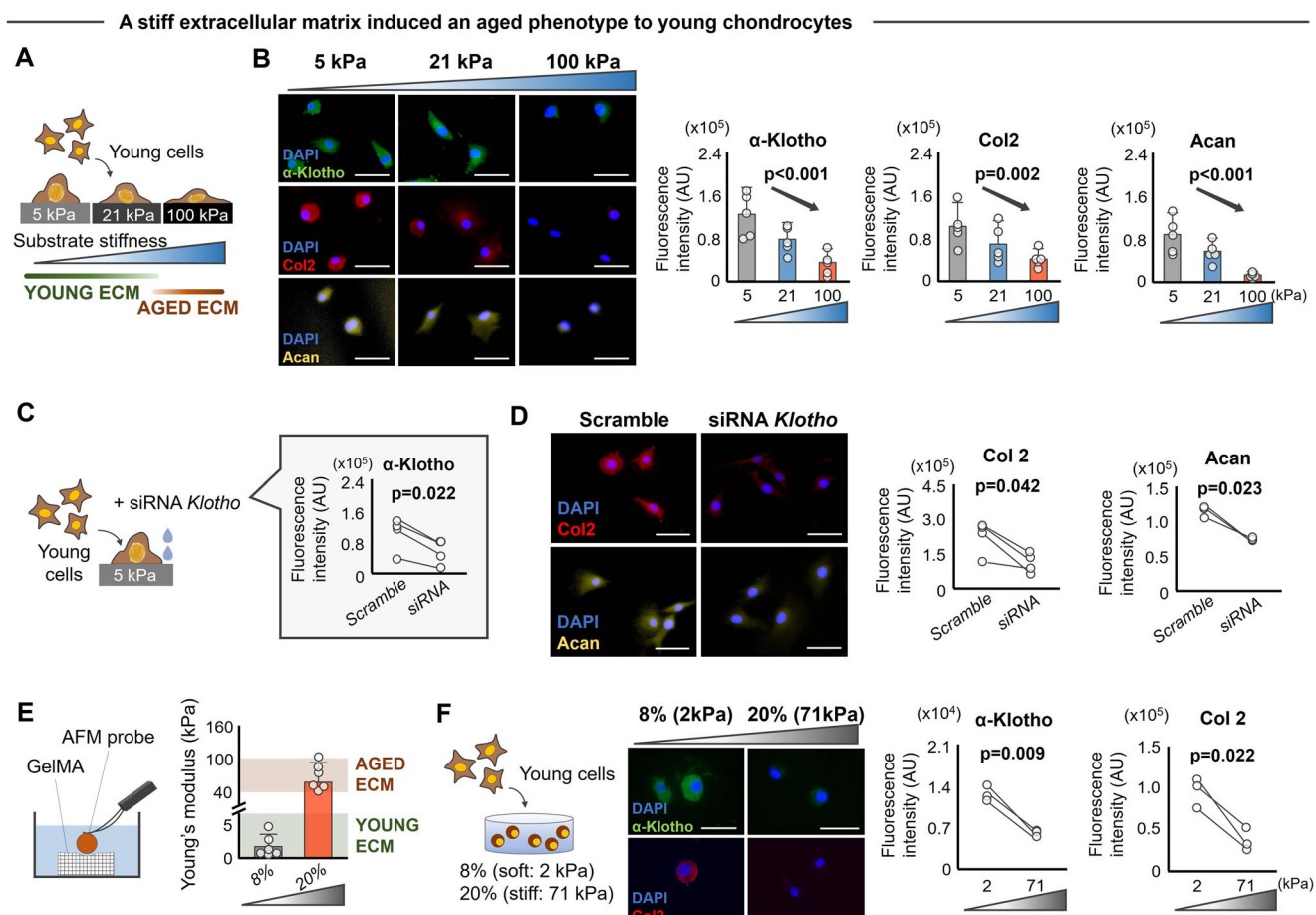

**Fig. 4 | Matrix stiffness regulates chondrocyte health via α-Klotho. A** Schematic showing the experimental protocol. Primary chondrocytes isolated from young murine cartilage were seeded onto polyacrylamide gels engineered to mimic a physiological range of cartilage matrix stiffness (5 kPa, 21 kPa, and 100 kPa). **B**. Stiff substrates reduced α-Klotho, type II collagen (Col2), and aggrecan (Acan) in young chondrocytes, as quantified by immunofluorescence ($n = 5$/group; 30–70 cells per individual sample). Scale bar: 50 μm. **C, D** siRNA *Klotho* treatment inhibited the beneficial effect of soft substrates on type II collagen and aggrecan expression, as quantified by immunofluorescence ($n = 3$/group; 20–40 cells per individual sample). Scale bar: 50 μm. **E** Measurement of matrix stiffness (Young's modulus) for 8% and 20% GelMA ($n = 6$/group) using atomic force microscopy (AFM). Young's modulus for young and aged articular cartilage was provided as reference values[34]. **F** A stiff 3D environment reduced α-Klotho, type II collagen, and aggrecan in chondrocytes quantified by immunofluorescence ($n = 3$/group; 30–60 cells per individual sample). Scale bar: 20 μm. Statistical analyses were performed using linear mixed effect model (**B**) or two-tailed paired *t* test (**C, D, F**). Data are presented as means ±95% confidence intervals. Source data are provided as a Source Data file.

test whether matrix stiffness regulates chondrogenicity via α-Klotho, siRNA to *Klotho* was added to young chondrocytes cultured on soft substrates (Fig. 4C). We found that siRNA inhibition of α-Klotho overrode the pro-chondrogenic effect of a soft matrix, as evidenced by a decrease in type II collagen and aggrecan expression (Fig. 4D).

Several studies have shown that ascorbic acid, an essential cofactor for lysyl hydroxylase and prolyl hydroxylase, is required for collagen biosynthesis in vitro[38]. Therefore, we performed a sensitivity analysis to determine whether the stiffness-dependent regulation of chondrogenicity markers (type II collagen and aggrecan expression) was influenced by the addition of ascorbic acid to the culture medium. We found that the expression of chondrogenic markers was not significantly influenced by the presence of ascorbic acid (Fig. S11A, B). When we assessed the secretion of type II collagen by chondrocytes under each of the experimental conditions (+/− ascorbic acid on soft, medium, or stiff substrates) using an enzyme-linked immunosorbent assay (ELISA), we found that chondrocytes cultured on stiff substrates secreted less type II collagen compared to those cultured on soft substrates, an effect that was independent from ascorbic acid supplementation (Fig. S11C). We note that the number of chondrocytes was similar across the groups, confirming that the reduced secretion of

type II collagen on stiff substrates was not a function of differences in chondrocyte confluence (Fig. S11D).

To determine whether similar changes occur in a three-dimensional (3D) microenvironment, which more closely recapitulates the native microenvironment, we fabricated gelatin-based hydrogels (GelMA). Atomic force microscopy (AFM) was used to confirm that the stiffness of the engineered soft (2 kPa) and stiff (71 kPa) hydrogels approximated the stiffness of native young and aged cartilage, respectively (Fig. 4E). Consistent with our findings using a two-dimensional (2D) culture, young chondrocytes cultured in a stiff three-dimensional environment displayed reduced α-Klotho and type II collagen levels (Fig. 4F). Taken together, our 2D and 3D models suggest that soft substrates promote a more youthful phenotype in aged chondrocytes but that stiff substrates accelerate the effects of time in young chondrocytes.

## Matrix stiffness epigenetically regulates α-Klotho expression in chondrocytes

It is well known that mechanical forces induce modulation of nuclear chromatin structures and epigenetic landscapes to impact gene transcription[37,39–41]. We, therefore, examined whether a loss of α-Klotho in response to stiff substrates is driven by epigenetic modifications. To do this, we measured methylation levels of the *Klotho*

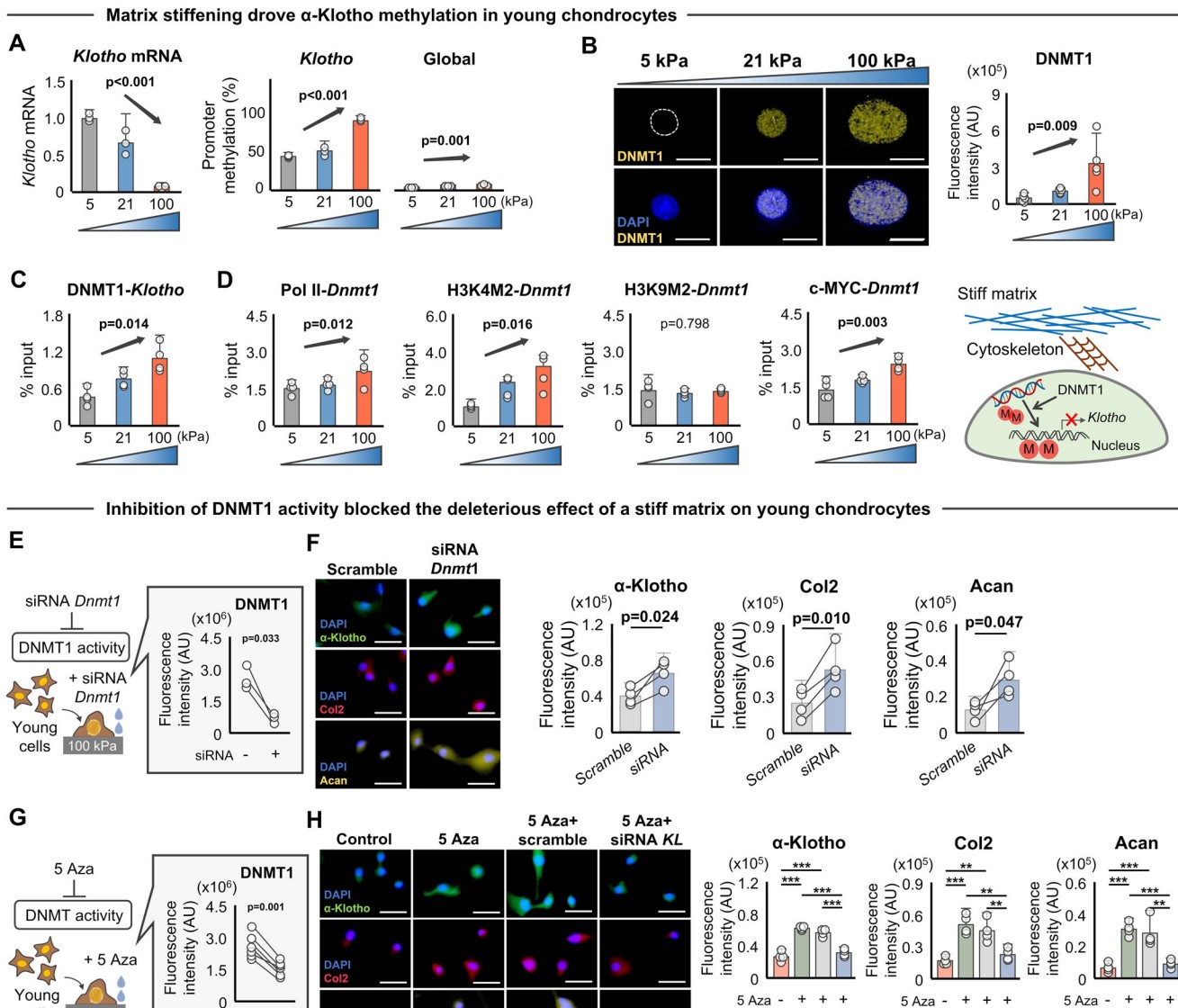

**Fig. 5 | Epigenetic regulation of α-Klotho by stiff matrix disrupts chondrocytes health. A** Stiff substrates downregulated *Klotho* expression and increased *Klotho* promoter methylation as well as global DNA methylation in young chondrocytes (*n* = 3/group). **B** Stiff substrates increased DNMT1 expression in young chondrocytes as quantified by immunofluorescence (*n* = 5/group; 30–50 cells per individual sample). Scale bar: 20 µm. **C** Stiff substrates increased binding of DNMT1 at *Klotho* promoter in young chondrocytes quantified by chromatin immunoprecipitation (ChIP) analyses (*n* = 4/group). **D** Stiff substrates increased binding of RNA Polymerase II (Pol II), active chromatin mark H3K4M2, and c-MYC, but not the repressive chromatin mark, H3K9M2, at the *Dnmt1* promoter in young chondrocytes, as quantified by ChIP analyses (*n* = 4/group). **E**, **F** siRNA *Dnmt1* treatment inhibited the deleterious effect of a stiff microenvironment on α-Klotho, type II

collagen (Col2), and aggrecan (Acan) expression, as quantified by immunofluorescence (*n* = 4/group; 40–50 cells per individual sample). Scale bar: 50 µm. **G**, **H** 5-Aza-2′-deoxycytidine (5 Aza) treatment inhibits the deleterious effect of a stiff microenvironment on α-Klotho, Col2, and Acan expression. The effects of 5 Aza treatment were blocked when combined with siRNA *Klotho* (siRNA *KL*) treatment. Data were quantified by immunofluorescence (*n* = 4/group; 30–45 cells per individual sample). Scale bar: 50 µm. Statistical analyses were performed using a linear mixed effect model (**A**–**D**), two-tailed paired *t* test (**E**–**G**) or analysis of variance with post-hoc Tukey–Kramer test (**H**). ***p* < 0.01, ****p* < 0.001. Data are presented as means ±95% confidence intervals. Source data are provided as a Source Data file.

promoter in young or aged chondrocytes cultured on substrates with varying stiffnesses. Stiff substrates triggered methylation of the *Klotho* promoter and decreased *Klotho* gene expression in young chondrocytes (Fig. 5A). Conversely, soft substrates significantly reduced methylation and increased *Klotho* gene expression in aged chondrocytes (Fig. S12A). Culturing young and aged chondrocytes on stiff substrates also promoted global DNA methylation, though the magnitude of effect was considerably lower than the effect observed on the *Klotho* promoter (Fig. 5A and Fig. S12A). Stiffness-dependent methylation of the *Klotho* promoter was further supported by a

corresponding increase in DNMT1 protein levels (Fig. 5B and Fig. S12B). DNMT1 is an enzyme that catalyzes the transfer of methyl groups to CpG dinucleotides, usually repressing gene transcription by altering chromatin structure and blocking the access of transcription factors at the gene regulatory region[42]. Further inspection by chromatin immunoprecipitation (ChIP) analysis revealed that increased ECM stiffness was associated with increased binding of DNMT1 at the *Klotho* promoter (Fig. 5C), suggesting that increased *Klotho* promoter methylation under conditions of a stiff microenvironment may be attributed, at least in part, to the increased binding of DNMT1. Of note, increasing

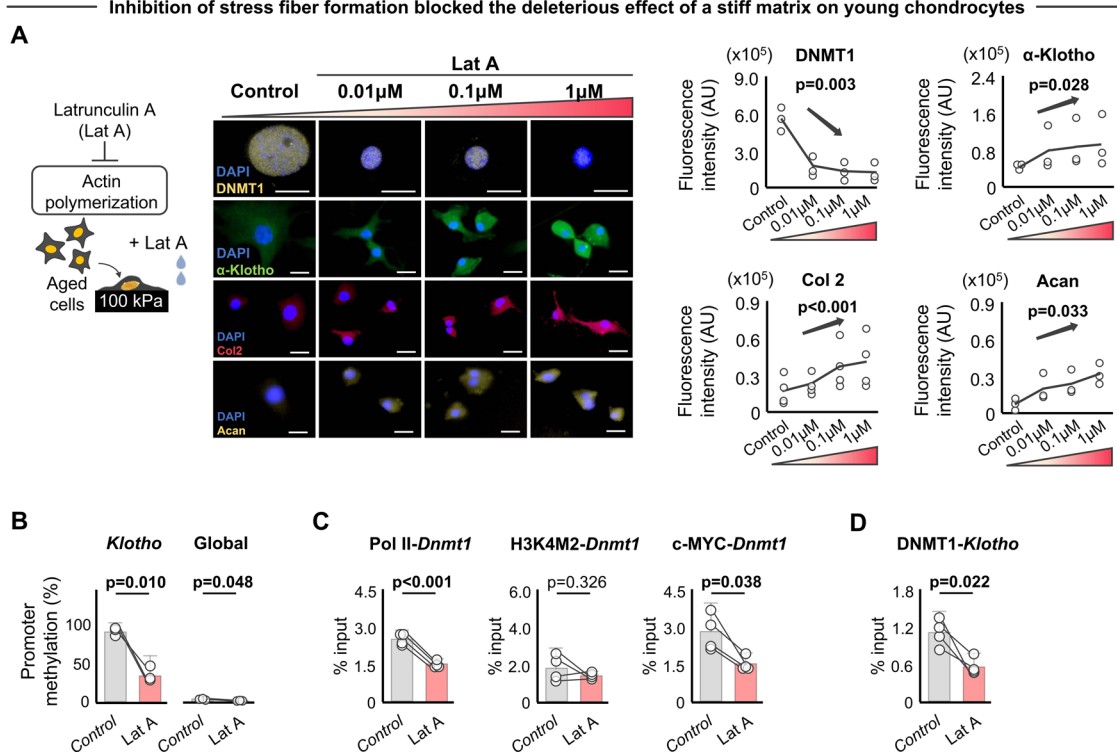

**Fig. 6 | Mechanotransductive signals delivered by stiff substrates drive epigenetic repression of α-Klotho. A** Latrunculin A, an inhibitor of actin polymerization, rescued stiff matrix-induced increases in DNMT1 and reduction of α-Klotho, type II collagen (Col2), and aggrecan (Acan) in aged chondrocytes, as quantified by immunofluorescence (n = 3/group except for Col2 [n = 4/group]; 30–60 cells per individual sample). Scale bar: 20 μm. **B** Inhibition of actin polymerization blocked the stiff substrate-driven increased *Klotho* promoter and global DNA methylation in aged chondrocytes (n = 3/group). **C** Inhibition of actin polymerization blocked the stiff substrate-driven increased binding of RNA Polymerase II (Pol II), c-MYC, but not H3K4M2, at *Dnmt1* promoter in aged chondrocytes as quantified by chromatin immunoprecipitation (ChIP) analyses (n = 4/group). **D** Inhibition of actin polymerization blocked the stiff substrate-driven increased binding of DNMT1 at *Klotho* promoter in aged chondrocytes as quantified by ChIP analyses (n = 4/group). Statistical analyses were performed using a linear mixed effect model (**A**) or two-tailed paired *t* test (**B–D**). Data are presented as means ± 95% confidence intervals. Source data are provided as a Source Data file.

ECM stiffness also showed a decreased binding of RNA Polymerase II (Pol ll) at the *Klotho* promoter, which was accompanied by decreased active chromatin mark (H3K4M2) and increased repressive chromatin mark (H3K9M2), (Fig. S13). These ChIP analyses support our general understanding that DNMT transcriptional changes are coupled with chromatin remodeling and an imbalance of activating and repressive histone modifications[43].

An additional series of ChIP analyses revealed that stiff substrates induced recruitment of Pol II, H3K4M2, but not H3K9M2 at the *Dnmt1* promoter (Fig. 5D), contributing to the increased DNMT1 expression. We also observed a significant increase in binding of c-MYC at the *Dnmt1* promoter in response to increased ECM stiffness (Fig. 5D). c-MYC is a transcription factor known to be stimulated and/or regulated by kinase signaling[44]. These findings indicate that *Dnmt1* is directly associated with substrate stiffness-mediated signaling. Notably, inhibition of *Dnmt1* by siRNA rescued the deleterious effect of a stiff matrix on both chondrocyte α-Klotho levels and chondrogenicity (Fig. 5E, F).

Given our observation that a stiff matrix-induced global DNA methylation changes, we cannot exclude the possibility that the effect of DNMT1 on chondrocyte health is independent of α-Klotho. To more definitively implicate DNMT1 as a regulator of α-Klotho and, ultimately, chondrocyte health, we performed simultaneous inhibition of α-Klotho by siRNA *Klotho* and DNMT by 5-Aza-2'-deoxycytidine (5 Aza). Results revealed that intercepting upregulation of *Klotho* expression following pharmacological inhibition of DNMT abrogated the beneficial effect of the DNMT inhibition alone (Fig. 5G, H). These results suggest that an aged-like stiff substrate mediates decreased

chondrogenicity through inhibition of *Klotho*. Of note, treating chondrocytes with siRNA for *Dnmt1* did not significantly alter nuclear eccentricity (Fig. S14A–C), suggesting that age-related nuclear deformation is either independent or upstream of DNMT1.

Since extrinsic mechanical signals from the microenvironment are transmitted to the nucleus via the actin cytoskeleton[45], we next tested whether disruption of the cytoskeleton inhibits the impact of a stiff matrix on downstream chondrocyte responses. For this, aged cells cultured on stiff substrates were treated with Latrunculin A (lat A), an inhibitor of actin polymerization. We confirmed that lat A treatment abolished stress fiber formation, decreased nuclear eccentricity (i.e., increased roundness), and decreased lamin A/C expression without influencing chondrocyte viability (Fig. S15A–C). Indeed, we found that reduced actin polymerization in aged chondrocytes cultured on stiff substrates decreased DNMT1 expression and increased α-Klotho levels, ultimately restoring type II collagen and aggrecan levels (Fig. 6A). Increased α-Klotho levels following lat A treatment were also accompanied by a decrease in methylation level of the *Klotho* promoter, while global DNA methylation was not drastically altered (Fig. 6B). Interestingly, the increased Pol II and c-MYC bindings at *Dnmt1* promoter induced by stiff substrates (shown in Fig. 5D) were abolished under lat A treatment conditions (Fig. 6C). We interpret these data to mean that the transcriptional machinery of *Klotho*/*Dnmt1* may be epigenetically regulated by mechanical signals. Treatment with lat A also abolished the change in binding of Pol II, H3K4M2, and DNMT1 at the *Klotho* promoter when cells were cultured on a stiff substrate (Fig. 6D and Fig. S15D). Treatments with Y-27632, an inhibitor of actin fiber formation, and PF-562271, an inhibitor of focal adhesion

kinase, similarly abolished the observed increase in DNMT1 and decrease in α-Klotho when chondrocytes were seeded on a stiff substrate (Fig. S16A, B).

## Reduction of ECM stiffness increases α-Klotho expression and improves cartilage health in aged mice

Our in vitro findings raise the possibility that targeting ECM mechanics by preventing or reversing matrix stiffening may be a potential therapeutic approach for age-related KOA. As described above, pathway analysis from mass spectrometry data revealed that aged cartilage displayed enrichment of ECM-related pathways, including Focal Adhesion, ECM Receptor Interaction, and Hippo signaling (Fig. 1C). Therefore, to identify potential target ECM proteins driving age-related ECM stiffening in vivo, we revisited the mass spectrometry data in search of ECM proteins associated with increased lamin A/C. PCA revealed that aged cartilage displayed an altered expression profile in 155 ECM-related proteins (Fig. 7A). Lysyl oxidase (LOX) emerged in the top 20 proteins contributing to PC2 and was positively correlated with lamin A/C (Fig. 7B, C). This finding is of particular interest given that LOX is one of the major enzymes involved in collagen cross-linking[46] and has been shown to contribute to PTOA pathogenesis by increasing ECM stiffness[47].

Cartilage stiffness increases by 2–3 fold with aging[34]. We, therefore, tested whether reducing the cartilage stiffness in aged mice improves α-Klotho levels and cartilage integrity. For this purpose, we administered β-aminopropionitrile (BAPN), a known inhibitor of LOX[48], as a physiological means to reduce cartilage stiffness in aged mice (Fig. 7D). We confirmed by AFM that four weeks of daily administration of BAPN significantly reduced stiffness of cartilage in the medial tibia (Fig. 7E, F). Histological analysis demonstrated that chondrocytes in aged mice treated with BAPN displayed a more spherical nuclear morphology, similar to young mice (Fig. 7G). PCA analysis further confirmed that chondrocytes from aged mice treated with BAPN exhibited a more youthful phenotype (Fig. 7H). Further, consistent with our in vitro findings, reducing matrix stiffness of aged cartilage in vivo significantly increased α-Klotho levels and improved cartilage integrity (Fig. 7I, J). However, the beneficial effect of BAPN on cartilage integrity was not observed in *Klotho*[+/−] mice (Fig. 7K), suggesting that improved cartilage integrity after BAPN injection was driven, at least in part, by increased α-Klotho levels. We note that BAPN supplementation in vitro did not influence α-Klotho levels in aged chondrocytes cultured on aged-like stiff substrates (Fig. S17), indicating that the increased α-Klotho levels after BAPN administration in vivo is not likely to be explained by a direct effect of BAPN on α-Klotho. These findings are consistent with a previous study showing that the effects of BAPN on chondrocyte function are primarily mediated by mechanotransductive mechanisms[46].

## Discussion

The objective of this work was to provide novel and translational insights into pathogenic mechanisms underlying age-related KOA. A graphical abstract summarizing our findings is shown in Fig. 8. Mass spectrometry proteomic analyses of murine cartilage with a subsequent genetic loss-of-function approach revealed that α-Klotho induces cartilage degeneration in mice in a sex-dependent manner. This age-related decrease in α-Klotho expression was also confirmed in human cartilage. In search of drivers of the decline in chondrocyte α-Klotho levels over time, we found that α-Klotho is highly responsive to biophysical signals from the surrounding ECM. Whereas increased matrix stiffness decreased α-Klotho expression, reducing matrix stiffness increased α-Klotho expression and improved chondrocyte health in vitro. The stiffness-dependent modulation of α-Klotho was attributed to the recruitment of epigenetic modifiers, including DNMT1, at the *Klotho* promoter. As a demonstration of the physiological

relevance of these in vitro findings, we presented evidence of enhanced cartilage integrity in vivo following pharmacological reduction of cartilage matrix stiffness in aged mice. Together, these findings suggest that preventing or reversing age-related matrix stiffening and the resulting pathogenic mechanotransductive signaling may serve as a promising therapeutic target to attenuate or even reverse the development of age-related KOA.

Although there are many studies characterizing tissue-level changes in cartilage with aging, there is a paucity of pre-clinical work examining cartilage integrity according to sex. This is a critical shortcoming, especially given that post-menopausal women present with more severe KOA compared to men[11]. To address this gap, we thoroughly described the trajectory of cartilage degeneration over time and according to sex. Inconsistent with clinical evidence of increased prevalence of KOA in women[11], we found that cartilage integrity was relatively preserved in aged female mice. This disconnect may be due to the fact that the aged female mice used in our study were likely non-menopausal, as suggested by maintained serum estrogen levels over time. Estrogen affects key signaling molecules in several distinct canonical and non-canonical estrogen signaling pathways, such as PI3K/Akt and PKC/MAPK signaling[49]. Whereas the beneficial role of systemic estrogen administration in OA development has been investigated in animal models, the underlying mechanism remains unknown[49]. Our data highlight the need for mechanistic studies into the effects of menopause on KOA in females. Investigating the link between estrogen and matrix-mediated epigenetic regulation is of particular interest given that estrogen modulates mechanotransductive pathways, such as Hippo signaling[50], which we found to be associated with cartilage aging (Fig. 1C).

In search of novel upstream candidates that regulate PI3K/Akt signaling, which we found to be significantly changed over time in male mice, a series of genetic and pharmacologic manipulations revealed declines in α-Klotho as a possible driver of cartilage degeneration. Future studies would benefit from chondrocyte-specific genetic manipulation of α-Klotho in vivo to further evaluate the specificity of this effect. Our results are consistent with previous in vitro and in vivo studies showing that α-Klotho overexpression counteracts chondrocyte dysfunction and cartilage degeneration in a PTOA model[25,51]. Another study also demonstrated that overexpression of α-Klotho, combined with soluble TGF-β supplementation, enhanced chondrocyte health and cartilage integrity in a chemically-induced model of KOA[52]. α-Klotho inhibits insulin growth factor receptor-mediated PI3K/Akt signaling and subsequently enhances FoxO[21,53]. FoxO, which protects chondrocytes from oxidative stress and prevents spontaneous cartilage degeneration[54,55], is markedly reduced with aging in both murine and human cartilage[56]. Our results suggest that the interaction of α-Klotho and FoxO expression to mitigate oxidative stress is an area worthy of future studies.

While age-related declines in α-Klotho have been linked to the onset of an aged tissue phenotype in many organ systems, our understanding of the molecular mechanisms driving these declines is lacking. The ECM plays a dynamic role in regulating cartilage homeostasis and undergoes extensive remodeling with increasing age[34,57]. It is well established that increased matrix stiffness disrupts chondrocyte functionality via mechanotransductive pathways[47,58,59], ultimately leading to cellular senescence. Additionally, biophysical cues influence gene expression through recruitment of epigenetic modifiers or by inducing reorganization of lamin-associated chromatin[37,40,60–64]. Here, we identified matrix stiffness as a potent epigenetic regulator of α-Klotho. Specifically, increased matrix stiffness increased *Klotho* promoter methylation, downregulated *Klotho* gene expression, and drove young chondrocytes towards an aged phenotype in vitro. Conversely, decreased matrix stiffness abrogated *Klotho* promoter methylation and promoted a youthful phenotype in aged chondrocytes. In these

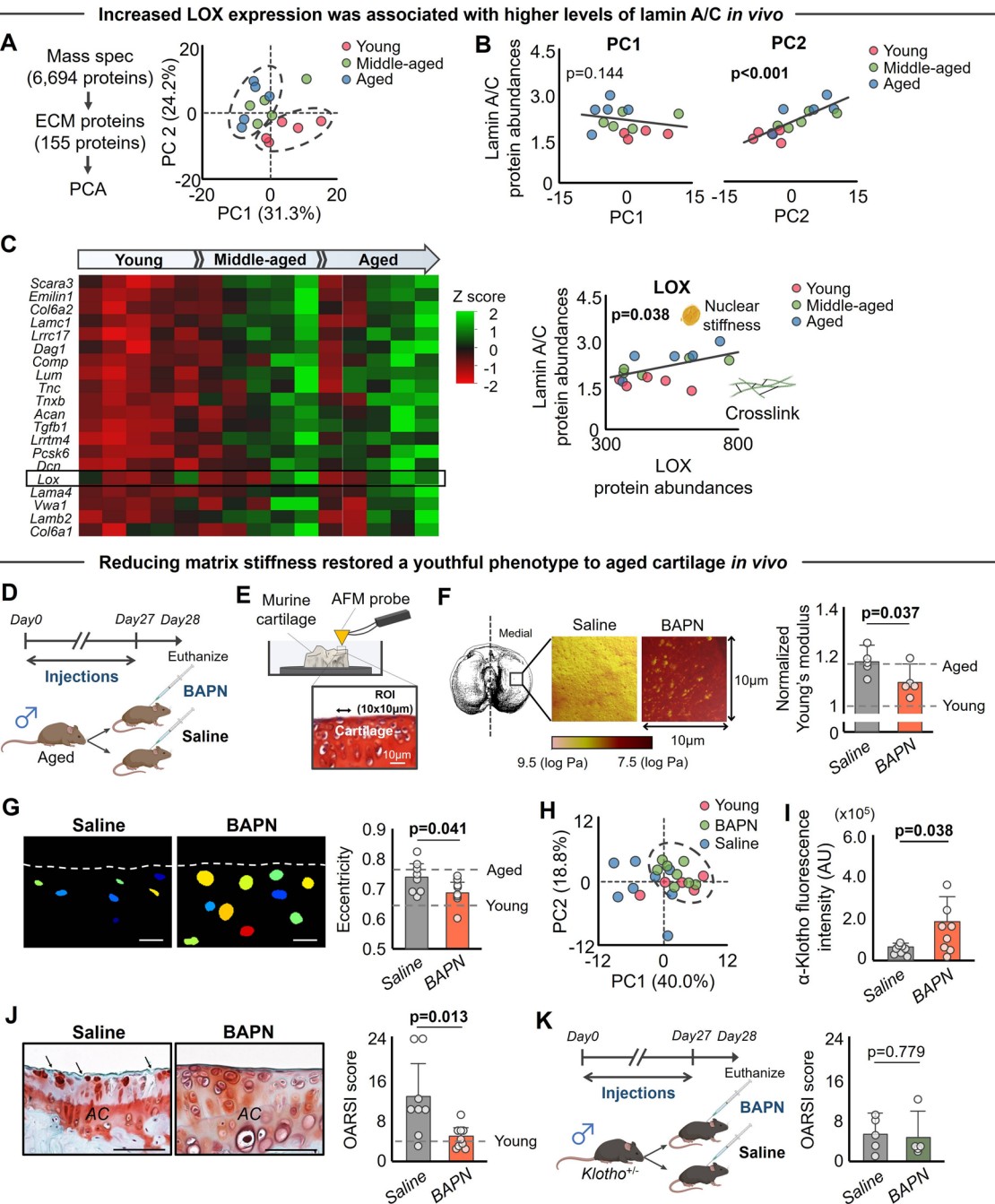

**Fig. 7 | BAPN injection improves α-Klotho expression and cartilage integrity in aged mice. A** Principal component analysis (PCA) showing the separate clusters in ECM proteins from young and aged cartilage (n = 5/group). **B** Principal component 2 (PC2) in ECM protein expression is positively correlated with lamin A/C expression. **C** Heat map of the top 20 ECM proteins contributing to PC2, showing age-related ECM remodeling. Color indicates z score for each variable. Lox, an enzyme that induces collagen cross-linking, was in the top 20 proteins contributing to PC2 and was positively correlated with lamin A/C, supporting a connection between LOX-mediated increased collagen cross-linking and nuclear stiffness. **D** Schematic showing the experimental protocol for 4 weeks of daily BAPN injections in aged mice (n = 8 for saline, n = 9 for BAPN). **E** Schematic showing the atomic force microscopy measurement of murine medial tibial cartilage with 10 × 10 μm region of interest. **F** BAPN treatment significantly reduced Young's modulus of medial tibial cartilage (n = 5/group). Young's modulus data were transformed into Log MPa and normalized by Young's modulus value in naive young group. **G**. Representative CellProfiler-generated nuclear images in murine medial tibial plateaus after 4-week injection of saline or BAPN. Nuclei were pseudocolored. White dashed

lines indicate cartilage surface. Scale bar: 10 μm. BAPN injection in aged mice decreases nuclear eccentricity towards young level (n = 8 for saline, n = 9 for BAPN; 30–40 nuclei per individual mouse). **H** PCA showing the same cluster of young and aged+BAPN injection. **I**. BAPN injection in aged mice improves α-Klotho expression in articular cartilage quantified by immunofluorescence (n = 8 for saline, n = 9 for BAPN; 30–65 cells per individual sample). **J** BAPN injection in aged mice improves cartilage integrity (n = 8 for saline, n = 9 for BAPN). Representative histological sections stained with Safranin O/Fast Green are provided. Black arrows indicate loss of cartilage matrix. OARSI score (0–24 points; higher value indicates more severe cartilage degeneration) assessed by blinded assessor is provided. OARSI score in young cartilage is provided as a reference. **K** Beneficial effect of BAPN injection was blocked in *Klotho*[+/−] mice (n = 5 each for saline and BAPN). Statistical analyses were performed using linear regression analysis (**B**, **C**) or two-tailed Student t test (**F**, **G**, **I**–**K**). Data are presented as means ±95% confidence intervals. Portions of the figures were created with biorender.com. Source data are provided as a Source Data file.

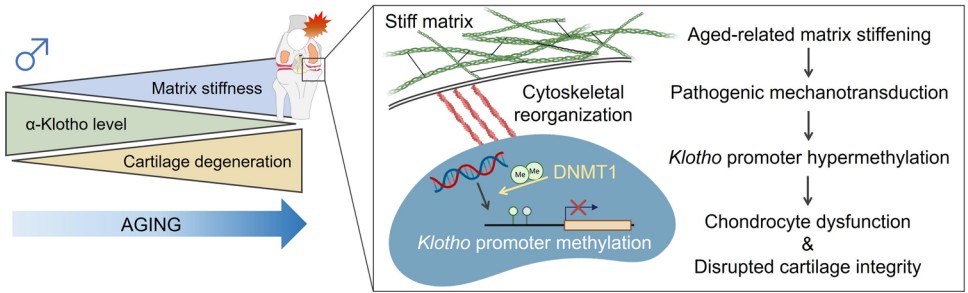

**Fig. 8 | Graphical abstract.** Age-related matrix stiffening in articular cartilage initiates pathogenic mechanotransductive signaling, driving chondrocyte dysfunction as well as disrupted cartilage integrity through *Klotho* promoter hypermethylation. Portions of the figure were created with biorender.com.

findings, epigenetic regulation of α-Klotho was partially controlled by DNMT1 recruitment at the *Klotho* promoter. These results are in line with previous reports showing that pharmacological inhibition of DNMTs improves cartilage integrity in a PTOA model[47,65]. Here, we showed that inhibition of actin polymerization rescued the stiff matrix-induced increases in binding of DNMT1 at the *Klotho* promoter, suggesting that an age-related increase in matrix stiffness increases DNMT1 recruitment at the *Klotho* promoter, which is attributed, at least in part, to cytoskeleton-mediated mechanotransduction. Nevertheless, whether mechanotransductive force directly drives the recruitment of DNMT1/transcription factors to the α-Klotho gene promoter region was not directly evaluated in this work and remains an interesting area for future investigation.

The beneficial effect of decreasing matrix stiffness on chondrocyte health is further supported by our in vivo findings demonstrating that BAPN administration improves α-Klotho expression and cartilage integrity in aged mice. These findings are consistent with previous work demonstrating the chondroprotective effects of BAPN intra-articular injection in a PTOA model[56]. While we confirmed that BAPN administration reduced cartilage stiffness, it is possible that LOX inhibition may have exerted direct biological effects, such as epigenetic modifications, on chondrocytes[66]. To test this possibility, we showed that BAPN administration did not alter α-Klotho levels in aged chondrocytes in vitro, suggesting that the chondroprotective effects of BAPN administration in vivo is not likely to be attributed to direct effects of BAPN on α-Klotho. Rather, our data suggest that the beneficial effect of BAPN on α-Klotho expression in vivo is likely attributed to modification of matrix biophysical properties.

Along a similar vein, it is also worth noting that the improved α-Klotho resulting from reduced ECM stiffness may, in turn, propagate a cascade of matrix remodeling and subsequent improvement in the matrix stiffness[67]. This possibility is supported by growing evidence that α-Klotho exerts anti-fibrotic effects in several tissues[68]. Accordingly, α-Klotho overexpression or supplementation protects against fibrosis in renal and cardiac diseases[68]. These anti-fibrotic effects of Klotho are mediated by the direct inhibitory effects on TGF-β1, Wnt, and FGF2 signaling[68]. This bidirectional interaction between α-Klotho in chondrocytes and their surrounding ECM, a phenomenon known as dynamic reciprocity[69], is an area of interest for future studies.

While this study focused on cartilage as a model, a major conceptual innovation of our work is the demonstration of the novel mechanistic link between age-related increases in ECM stiffness and epigenetic repression of α-Klotho. It is widely recognized that α-Klotho plays a role in the attenuation of an aging phenotype in tissues throughout the body[70], including, for example, skeletal muscle[71,72] and kidney[73], both of which display age-related increases in matrix stiffness[33,74–76]. Yet, the mechanisms provoking these declines over time remain unknown. As such, we anticipate that these studies may have broader implications in the field of aging research, even beyond cartilage.

## Methods

### Steps to ensure methodological rigor

This study was conducted according to the ARRIVE essential 10[77]. Where possible, power analysis from pilot data was done to select the number of animals needed for the study using Power and Sample Size Program (version 3.1.2; Vanderbilt University Medical Center, Nashville, TN)[78]. For example, a sample size calculation estimated that 10 mice were required to achieve statistical power of 0.8 based on the histology score for the aging cohort ($n = 5$ in male young and male aged mice). In addition, a priori power analyses for mass spectrometry estimated that five mice in each group would provide statistical power of 0.9 for detection of a two-fold change, assuming 20% variation[79]. For in vivo experiments utilizing BAPN injection, animals were randomly allocated to treatment or control groups using computer-generated randomization using the random command in Microsoft Excel. The treatments of BAPN were conducted under the same conditions as the control group, and the order of the treatments was performed randomly. We additionally randomly altered the cage location to prevent any bias from the environment. All histological outcome assessments were conducted in a blinded manner.

### Animal and human cartilage samples

Experiments were performed using young (4–6 months), middle-aged (10–14 months), and aged (18–24 months) male and female C57/BL6 mice, as well as young (4–6 months) and middle-aged (10–14 months) male and female *Klotho* heterozygote mice (*Klotho*+/−; B6; 129S5-Kltm1-Lex, UC Davis). C57/BL6 mice were obtained from the NIA Rodent Colony and Jackson Laboratories. Prior to inclusion in experiments, animals with visible health abnormalities were excluded. Mice were housed in cages holding an average of 3–4 mice per cage with a temperature-controlled environment and 12-h light/dark cycles. The animals had access to food and water ad libitum. All animal experiments were approved by the University of Pittsburgh's Institutional Animal Care and Use Committee.

Healthy human articular cartilage tissue in knee joints from young (<40 years old; $27.6 \pm 6.85$ years; $n = 7$ [2 females]) and older (≥65 years old; $71.9 \pm 2.91$ years; $n = 7$ [1 female]) donors were obtained through the National Disease Research Interchange (Philadelphia, PA) with approval from the University of Pittsburgh Committee for Oversight of Research and Clinical Training Involving Decedents (CORID #878). There was no patients' consent or compensation involved.

### BAPN injection of mice

Aged male C57/BL6 mice were randomized into one of two groups, receiving daily subcutaneous injections of either saline as a vehicle control, or BAPN (290 μg/μl in saline) for 4 weeks (aged + saline, $n = 8$, baseline body mass, $33.0 \pm 2.29$ g, aged+BAPN, $n = 10$, baseline body mass, $35.3 \pm 3.03$ g). Middle-aged male *Klotho*+/− mice were also randomized into one of two groups, receiving either saline or BAPN injections for 4 weeks (*Klotho*+/−+saline, $n = 5$, baseline body mass, $36.3 \pm 4.50$ g, *Klotho*+/−+BAPN, $n = 5$, baseline body mass, $40.6 \pm 1.98$ g).

The volume of the injection (40–80 µl) was determined according to the mouse's body mass (*m*) with following protocol: 40 µl (18.45 g ≤ *m* < 23.73 g), 50 µl (23.73 g ≤ *m* < 29.00 g), 60 µl (29.00 g ≤ *m* < 34.27 g), 70 µl (34.27 g ≤ *m* < 39.55 g), 80 µl (39.55 g ≤ *m* < 44.82 g). Mice were weighed every other day beginning on day 0 and ending on day 28 to ensure proper dosing of BAPN. All injections were performed at the same time during the light period (8–9 am). One animal in the aged+BAPN group was excluded from the analysis because of severe inflammation of the knee joint.

### Serum collection
We collected serum from animals under anesthesia by isoflurane in accordance with an established protocol[71]. After collection, the animals were euthanized via cervical dislocation. The collected blood was allowed to clot in a 2 mL tube at room temperature for one hour. Then, the blood was centrifuged for 30 minutes at 13,000 rpm at 4 °C in a microcentrifuge. The serum was collected and aliquoted into 50 µl tubes and stored at −20 °C until used. Any samples displaying hemolysis (as evidenced by pink/red coloration) were not included in the analysis.

### Estrogen ELISA
Circulating estrogen levels were quantified according to the manufacturer's protocol using the 17 beta Estradiol ELISA Kit (cat no. ab108667, Abcam). This kit was validated by Marino FE et al.[80]. Briefly, 25 µl of blood serum samples, prepared standards, or controls were added in duplicate to a 96 well plate. 200 µl of the 17 beta Estradiol-HRP Conjugate were added to each well followed by incubation at 37 °C for two hours. Samples, standards, and controls were aspirated, and wells were washed three times with 300 µl of diluted washing solution (soak time >5 seconds) using an automated plate washer (BioTek 50TS). After washing, 100 µl tetramethylbenzidine (TMB) Substrate Solution was added to each well, and the plate was incubated for 30 minutes in the dark at room temperature. After incubation, 100 µl of Stop Solution was added into all wells in the same order and at the same rate as the substrate solution. Absorbance was measured at 450 nm with Spectramax M3 plate reader (Molecular Devices) within 30 minutes of adding the Stop Solution. Construction of standard curve and subsequent analyses were performed in Microsoft Excel.

### Histological preparation and semi-quantitative histological score for cartilage degeneration
Mouse knee joints were fixed in 4% paraformaldehyde overnight at 4 °C and decalcified in 20% ethylene diamine tetra acetic acid solution for 10 days. Decalcified paraffin sections (5 µm thickness) were prepared from the central region of the mouse knee joints in the frontal plane in accordance with Osteoarthritis Research Society International (OARSI) recommendations[81]. Paraffin sections from human cartilage specimen (6 µm thickness) were also prepared following fixation in 10% buffered formalin phosphate solution overnight at 4 °C.

The paraffin sections were stained with Safranin O/Fast Green/Hematoxylin to evaluate the severity of cartilage lesions. This study focused on medial tibial cartilage, given this is the region most typically affected in humans[7] and its severity is typically equal to or higher than those for the femoral condyle in the majority of aged mice[82]. The OARSI scoring system, consisting of six grades and four stages on a scale from 0 (normal) to 24 (severe cartilage lesion), was used for semi-quantitative evaluation of cartilage lesion severity[83]. The most severe score for each joint was selected as the maximum OARSI score. We did not add the summed scores generated from different sections, as discriminative ability for age-related cartilage degeneration is comparable to the maximum score[82]. A trained examiner (ACB) performed grading in each histological section in a blinded manner with excellent inter-rater reliability (intraclass correlation coefficient [2,1], 95% confidence interval: 0.885, 0.814–0.925) with another trained examiner (HI).

### Computational histological image analysis for cartilage surface roughness
Histologic sections stained with Safranin O/Fast Green/Hematoxylin were imaged at ×10 magnification. We detected the contour line of the cartilage surface on medial tibia and converted it into an X-Y coordinate pixel data using Image J software. We then calculated error, a deviation between the 4th order fitted curve and actual surface shape, using MATLAB version R2020a (MathWorks Inc., Apple Hill Drive, Natick, MA, USA). The computed error thus was an index of the mean discrepancy between the cartilage surface and the expected curve. Since the error value is a function of medial tibia size, the cartilage surface roughness was calculated as the error value divided by length of cartilage in medial tibia (i.e., distance between the edges of medial tibia).

### Chondrocyte isolation
Primary mouse chondrocytes were isolated in accordance with an established protocol[84]. Briefly, cartilage tissue was harvested from the femoral head, femoral condyle, and tibia using small scissors and tweezers while ensuring that minimal synovium, fat, muscle, ligament, and tendons were included in the tissue harvest. After brief wash with PBS two times, cartilage pieces were digested with 0.1% (w/w) type II collagenase (cat no. 4176, Worthington Biochemical corp., NJ) in low-glucose DMEM (cat no. 11965-092, Gibco) with 10% (vol/vol) FBS (cat no. SH30070.03, Hyclone) and 1% (vol/vol) Pen/Strep at 37 °C for overnight at 5% $CO_2$ in a petri dish. Following digestion, the filtrate was passed through a 50-µm strainer and cells were cultured in growth media containing DMEM supplemented with 10% FBS and 1% Pen/Strep until the cells reached confluence. First-passage cells were used for all experiments. The isolated cells were characterized by type II collagen immunofluorescence, with over 95% cells positive for type II collagen expression.

### Preparation of fibronectin-coated pAAm substrates
We prepared pAAm gels with different stiffnesses (5 kPa, 21 kPa, and 100 kPa) in accordance with a previous study[35]. The pAAm gels were made on glass coverslips pre-treated with 0.1 N sodium hydroxide (cat. SS255-1, Fisher Scientific, IL), 0.5% 3-aminopropyltrimethoxysilane (cat no. AC313251000, Acros Organics, Belgium) and 0.5% glutaraldehyde (cat no. BP25481, Fisher Scientific, IL) to improve gel adhesion. Detailed gel composition is provided in Table S2. To facilitate cell adhesion, the surfaces of prepared hydrogels were further conjugated with fibronectin (100 µg/mL, from bovine plasma, Sigma) using sulfo-SANPAH (cat no. NC1314883, Proteochem Inc., UT) as a crosslinker. Prior to seeding cells, gels were UV-sterilized in a cell culture hood for 30 minutes. Gels were kept hydrated in HEPES or PBS during all preparation steps.

### Evaluation of the impact of matrix stiffness on cell fate
Isolated primary chondrocytes from young and aged mice were plated (5000 cells per cm²) on 5 kPa, 21 kPa, and 100 kPa pAAm gels and cultured in low-glucose DMEM supplemented with 10% FBS and 1% Pen/Strep with or without 50 µg/mL ascorbic acid-2-phosphate (cat no. A8960, Sigma) at 37 °C at 5% $CO_2$. We selected the concentration of ascorbic acid based on previous literature demonstrating enhanced chondrogenic gene expression in murine chondrocytes with the ascorbic acid supplementation[85]. On day 5, the culture medium was removed, and seeded pAAm gels were fixed in 2% PFA for 10 minutes. After a triple wash by PBS, cells were kept in PBS at 4 °C until used.

To further evaluate the impact of matrix stiffness on the ability of chondrocytes to secrete matrix proteins, soluble type II collagen and α-Klotho were quantified from conditioned medium collected at day 5 of culture on pAAm gels after concentration (Amicon® Ultra-4 Centrifugal Filter Devices). ELISA was used to quantify type II collagen (cat no. SEA572Mu, Cloud Cone Corp, TX, USA). Multiple freeze-thaw cycles were avoided before the experiment.

**Treatments with Latrunculin A, Y-27632, PF-562271, and BAPN**

Isolated primary chondrocytes from aged mice were plated (5000 cells per cm²) on 100 kPa pAAm gels and cultured in low-glucose DMEM supplemented with 10% FBS and 1% Pen/Strep at 37 °C at 5% $CO_2$. On day 3, chondrocytes were treated with latrunculin A (0.01, 0.1, and 1.0 µM) for 1 hour, Y-27632 (50 µM) for 48 hours, PF-562271 (1 µM) for 48 hours, or BAPN (200 µM) for 48 hours. The chondrocytes were washed with media two times and cultured for additional two days. The culture medium was removed, and constructs were fixed with 2% PFA for 10 minutes. After a triple wash by PBS, cells were kept in PBS at 4 °C until analysis.

In a separate analysis, the cytotoxic effect of latrunculin A was determined using the live/dead reduced biohazard viability/cytotoxicity kit (cat no. L7013, Molecular probes, Inc., OR) in accordance with manufacturer's protocol. Briefly, isolated primary chondrocytes from aged mice were plated (1000 cells per cm²) on chamber slides after which they were cultured and treated by latrunculin A (0.01, 0.1, and 1.0 µM) with the same protocol as above. After washing with HEPES-buffered saline solution, chondrocytes were incubated in live/dead viability/cytotoxicity solution for 15 min after which they were fixed with 4% glutaraldehyde for 1 hour. A positive control for dead cells was generated by fixing the chondrocytes by 4% PFA for 15 minutes prior to incubation with the live/dead viability/cytotoxicity solution. Cells were imaged using a Zeiss Observer Z1 semi-confocal microscope with ZEN 2.3 software (Zeiss, Jena, Germany). Positive control slide was used to set the threshold for the signal intensity and the exposure time for individual channels.

**In vitro inhibition of *Klotho* and *Dnmt1***

Isolated primary chondrocytes from young mice were plated (5000 cells per cm²) on 5 kPa pAAM gel and cultured in low-glucose DMEM supplemented with 10% FBS and 1% Pen/Strep at 37 °C at 5% $CO_2$. On day 3, the chondrocytes were treated with 25nmol of siRNA to *Klotho*, or a non-targeting scramble control (GE healthcare Dharmacon) in antibiotic-free chondrocyte growth media for 6 hours. The chondrocytes were then washed twice with media and cultured for an additional 48 hours. The culture medium was removed, and cells were fixed with 2% PFA for 10 minutes. After a triple wash by PBS, cells were kept in PBS at 4 °C until further analysis. For the genetic inhibition of *Dnmt1*, the same protocol described above (25 nmol of siRNA to *Dnmt1*) was used.

To pharmacologically inhibit DNMT activity, the DNMT inhibitor, 5-Aza-2'-deoxycytidine (10 µM; 5 Aza, MCE-MedChemExpress, NJ, US) was used in a separate experimental cohort. After 3 days chondrocytes culture on 5 kPa pAAM gel in low-glucose DMEM supplemented with 10% FBS and 1% Pen/Strep at 37 °C at 5% $CO_2$ (5000 cells per cm²), 5-Aza-2'-deoxycytidine was supplemented in the growth medium and cultured for additional 48 hours. The culture medium was removed, and cells were fixed with 2% PFA for 10 minutes. After a triple wash by PBS, cells were kept in PBS at 4 °C until further analysis. To inhibit DNMT1 activity and *Klotho* expression simultaneously, chondrocytes were treated by 5-Aza-2'-deoxycytidine (10 µM) and siRNA *Klotho* (25nmol) together on day 3, and the protocol described above to analyze 5 Aza was repeated.

**Fabrication of gelatin-based hydrogels and chondrocytes encapsulation**

Gelatin-based hydrogels (GelMA) were fabricated by photo-cross-linking[86]. Briefly, GelMA (20% stock solution, Cellink, Boston, MA) was diluted to 8% by adding reconstitution buffer (Agent P, Cellink) and equal proportion of photoinitiator (LAP, Cellink) in accordance with manufacturer's protocol. Subsequently, male mouse chondrocytes were suspended in the GelMA solution at a concentration of $2.0 \times 10^7$ cells/ml. The suspension was pipetted into cylindrical silicon molds 2 mm in height and 5 mm in diameter. The GelMA was then crosslinked with 395 nm UV light for 2 minutes. Constructs were cultured for 5 days in growth media containing DMEM supplemented with 10% FBS and 1% Pen/Strep.

**Evaluation of mechanical properties of GelMA using AFM**

GelMA (8% and 20% w/v) was added into glass bottom dishes (D110300, Matsunami, Osaka, Japan) and maintained at 4 °C for 10 minutes. Subsequently, distilled water was added to the gel. We use these procedures to keep surface structure stable and ensure reliable AFM measurement. GelMA was then crosslinked with 395 nm UV light for 2 minutes and maintained overnight at room temperature before further measurement. The elasticity of GelMA hydrogel were evaluated using atomic force microscopy (NanoWizard® 3 NanoOptics, JPK Instruments, Berlin, Germany). Briefly, a cantilever with 5 µm diameter spherical beads attached to the tip (CP-CONT-BSG-A, NanoAndMor-eGmbH, Watsonville, CA) and Hertzian contact model were used to obtain force and displacement values, and Young's modulus was calculated using JPK DP Data Processing Software (JPK Instruments). At least three specimens were used for each testing condition.

**Evaluation of the effect of BAPN on mechanical properties of murine cartilage using AFM**

Freshly dissected medial tibia cartilage in mice knee joints were maintained in PBS with protease inhibitors at 4 °C for <24 h prior to mechanical testing[87]. Excised knee joints were mounted onto steel sample disks using Loctite 454 instant adhesive gel. Samples were immersed in PBS buffer and fluid mode AFM measurements were collected. Peak Force Tapping Quantitative Nanomechanical maps were generated using a Bruker Dimension Icon Atomic Force Microscope and NanoScope 6 Controller. MikroMasch HQ:NSC18 AFM probes with nominal cantilever spring constants of 2.8 N/m for the aged, aged+saline, and aged+BAPN treated group. Bruker MLCT-BIO AFM probes with nominal 0.6 N/m spring constants were used for the young group. Cantilever spring constants were determined using the thermal tunning method. Cantilever deflection sensitivities, QNM synchronization distances, and Peak Force Tapping amplitude sensitivities were determined by analysis of force curves collected on a sapphire surface submerged in PBS buffer. Probe radii and tip shapes were estimated by tip qualification analyses of titanium roughness standard micrographs, Bruker RS-12M. During sample QNM measurements, the force setpoint was adjusted to maintain between 10–20 nm of nominal sample indentation. In each cartilage sample, a $10 \times 10$ µm field (512 scan lines by 512 points) was scanned. To avoid high-frequency noise in the Young's modulus data, a median filter was applied. All data analyses were performed using NanoScope Analysis 2.0 (Bruker Co., MA, USA).

**Immunofluorescence and imaging**

Immunofluorescence analysis for tissue sections, cell-seeded pAAm gels, and cell-encapsuled GelMA were performed to determine the signal intensity of type II collagen, α-Klotho, aggrecan, lamin A/C, lamin B2, DNMT1, and FAK in accordance with established protocols[71]. Briefly, after a triple wash by PBS, cells were permeabilized with 0.1% triton-X (Fluka 93420) for 15 minutes, followed by blocking for 1 hour in 0.1% triton-X and 3% bovine serum albumin (BSA, Sigma A7906) in PBS. For the decalcified paraffin-embedded tissue sections, antigen-retrieval was performed by incubating sections in sodium citrate buffer for 2 hours at 60 °C prior to blocking. After blocking, the tissue or cells were incubated overnight at 4 °C with primary antibodies in antibody solution (0.1% Triton-X + 3% BSA + 5% Goat Serum), at the dilutions provided in Table S3. One negative control slide per staining set was generated by omitting the primary antibody in the antibody solution.

After a triple wash by PBS, the samples were incubated with host-specific secondary antibodies conjugated with Alexa Fluor 488 (Fisher

 

Scientific) in antibody solution for one hour at room temperature at a 1:500 dilution. Following a triple wash with PBS, the samples were stained with DAPI for 2 minutes and then washed with PBS again. Samples were mounted with coverslips using Gelvatol mounting medium (Source: Center of Biologic Imaging, University of Pittsburgh). The α-Klotho antibody (R&D systems, MAB1819, Lot# KGN0315101) was validated for skeletal muscle histological sections in a previous study[71]. We also validated the same α-Klotho antibody using the decalcified paraffin section from a wild-type and a KL$^{-/-}$ mouse knee joint. We observed minimal staining in the Kl$^{-/-}$ compared to the wild-type counterparts. Slides were imaged using a Zeiss Observer Z1 semiconfocal microscope with ZEN 2.3 software (Zeiss, Jena, Germany). All images were collected at ×20 or ×63 magnification. Negative control slides were used to the threshold for the signal intensity and to set the exposure time for individual channels. All images for quantitative analysis in a given experiment were taken under the same imaging conditions with the same threshold for the signal intensity and the same exposure time. Fluorescence intensity was quantified using Image J. More specifically, integrated density for each channel was divided by number of cells in each image (i.e., fluorescence intensity per cell). In each immunofluorescence-based quantification, we used 5 or 10 randomly selected images for ×20 and ×63 images, respectively, per individual sample. Data for the random images were then averaged in each independent sample and used for statistical analysis (e.g., if we analyzed five independent samples in total, data was shown as $n = 5$ wells/group).

### Real-time-PCR
One μg of total RNA from chondrocytes cultured on pAAm gels was reverse transcribed using iScript™ Advanced cDNA Synthesis Kit (BIO-RAD, Hercules, CA). The mRNA levels of the *Klotho* gene were quantified by SYBR Green-based real-time PCR (qPCR) using SsoAdvanced™ Universal SYBR® Green Supermix (BIO-RAD, Hercules, CA). *Klotho* gene expression levels were normalized to the expression level of *Rpl44*, and the fold change of *Klotho* relative to universal mouse reference RNA was calculated using 2−ΔΔCt method.

### Methylation-specific PCR
200 ng of genomic DNA was subjected to bisulfite conversion with an EZ DNA Methylation Kit (Zymo Research, Irvine, CA) followed by methylation-specific PCR (MSPCR) using primers specific for methylated DNA of *Klotho* promoter. Primer sequences were previously reported for MSPCR of muscle DNA[88]. Fully methylated control DNA (Zymo Research, Irvine, CA) was used as a reference to calculate the percentage methylation of DNA samples.

### Global DNA methylation assay
Levels of 5 methylcytosine (5mC) in DNA were measured using the 5mC DNA ELISA Kit (Zymo Research, Irvine, CA). A total of 200 ng of DNA was used and the percent of 5mC in the samples was quantified, normalized to total DNA, and compared to the standard curve provided in the kit.

### ChIP analysis
We performed ChIP analysis as described previously[71,89]. Briefly, cells were fixed with 1% formaldehyde followed by quenching with glycine solution. Crosslinked cells were pelleted and subject for chromatin isolation and immunoprecipitation using ChIP-IT High Sensitivity kit (Active Motif, Carlsbad, CA), following the manufacturer's instructions. One to two μg of ChIP-validated antibodies against RNA Polymerase II (A2032, Epigentek, Farmingdale, NY), DNMT1 (A1001, Epigentek, Farmingdale, NY), H3K9M2 (ab1220, Abcam, Cambridge, MA), H3K4M2 (39913, Active Motif, Carlsbad, CA), c-MYC (sc-40, Santa Cruz Biotechnology, Dallas, TX) or nonspecific negative control IgG (Active Motif, Carlsbad, CA) were used for immunoprecipitation of

each sample. Non-immunoprecipitated chromatin was used as input. Following reverse cross-linking and elution of chromatin, DNA from each sample was purified with QIAquick PCR purification kit (Qiagen, Germantown, MD). Purified DNA sample concentrations were quantified by Qubit DNA HS assay kit (Thermo Fisher Scientific, Waltham, MA). Ten ng of purified DNA was used for PCR of *Dnmt1* or *Klotho* promoters. The percent enrichment of immunoprecipitated DNA relative to the input was calculated.

### Quantification of cellular and nuclear morphology
DAPI and F-actin images were obtained at ×63 and ×20 magnification using a Zeiss Observer Z1 semi-confocal microscope with ZEN 2.3 software (Zeiss, Jena, Germany), respectively. Image processing and morpheme feature extraction were performed using CellProfiler software (v4.0, The Broad Institute)[30]. Fifty-three shape features of cells and nuclei were determined using the "identify primary objects" followed by the "measure object size shape" and "export to spreadsheet" modules.

### LC/MS-MS mass spectrometry-based proteomics
Knee cartilage from male and female young, middle-aged, and aged C57/BL6 mice ($n = 5$/age/sex) were microdissected as detailed previously[90]. Samples were lyophilized overnight and stored at −80 °C until shipment to the Proteome Exploratory Laboratory at Cal Tech.

Cartilage samples from each knee were lysed in 8 M urea/100 mM TEAB by grinding for 1 min with size tissue grinder pestles (cat. 12141363, Fisher Scientific), tip sonication with a Fisher Scientific 550 Sonic Dismembrator on ice at 20% power using cycles of 20 sec on/ 20 sec off for 4 minutes total, followed by another grinding step for 1 min. Samples were then clarified by centrifugation at 16,000 × *g* for 5 minutes at room temperature. Each lysate was then reduced with 500 mM TCEP for 20 min at 37 °C and alkylated with 500 mM 2-Chloroacetamide for 15 min at 37 °C in the dark. Samples were then digested with a 1:200 ratio of LysC to lysate for 4 hr at 37 °C, followed by dilution with 100 mM TEAB, addition of 100 mM CaCl2, and digestion overnight with 1:30 Trypsin at 37 °C. Digestions were stopped by acidifying with 20% TFA, desalted on C18 spin columns (cat. 89870, Fisher Scientific) according to manufacturer instructions, and lyophilized to dryness. Peptides were then resuspended in 0.1% formic acid and peptide amounts measured with the Pierce Quantitative Colorimetric Peptide Assay.

15 μg peptides from each sample were lyophilized, resuspended in 100 mM TEAB, labeled with TMTpro reagents dissolved in anhydrous acetonitrile for 1 hr at room temperature, and quenched with 5% hydroxylamine for 15 min at room temperature. All 15 male samples were then combined into one sample and all 15 female samples were combined into a second sample 100 μg of each sample was then fractionated with the Pierce High pH Reversed-Phase Peptide Fractionation Kit (Thermo #84868) according to manufacturer instructions and the resulting 8 fractions were lyophilized. Each fraction was resuspended in 20 μl 0.2% formic acid and peptide quantitation was performed with the Pierce Quantitative Colorimetric Peptide Assay. Fractions 7 and 8 from both samples had very low peptide amounts and were thus combined with that sample's fraction #6 for a total of 6 fractions per sample.

Liquid chromatography-mass spectrometry (LC-MS) analysis of peptide fractions was carried out on an EASY-nLC 1000 coupled to an Orbitrap Eclipse Tribrid mass spectrometer (Thermo Fisher Scientific). Each fraction was loaded onto an Aurora 25 cm × 75 μm ID, 1.6 μm C18 reversed-phase column (Ion Opticks, Parkville, Victoria, Australia) and separated over 136 min at a flow rate of 350 nL/min with the following gradient: 2−6% Solvent B (7.5 min), 6−25% B (82.5 min), 25−40% B (30 min), 40−98% B (1 min), and 98% B (15 min). MS1 spectra were acquired in the Orbitrap at 120 K resolution with a scan range from 350−1800 m/z, an AGC target of 1e6, and a maximum injection time of

50 ms in Profile mode. MS2 scans were then acquired in the Orbitrap at 50 K resolution in Centroid mode with the first mass fixed at 110. Cycle time was set at 3 seconds.

Analysis of LC-MS proteomic data was performed in Proteome Discoverer 2.5 (Thermo Scientific) utilizing the Sequest HT search algorithm with the mouse proteome (UniProt UP000000589; 55,485 proteins covering 21,989 genes with a BUSCO assessment of 99.8% genetic coverage). Search parameters were as follows: fully tryptic protease rules with 2 allowed missed cleavages, precursor mass tolerance set to 20 ppm, fragment mass tolerance set to 0.05 Da with only b and y ions accounted for. Percolator was used as the validation method, based on q-value, with a maximum FDR set to 0.05. GO Biological Process terms were generated via Proteome Discoverer Sequest HT algorithm and were used for sorting proteins associated with PI3K/Akt signaling.

Quantitative analysis is based on TMT MS2 reporter ions generated from HCD fragmentation, with an average reporter S/N threshold of 10, used a co-isolation threshold of 50 with SPS mass matches set at 65%. Normalization was performed at the peptide level, and protein ratios were calculated from the grouped ion abundances, with protein FDR set to a maximum of 0.05.

The mass spectrometry proteomics data have been deposited to the CalTech Proteomics repository[91] (https://doi.org/10.22002/ee2yc-fg857).

### Bioinformatic analysis
To assess enriched pathways associated with aging in cartilage, KEGG enrichment analyses were performed using ROnToTools R Code[92] with mass spectrometry data stratified by sex. The code was used as written, with the exception of changing 'hsa' to 'mmu' such that mouse pathways were referenced. Our dataset matrices were generated in Excel following the format provided in the sample dataset[92]. To consider transcript level changes relative to significantly changing pathways identified by mass spectrometry, we accessed archived RNA-seq data assessing global transcriptomic changes between young (3 months old) and aged (24 months old) C57BL6 mice[16]. Differentially expressed genes/proteins across the protein and transcripts data were integrated and were used for KEGG enrichment analyses by Enrichr software (https://amp.pharm.mssm.edu/Enrichr/). In this analysis, the protein name from mass spectroscopy data was translated into its gene symbol as in the RNA-seq data.

To identify possible regulators of aging process in cartilage, upstream regulator analysis was performed using Ingenuity Pathways Analysis software (QIAGEN, Hilden, Germany)[17] with mass spectrometry data. As an input variable, we used fold change value of protein abundance and p-values of young versus aged male mice. This study included following molecule types as upstream regulators: cytokines, growth factor, kinases, and group. Activation z-scores were calculated, where a higher score indicates activation of the identified upstream regulators.

To determine whether treatment of siRNA to Klotho changes PI3K/Akt signaling, single sample gene set enrichment analysis (ssGSEA) was performed by Gene Set Enrichment Analysis software (http://software.broadinstitute.org/gsea/index.jsp)[93]. As an input variable, we used gene scores defined by value of log2 fold change of gene expression profiles after treatment of siRNA compared to no treatment control group provided by previous study (GSE80285)[25]. We used the PI3K/Akt signaling involved in KEGG pathways (KEGG_2019_Mouse) as a gene set downloaded from Enrichr.

### Unsupervised machine learning
PCA was performed for data reduction to identify the principal components that represent differences in the cellular and nuclear morphology. To determine variables of cellular and nuclear shape contributing to PCs, the loading matrix, a correlation between the original variables and principal components, was extracted. In addition, PCA was used to visualize the separation of (1) nuclear morphology in young, middle-aged, and aged cartilage, (2) young and aged ECM protein abundance assessed by mass spectrometry-based proteomics, (3) the OARSI score in BAPN-injected and saline-injected knees, and (4) nuclear morphology in chondrocytes cultured on stiff pAAm gel with and without siRNA to *Dnmt1*.

### Statistical analysis
All statistical analyses were performed using JMP Pro 14 software (SAS Institute, Cary, NC) or SPSS Statistics for Windows, Version 28.0 (IBM Corp., NY, USA). Except where indicated, data are displayed as means, with uncertainty expressed as 95% confidence intervals (mean ± 95% CI). For unpaired experiments, two-tailed Student *t* test, linear regression analysis, or two-way ANOVA was performed. For paired experiments, two-tailed paired *t*-test or linear mixed effect models were utilized. We checked the features of the regression model by comparing the residuals vs. fitted values (i.e., the residuals had to be normally distributed around zero) and independence between observations. No correction was applied for multiple comparison because outcomes were determined a priori and were highly correlated. No statistical analyses included confounders (e.g., body mass in each animal) due to the small sample size. We conducted a complete-case analysis in the case of missing data. In all experiments, *p* values <0.05 were considered statistically significant. Throughout this text, "*n*" represents the number of independent observations of knees or cells from different animals. Specific data representation details and statistical procedures are also indicated in the figure legends.

### Data availability
The raw data that support the experimental findings are included as Source Data. The mass spectrometry proteomics data have been deposited to the CalTech Proteomics repository (https://doi.org/10.22002/ee2yc-fg857). Source data are provided with this paper.

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

## Acknowledgements

This study was supported in part by (1) a Grant-in-Aid from the Japan Society for the Promotion of Science for Overseas Research Fellowships for H.I., (2) research grants from the Uehara Memorial Foundation for H.I., (3) NIA R01AG052978, NIA R01AG061005, and P2CHD086843 for F.A., (4) the National Institute of General Medical Sciences of the National Institutes of Health under Award Number T32GM008208 for G.G, (5) JSPS KAKENHI Grant Number JP21H03302 for A.I., (6) Kyoto University Nano Technology Hub in "Nanotechnology Platform Project", sponsored by the Ministry of Education, Culture, Sports, Science and Technology (MEXT), Japan, and (7) the John and Posy Krehbiel Professorship in Orthopedics for C.E. The funders had no role in study design, data collection and analysis, decision to publish, or preparation of the manuscript.

## Author contributions

All authors made substantial contributions in the following areas: (1) conception and design of the study, acquisition of data, analysis and interpretation of data, drafting of the article; (2) final approval of the

article version to be submitted; and (3) agreement to be personally accountable for the author's own contributions and to ensure that questions related to the accuracy are appropriately investigated, resolved, and the resolution documented in the literature. The specific contributions of the authors are as follows: H.I., G.G., K.W., and F.A. provided the concept, idea, and experimental design for the studies. H.I., G.G., and F.A. wrote the manuscript. H.I., G.G., K.W., A.C.B., Y.H., H.L., W.Y.T., D.L., C.T., A.I., J.J., C.E., and F.A. provided data collection, analyses, interpretation and review of the manuscript. H.I., G.G., and F.A. obtained funding for the studies.

## Competing interests

All authors declare no competing interests.

## Additional information

**Peer review information** *Nature Communications* thanks Danny Chan, Makoto Kuro-o and the other, anonymous, reviewer(s) for their

contribution to the peer review of this work. Peer reviewer reports are available.

