## [Peer Review File · Nature Communications]

Age-related matrix stiffening epigenetically regulates α -Klotho expression and compromises chondrocyte integrityREVIEWER COMMENTS

Reviewer #1 (Remarks to the Author):

In this study, Iijima et al investigated whether α -Klotho plays a role in cartilage aging. The authors reported that an age-related loss of Klotho drives progressive cartilage degeneration. The authors believe that loss of Klotho was a consequence of age-related matrix stiffening, whereby a stiff matrix epigenetically inhibited Klotho expression in chondrocytes through lamin A/C-mediated mechanotransductive signaling. Exposing aged chondrocytes to a soft matrix restored a more youthful phenotype in vitro and enhanced cartilage integrity in vivo. They concluded that age-related alterations in ECM biophysical properties initiate pathogenic mechanotransductive cascades that promote Klotho promoter methylation and compromise cartilage integrity.

Overall, this is a straightforward study which tests a hypothesis that age-related alterations in mechanotransductive signaling may epigenetically suppress α -Klotho expression. There are several concerns with this report.

Major

1. The novelty of this study is compromised partly because the roles of Klotho in the regulation of PI3k/AKT has been well established (Ref # 17, 40).
2. It is not clearly justified why Klotho was singled out for study because aging affects numerous genes and proteins. Upregulation of DNMT1 increases methylation of numerous genes which may be involved in the maintenance of cartilage integrity.
3. This study could benefit from an evidence-based RNA-seq analysis which will provide genes altered by aging.
4. How inhibition of LOX increases Klotho levels is not addressed. It is important to show whether LOX directly regulates DNMT1 expression and Klotho gene promoter methylation.
5. It is not convincing that alterations of mechanotransductive signaling directly increases DNMT1 expression and Klotho promoter methylation. There is a lack of critical mechanistic investigation into how mechanotransductive signaling regulates DNMT1 and Klotho promoter methylation.
6. How the mechanotransductive force of lamin A/C is measured is not clearly stated.
7. The authors should investigate whether chromosome remodeling alters DNMT1 expression and Klotho promoter methylation. It was reported that histone methylation regulates Klotho gene transcription and expression.
8. There is a concern on the systemic delivery of Klotho protein which complicates interpretation of the direct effect of the exogenous Klotho in chondrocytes in cartilage. The changes in cartilage may be secondary to the systemic effects of Klotho.
9. It is not convincingly addressed whether downregulation of Klotho is the cause or consequence of

cartilage stiffness and remodeling. It is known that a decrease in Klotho leads to aging phenotype (tissue remodeling, calcification and stiffening). Thus, aging-related downregulation of Klotho may be the cause of cartilage stiffness.

10. A concern is that the role of Klotho in the blood in regulation of cartilage homeostasis is ignored. Klotho is primarily expressed in the kidney and released to the circulation.

11. There is no convincing data showing that exogenous Klotho is uptaken by chondrocytes in cartilage.

Reviewer #2 (Remarks to the Author):

The authors present support for a mechanism of age-related osteoarthritis (OA) driven by matrix stiffening, methylation of the Klotho promoter, and over-activity of PI3K/Akt. The series of experiments are logical and rigorously performed, and the results are very clearly presented. The baseline experiments of OA and mass spec on male and female mice are important benchmarks for the other findings and are well done. While some of the individual findings are only modest extensions of published work, the “total package” provides a compelling new paradigm that will be important for the field in terms of understanding the connection between matrix changes and risk for OA with aging. The comments below are intended to improve the impressive body of work already present in the manuscript.

1) Are the methylation changes that occur in response to substrate stiffness specific to Klotho promoter? The authors show that DNMT1 levels do change as well, which makes it difficult to tell whether the effects on Klotho are just due to a general mechanism of increased methylation under stiff substrate conditions. Perhaps a series of control sites could be selected for methylation-specific qPCR to compare to Klotho? Or better yet a genome-wide analysis to normalize the changes occurring at Klotho to rest of genome?

2) Figure 2G/H is impressive, but a bit surprising given the poor regenerative capacity of cartilage in older mice (and humans). What is the explanation from the authors that just 2 weeks of systemic Klotho would actually improve cartilage quality (as opposed to preventing further degradation). More support is needed for such a claim, but it is recognized that these experiments are challenging to perform. The legend indicates n=7/8. Were these males? What age? Were timecourse experiments performed or was the 2-week implantation the only experiment performed? There can be a transient increase in GAG staining during the early phases of OA. It would be important to show that these acute improvements persisted to some extent. Would the Klotho pumps “rescue” the excessive degradation in the Klotho +/- middle-aged mice? That would presumably provide more support that Klotho is sufficient to improve joint health after a moderate level of OA, especially given that in this case the phenotype is in part caused by insufficient Klotho.

3) One relevant reference seemed to be missing: Martinez-Rodono, Protein Cell, 2020:
<https://pubmed.ncbi.nlm.nih.gov/31950348/>

4) In general, the methods were quite detailed, but one exception that should be clarified is the number of cells used to quantify immunofluorescence. This is an important readout for both figures 3 (tissue sections) and 4 (cells) because I did not see any bulk analysis using western blot that would encompass many more cells. Could these type of bulk data (for Klotho, Col2) be pulled and highlighted from the mass spec experiments?

One example – the Fig. 4 legend gives n=5. I assume that is 5 different chondrocyte isolations (from individual or pooled mice). How many cells were quantified? Only 3-4 cells are shown in representative images, but of course having many cells over numerous random fields across the culture would make sure this is a consistent effect.

5) The following sentence should probably be re-worded, as the osmotic pump for gain of function is also a systemic approach and not specific to the joints. “Therefore, to more directly evaluate whether α -Klotho attenuates an aged cartilage phenotype, we employed a gain-of-function paradigm. While serial intra-articular injections or some form of stable Klotho overexpression in the joint would be interesting, that may be beyond the scope of this study.

Reviewer #3 (Remarks to the Author):

General comments

This paper identified Klotho as a chondroprotective factor and may play a critical role in pathogenesis of age-related chondrocyte degeneration. This finding may promote better understanding of pathophysiology of osteoarthritis to develop a novel therapeutic strategy.

Specific comments

Major points:

The authors' main findings are:

1. Cartilage degeneration with age was associated with disruption of PI3K/Akt signaling and with decrease in Klotho expression in chondrocytes in mice.
2. A decrease in Klotho expression was associated with accelerated cartilage degeneration in mice.
3. Administration of soluble Klotho ameliorated cartilage degeneration in mice.
4. Treatment of cultured chondrocytes with soluble Klotho attenuated PI3K/Akt signaling.
5. A decrease in endogenous Klotho expression in cultured chondrocytes enhanced PI3K/Akt signaling.
6. Chondrocytes cultured on stiff substrates had lower Klotho expression levels than those cultured on soft substrates.

These findings support the authors' conclusion that increase in matrix stiffness with age may accelerate cartilage degeneration through suppressing Klotho expression.

However, the following points are not clear:

1. It is not clear whether soluble Klotho ameliorated cartilage degeneration through inhibiting PI3K/Akt signaling. The authors did not provide data in support of the hypothesis that the therapeutic effect of soluble Klotho on cartilage degeneration might depend on the ability of soluble Klotho to inhibit PI3K/Akt signaling. The findings #4 and #5 were not proof of the hypothesis, because they were consistent with previous reports showing that soluble Klotho inhibited PI3K/Akt signaling (Kurosu H, et al. *Science* 309:1829-33, 2005). Besides, soluble Klotho was reported to regulate activity of several ion transporters/channels and growth factors (for review; Kuro-o M. *Nat Rev Nephrol* 15:27-44, 2019). The authors should discuss these points (i.e. the meaning of the horizontal arrow in Figure 5I).

2. It is not clear whether Klotho may act on synoviocytes and macrophages in the knee joint. The PI3K/Akt signaling affects not only chondrocyte function but also function of other cells in the knee joint, including synoviocytes and macrophages (Sun K, et al. *Osteoarthritis Cartilage* 28:400-9, 2020). All these cells participate in the development of osteoarthritis. Therefore, it is possible that soluble Klotho may act on all these cells in an autocrine, paracrine, and/or endocrine manner. The authors should discuss why they focused on chondrocytes.

3. It is not clear which soluble Klotho may be of physiological importance for maintenance of chondrocytes, autocrine/paracrine Klotho from chondrocytes or endocrine Klotho secreted from renal tubular cells. The major source of circulating soluble Klotho is the kidney (Hu MC, et al. *J Am Soc Nephrol* 27:79-90, 2016). The reviewer was unable to find many papers reporting that patients with chronic kidney disease, whose serum Klotho levels should be low, were at high risk for osteoarthritis. However, the authors showed that systemic administration of soluble Klotho prevented chondrocyte degeneration. The authors should discuss a mode of action of soluble Klotho in chondrocytes.

4. How were the serum levels of soluble Klotho, FGF23, phosphate, calcium, and active vitamin D changed in mice treated with soluble Klotho protein? Both membrane and soluble Klotho form binary complexes with receptors for fibroblast growth factor (FGF) to function as the receptor for FGF23 (Chen G, et al. *Nature* 553:461-6, 2018). FGF23 is a bone-derived endocrine factor that functions as a phosphaturic hormone and a counter-regulatory hormone against active vitamin D and parathyroid hormone. Administration of soluble Klotho may affect these factors and thus pathophysiology of OA.

Minor points

1. Figure 2B, 2D, 3H, S4b, S4C, S4D, S4F, 5B, 5C, 5G: What does the y-axis represent? Protein levels determined by the mass-spec or mRNA levels determined by qPCR?

2. Figure S2, 5C: What does the x-axis represent? Protein levels determined by the mass-spec or mRNA levels determined by qPCR?

3. Page 11, Line 248: via via -> via

Reviewer #4 (Remarks to the Author):

This is an interesting manuscript providing mechanistic insight into the relationship between changes in mechanical property with age and knee osteoarthritis (KOA). The authors have used a combination of in vitro/in vivo biochemical /biophysical assessments to link changes in the stiffness

of the ECM in the knee cartilage that could regulate the epigenetic changes in the promoter of apha-Klotho, a gene well-known to be involved in ageing and longevity. Their conclusion is based on computational, statistical, and functional assessments in consolidating their findings. Much of the pathways and genes identified/selected for assessments are known to be related to cartilage biology and degeneration, and the novelty is in linking these together providing a mechanistic insight. Data provided are clear, but some key associations are of less rigour that need strengthening. Some minor editing needed to correct typos and inconsistencies in terminologies.

Leveraging on the proteomic assessment of the mouse knee joint with age (young, middle aged, and aged), the authors showed significant changes in a number of molecular signals, more so in male than female mice, and the PI3K/Akt pathway is one linking the progression from young to aged mice. PI3K/Akt is one of the major pathways modulating cell growth, proliferation, metabolism, survival, and angiogenesis. The authors then jumped to apha-Klotho as a candidate gene for further assessment. This link is rather weak as Klotho are known to link with numerous pathways such as IGF, FGF, Wnt, and TGFbeta. Thus, the authors need to demonstrate an unbiased selection for apha-Klotho amongst other candidate genes from the proteomic data.

The functional assessment of apha-Klotho in knee cartilage degeneration in mice is robust, that further substantiated previous observations for a protective role of apha-Klotho against KOA. Leveraging on a previous microarray dataset in vitro for a gain- and loss-of-function study of apha-Klotho, they made a connection to PI3K/Akt pathway. It would be relevant to show and rank PI3K/Akt with all pathways with significant changes.

The progression from apha-Klotho to the nuclear lamina is also unclear and has another “cheery picking” feel. Further, the assessment of changes in lamin A/C and B was done using the proteomic data that the authors have not presented data on the level of apha-Klotho changes. The link to changes in nuclear morphology is interesting and the assessments were robust with increased eccentricity to decreased expression of apha-Klotho.

As matrix stiffness is known to alter lamin A/C expression, the authors then attempt to link apha-Klotho expression to matrix stiffness via the cytoskeleton to the nuclear envelope. They showed increased matrix stiffness is associated with reduced expression of apha-Klotho and type II collagen, a major extracellular matrix for the function of articular cartilage. There are couple minor issues here. One is the use of immunostaining for type II collagen assessing intracellular proteins. This is problematic as type II collagen is a secreted protein and majority of the type II collagen produced would be outside the cells, and that this could reflect a secretion problem with age. Second is whether cells are cultured in the presence of ascorbic acid that is needed for the hydroxylation of collagens and elastin. Hydroxylation Proline and lysine residues in collagen is essential for collagen production and secretion. Also, assessment for Aggrecan as a control would further strengthen the argument.

Mechanical cues in epigenetic regulation are known, and the authors showed matrix stiffness increases methylation of apha-Klotho promoter, and the potentials of changing the methylation pattern of young and old chondrocytes through the matrix environment, and proposed that it is via the variable action of DNA methyltransferase 1 (DNMT1), and substantiated the finding using an inhibitor of DNMT1.

Finally, the author looked for ECM and related proteins in the proteomic data associated with

increased lamin A/C. This part was a bit unclear to me as to how the authors have correlated ECM proteins with the level of lamin A/C in the samples with ageing. They identified 155 such proteins and selected Lysyl Oxidase (LOX) as the candidate from one of the top 20 genes. Again, there is a feeling of “cherry picking” here. Would be good to show all 155 proteins or at least the top 20 and the ranking of LOX in relation to others. The use of BAPN would inhibit crosslinking of all collagens and elastin fibres. While in vivo assessment is important, the interpretation here needs to be careful as tendons and ligaments will be affected and the mechanical load on the joints could be impaired, not just changes in the stiffness of the cartilage. It would be good if the stiffness of the BAPN-treated cartilage can be measured to show the specific mechanical changes.

Overall, a very interesting study. The authors have provided a number of good association and functional studies. While they have proposed a model in Figure 5I to summarize their finding, I am not convinced that they have clearly demonstrated the link between nuclear stiffness to DNMT1 expression, and as DNMT1 is a global methyltransferase, how this can lead to changes in the PI3K/Akt pathway that are related specifically to the functional level of alpha-Klotho. Perhaps the authors can discuss this more clearly with the limitations of their findings in mind. The difference in gender is also intriguing and if indeed the level of estrogen in female mice has a protective effect on this matrix stiffness and degenerative outcomes, perhaps the authors should consider testing this directly in relation to their model, so that the finding is not mouse specific and strengthens the relevance to human. The authors could also speculate whether their finding is specific to KOA or has more general conceptual implication to other degenerative diseases and cancer biology.

Reviewer #5 (Remarks to the Author):

Summary

The manuscript “Epigenetic regulation of α -Klotho by age-related matrix stiffening drives cartilage degeneration” by Iijima and colleagues presents interesting and important results linking cartilage ECM mechanical changes associated with aging and changes in the aging protein α -Klotho and methylation of its promoter, nuclear lamina, and DNMT1 expression. The manuscript’s findings address a timely topic in understanding how biophysical cues associated with aging/diseased tissue initiate and/or amplify pathogenic signaling to contribute to tissue degeneration. In particular, the connection between ECM mechanosignaling and epigenetic remodeling is an emerging area of interest, with several notable recent reports elucidating novel pathways in this regard. Despite the enthusiasm for the topic and interesting results, further experiments are needed to fully establish the proposed mechanism and provide causal evidence for the pathway presented. In the current state, the causal links between different constituents of the molecular mechanism are not sufficiently supported by the data presented.

Major Comments

1) One premise of the proposed pathway is that α -Klotho inhibits PI3K/Akt signaling. Can the authors provide evidence that differences in α -Klotho expression are actually altering PI3K/Akt signaling? In Fig. 2I-K, there is some evidence from an α -Klotho knockdown/overexpression gene expression dataset that PI3KCA and AKT2 gene expression is modestly up and down regulated respectively. However, this does not necessarily result in a change in PI3K/Akt activity in response to α -Klotho

levels, and the authors suggest a more direct inhibitory mechanism. Differences in Akt phosphorylation with respect to α -Klotho would provide much more compelling evidence.

2) Much of the expression data relies on quantification of immunofluorescent images, which may not be the most reliable method to quantify protein abundance. Can the authors validate that these measurements reflect the true protein abundance in the cell through Western blots or other quantitative technique?

3) It is not clear how the changes in nuclear morphology and Lamin A/C levels are related to the DNMT1 expression changes and α -Klotho promoter methylation levels measured. Are these two phenomena linked and if so, can the authors provide evidence that one effects the other? As is, this is presented in a correlative manner, and even in the schematic pathway (Fig. 5I) it is not shown how nuclear stiffness and Lamin A/C are mechanistically connected to the other constituents.

4) Can the authors show that inhibition of DNMT1 results in more α -Klotho and less degeneration to support their molecular model? This would provide more direct evidence that DNMT1 is mediating α -Klotho expression and leading to phenotypic changes.

5) Is enhanced PI3K/Akt signaling the driver of increased DNMT1 expression and activity to reduce α -Klotho expression through promoter methylation? This is one potential mechanism, but it is not clear what is driving the increase in DNMT1 activity or why it is methylating the α -Klotho promoter specifically.

6) Similarly, is PI3K/Akt more activated on or in stiff matrices compared to soft (as reported in many other papers with a variety of cell types)? If so, is this due to a reduction in α -Klotho and subsequent loss of inhibition, or due to more direct integrin-mediated signaling?

7) Does the BAPN treatment actually reduce the stiffness of cartilage? Similarly, does it reduce DNMT1 expression/activity, or PI3K/Akt signaling, or other components of the proposed pathway?

Minor Comments

1) Can the authors show images of the Lamin A/C staining in Fig. 3 that were used to quantify expression? These images would demonstrate differences in nuclear morphology and the nuclear lamina that would be of interest to the nuclear mechanotransduction audience.

2) It is not clear what the shape metrics presented in Fig. 3F represent, how they are measured, or why they were selected. The authors should provide more context surrounding this experimental design and interpretation.

3) As mentioned in the summary above, several prominent articles demonstrating connections between mechanotransduction and epigenetic remodeling have been published very recently. Including a brief discussion of these findings in the context of these other recent works would be beneficial. Examples include:

- Killaars, Walker, Anseth. PNAS, 2020. <https://doi.org/10.1073/pnas.2006765117>
- Walker et al, Nature Biomedical Engineering, 2021. <https://doi.org/10.1038/s41551-021-00748-3>
- Stowers et al, Nature Biomedical Engineering, 2019. <https://doi.org/10.1038/s41551-019-0420-5>

- Jang et al. Nature Biomedical Engineering, 2020. <https://doi.org/10.1038/s41551-020-00657-x>

- Jones et al. Journal of Cell Biology, 2021. <https://doi.org/10.1083/jcb.202007152>

4) It is not clear why the phrase “PI3K/Akt signaling” is italicized in every instance in the paper.

5) More details regarding how fluorescence intensity was quantified should be provided in the Methods section.

6) The acronyms pOAR and OARSI should be defined in the manuscript.

7) In figure panels with gradients (for example, Fig. 2A,B), the text overlapping the gradient is somewhat difficult to read. Perhaps these can be modified to readability.

8) In Fig. 3A, how were these proteins quantified? It may be more informative to present the data in a dot plot, similar to other data in this manuscript. If that is what is presented in Fig. 3B, that was not immediately clear to me, especially given the significant difference in Lamin A/C but not Lamin B2, despite similar trends in the dot plot.

We are very grateful to the reviewers for providing such valuable, thoughtful, and thorough feedback. The comments and questions raised have prompted us to consider our data in new and interesting ways, the culmination of which we feel has greatly strengthened the work. Thank you for your time.

Reviewer #1

[Comment #1]

In this study, Iijima et al investigated whether α -Klotho plays a role in cartilage aging. The authors reported that an age-related loss of Klotho drives progressive cartilage degeneration. The authors believe that loss of Klotho was a consequence of age-related matrix stiffening, whereby a stiff matrix epigenetically inhibited Klotho expression in chondrocytes through lamin A/C-mediated mechanotransductive signaling. Exposing aged chondrocytes to a soft matrix restored a more youthful phenotype in vitro and enhanced cartilage integrity in vivo. They concluded that age-related alterations in ECM biophysical properties initiate pathogenic mechanotransductive cascades that promote Klotho promoter methylation and compromise cartilage integrity. Overall, this is a straightforward study which tests a hypothesis that age-related alterations in mechanotransductive signaling may epigenetically suppress α -Klotho expression. There are several concerns with this report.

Major

1. The novelty of this study is compromised partly because the roles of Klotho in the regulation of PI3K/AKT have been well established (Ref # 17, 40).

[Author action #1]

We agree that the role of α -Klotho in the regulation of PI3K/AKT is not the novel aspect of our work given that previous studies have shown that α -Klotho inhibits PI3K/Akt signaling in a variety of cell types¹⁻⁴. Rather, the main novelty of this study lies in the mechanistic link between age-related increases in ECM stiffness and epigenetic repression of α -Klotho.

To further add to the novelty of this work, we have considerably expanded the analyses by evaluating global DNA methylation change and performing a ChIP assay on chondrocytes cultured on matrices of varying stiffnesses. The assessment of 5 methylcytosine (5mC) at the genome revealed that culturing young and aged chondrocytes on stiff substrates impacted on global promoter methylation. However, the magnitude of effect was considerably lower than the effect observed on the *Klotho* promoter (**Figure 5A**). These findings suggest that *Klotho* hypermethylation is not solely a result of an overall increase in global methylation under stiff substrate conditions. In addition, a ChIP assay revealed that increasing stiffness induced recruitment of RNA Polymerase II (Pol II), active chromatin mark (H3K4M2), and c-MYC, a transcription factor known to be stimulated and/or regulated by kinase signaling at the *Dnmt1* promoter (**Figure 5D**). Further, a second series of ChIP assays revealed that increasing ECM stiffness promoted binding of DNMT1 at the *Klotho* promoter (**Figure 5C**). Taken together, these data show that the stiffness-dependent modulation of α -Klotho protein levels is attributed to recruitment of epigenetic modifiers, including DNMT1 at the *Klotho* promoter.

We also performed ChIP-qPCR on chondrocytes cultured on a stiff substrate with and without Latrunculin A to determine whether mechanotransductive signaling directly modulates *Dnmt1* and *Klotho* transcription (binding of PolII and other transcription factors at *Dnmt1* and *Klotho* promoter). Interestingly, the increased Pol II and c-MYC bindings at the *Dnmt1* promoter induced by stiff substrates were abolished with lat A treatment (**Figure 6C**). Lat A treatment also abolished the binding of DNMT1 at the *Klotho* promoter that was induced by a stiff substrate (**Figure 6D**). In summary, these data suggest that age-related matrix stiffness drives DNMT1 recruitment at the *Klotho* promoter via a cytoskeleton-mediated mechanotransductive pathway. In addition to the new data presented, we have also revised the INTRODUCTION, and DISCUSSION sections as follows:

Page 4 / Line 75-81 in INTRODUCTION: “Notably, this inhibition of α -Klotho was attributed to Klotho promoter methylation resulting from increased binding of DNA methyltransferase 1 (DNMT1). Finally, pharmacological reduction of cartilage stiffness in aged mice increased α -Klotho expression, ultimately restoring cartilage integrity. Taken together, these findings suggest that age-related alterations in matrix stiffness initiate pathogenic mechanotransductive cascades in chondrocytes, leading to epigenetic repression of the α -Klotho promoter and compromised cartilage integrity.”

Page 15-16 / Line 352-358 in DISCUSSION: “The stiffness-dependent modulation of α -Klotho was attributed to recruitment of epigenetic modifiers, including DNMT1, at the Klotho promoter. As a demonstration of the physiological relevance of these *in vitro* findings, we demonstrated enhanced cartilage integrity *in vivo* following pharmacological reduction of cartilage matrix stiffness in aged mice. Together, these findings suggest that preventing or reversing age-related matrix stiffening and the resulting pathogenic mechanotransductive cascades may serve as a promising therapeutic target to attenuate or even reverse the development of age-related KOA.”

References

1. Lim SW, et al. Klotho enhances FoxO3-mediated manganese superoxide dismutase expression by negatively regulating PI3K/AKT pathway during tacrolimus-induced oxidative stress. *Cell Death Dis* 8, e2972 (2017).
2. Zhu Y, et al. Klotho suppresses tumor progression via inhibiting PI3K/Akt/GSK3 β /Snail signaling in renal cell carcinoma. *Cancer Sci* 104, 663-671 (2013).
3. Shu G, et al. Restoration of klotho expression induces apoptosis and autophagy in hepatocellular carcinoma cells. *Cell Oncol (Dordr)* 36, 121-129 (2013).
4. Yamamoto M, et al. Regulation of oxidative stress by the anti-aging hormone klotho. *J Biol Chem* 280, 38029-38034 (2005).

[Comment #2]

2. It is not clearly justified why Klotho was singled out for study because aging affects numerous genes and proteins. Upregulation of DNMT1 increases methylation of numerous genes which may be involved in the maintenance of cartilage integrity.

[Author action #2]

We are grateful for the opportunity to better clarify our rationale. Our recent systematic review with meta-analysis identified elevated inflammation, increased senescence, and impaired autophagy as common denominators associated with aging-induced KOA¹, all of which are downstream targets of α -Klotho²⁻⁴. Moreover, it is well-established that α -Klotho is a regulator of the PI3K/Akt signaling pathway through inhibiting INS and INSR⁵, which we identified in our mass spectrometry-based analysis to be a critical driver of aging-induced progressive cartilage degeneration in male mice (see **Figure 1**). Indeed, our newly added ingenuity pathway analysis predicted INS and INSR as significantly activated upstream regulators for age-related protein changes in male mice cartilage (**Figure S3** shown below), further supporting our hypothesis that α -Klotho is a driver of age-related KOA. To better justify our focus on α -Klotho, we have revised the text as follows:

Page 7 / Line 143-150 in RESULTS: “Ingenuity pathway analysis¹⁷ using mass spectrometry data predicted activated INS/INSR signaling as significant regulators for age-related change in proteins in male mice cartilage (Figure S3). INS/INSR is upstream of PI3K/Akt signaling¹³. Therefore, we next sought to identify factors that inhibit the INS/INSR/PI3K axis. One key regulator of this pathway is the longevity protein, α -Klotho (Figure 1F)¹⁸. Accordingly, a recent systematic review identified elevated inflammation, increased senescence, and impaired autophagy as common denominators associated with aging-induced KOA in mice¹⁹, all of which are downstream of α -Klotho²⁰⁻²⁴.”

With respect to the second part of the comment, in this revised version, we sought to focus on demonstrating the direct link between α -Klotho and DNMT expression and cartilage integrity. To this end, we have added new studies investigating whether inhibition of *Dnmt1* blocks the negative effect of a stiff substrate on chondrocyte health. First, we showed that inhibition of *Dnmt1* by siRNA rescued the deleterious effect of a stiff matrix on both chondrocyte α -Klotho expression and chondrogenicity. Next, to demonstrate that the beneficial effect of *Dnmt1* expression on chondrocytic markers is mediated by α -Klotho, we treated cells seeded onto a stiff substrate with *Klotho* siRNA treatment in addition to *Dnmt1* inhibition by 5-Aza-2'-deoxycytidine (5 Aza). Indeed, we found that *Klotho* inhibition abrogated the beneficial effect of *Dnmt1* inhibition, as documented by decreased collagen II expression (see **Figure 5G-H** shown below). These findings are consistent with our finding that siRNA treatment to *Klotho* overrode the chondrogenic effects of a soft matrix, resulting in a significant decrease in type II collagen and aggrecan expression (see **Figure 4D**). We have revised the RESULTS section accordingly (see **Page 12-13 / Line 281-289 in RESULTS**).

Figure S3 in the revised manuscript

Figure 5G-H in the revised manuscript

Reference

1. Iijima H, et al. Meta-analysis integrated with multi-omics data analysis to elucidate pathogenic mechanisms of age-related knee osteoarthritis in mice. *J Gerontol A Biol Sci Med Sci*, (2022).
2. Zhou H, Pu S, Zhou H, Guo Y. Klotho as Potential Autophagy Regulator and Therapeutic Target. *Front Pharmacol* 12, 755366 (2021).
3. Kuro-o M. Klotho as a regulator of oxidative stress and senescence. *Biol Chem* 389, 233-241 (2008).
4. Hui H, et al. Klotho suppresses the inflammatory responses and ameliorates cardiac dysfunction in aging endotoxemic mice. *Oncotarget* 8, 15663-15676 (2017).
5. Yamamoto M, et al. Regulation of oxidative stress by the anti-aging hormone klotho. *J Biol Chem* 280, 38029-38034 (2005).

[Comment #3]

3. This study could benefit from an evidence-based RNA-seq analysis which will provide genes altered by aging.

[Author action #3]

Thank you for this suggestion. Given that previous studies have evaluated changes in the cartilage transcriptome with aging¹, in this current work, we focused on protein-level changes in aging cartilage, which, to our knowledge, has never been comprehensively assessed over time and according to sex. Nevertheless, to address this comment, we accessed previously archived RNA-seq data¹ and identified differentially expressed genes between male young and aged C57BL6 mice. We then compared the differentially expressed genes from the RNA-seq with differentially expressed genes from mass spectroscopy data, in which proteins were converted into their corresponding genes. We identified a total of 125 overlapping genes across the two datasets (see **Figure S2** below). Consistent with our primary analysis, KEGG enrichment analysis from the overlapping genes revealed that the PI3K/AKT pathway was still identified among top 5 differentially expressed pathways. Importantly, ECM-receptor interaction also emerged at the top of overlapping pathways, further supporting our focus on matrix-related alterations. We have revised the RESULTS section as follows:

Page 6 / Line 130-139 in RESULTS: “To expand our systems biology approach to include transcript level changes, we accessed archived RNA-seq data from young (3 months old) and aged (24 months old) C57BL6 mice¹⁶ and compared changes to those observed from mass spectrometry data. To do this, we first identified differentially expressed proteins between young and aged cartilage (yielding 1,057 proteins) and converted the identified proteins to their corresponding genes. When we compared the genes identified through mass spectrometry analysis with differentially expressed genes from young and aged transcriptomic data (yielding 3,506 genes), we identified 125 overlapping genes (Figure S2). KEGG enrichment analysis for the overlapping genes revealed that the PI3K/Akt signaling pathway was among the top five differential pathways.”

Figure S2 in the revised manuscript

Reference

1. Sebastian A, et al. Global Gene Expression Analysis Identifies Age-Related Differences in Knee Joint Transcriptome during the Development of Post-Traumatic Osteoarthritis in Mice. *Int J Mol Sci* 21 (2020).

[Comment #4]

4. How inhibition of LOX increases Klotho levels is not addressed. It is important to show whether LOX directly regulates DNMT1 expression and Klotho gene promoter methylation.

[Author action #4]

We apologize for the lack of clarity. In our animal experiments, we used LOX inhibitors as a physiological tool to reduce the cartilage stiffness of aged mice, given the age-related increased LOX protein abundance in our proteomics data set. LOX inhibitors are commonly used for reducing tissue stiffness via inhibition of crosslinking¹. Indeed, we confirmed that BAPN successfully reduced cartilage stiffness in aged mice, which was assessed by atomic force microscopy. In addition, mass spec data analysis revealed that LOX protein expression is significantly associated with lamin A/C expression, which is correlated with tissue stiffness².

Page 14 / Line 327-330 in RESULTS: “*Cartilage stiffness increases by approximately 2-3 fold with aging³⁴. We therefore tested whether reducing the cartilage stiffness in aged mice improves α -Klotho levels and cartilage integrity. For this purpose, we administered β -aminopropionitrile (BAPN), a known inhibitor of LOX⁴⁶, as a physiological means to reduce cartilage stiffness in aged mice (Figure 7C).*”

Still, we recognize that LOX can act intracellularly to modulate various cellular processes including epigenetic modification³, as pointed out by the reviewer. We have acknowledged these points in the DISCUSSION as follows:

Page 18 / Line 410-413 in DISCUSSION: “*While we confirmed that BAPN administration reduced cartilage stiffness, as was our primary goal, we acknowledge the possibility that LOX inhibition may have exerted direct biological effects, such as epigenetic modifications, on the cartilage⁶⁵. Investigating the mechano-dependent effect of reducing cartilage stiffness on α -Klotho level is warranted.*”

Reference

1. Kagan HM, et al. Lysyl oxidase: properties, specificity, and biological roles inside and outside of the cell. *J Cell Biochem* 88, 660-672 (2003).
2. Swift J, et al. Nuclear lamin-A scales with tissue stiffness and enhances matrix-directed differentiation. *Science* 341, 1240104 (2013).
3. Black JC, Whetstone JR. LOX out, histones: a new enzyme is nipping at your tails. *Mol Cell* 46, 243-244 (2012).

[Comment #5]

5. *It is not convincing that alterations of mechanotransductive signaling directly increases DNMT1 expression and Klotho promoter methylation. There is a lack of critical mechanistic investigation into how mechanotransductive signaling regulates DNMT1 and Klotho promoter methylation.*

[Author action #5]

We agree that additional mechanistic studies establishing a direct effect between mechanotransductive signaling-*Dnmt1*-*Klotho* methylation would strengthen the work. Our initial results revealed that chondrocytes cultured on an aged-like (stiff) substrate displayed increased DNMT1 expression, decreased α -Klotho expression, and increased *Klotho* promoter methylation. To more comprehensively establish whether these effects are mediated via mechanotransductive pathways, we have added a series of new experiments.

First, we inhibited actin polymerization of chondrocytes using Latrunculin A to evaluate whether blocking mechanotransductive signaling interferes with epigenetically-regulated chondrocyte responses to a stiff substrate. Indeed, we found that treatment with Latrunculin A reduced DNMT1 expression, decreased *Klotho* promoter methylation, increased *Klotho* gene expression, and enhanced the chondrogenicity of aged cells cultured on a stiff substrate. We also performed a ChIP assay for chondrocytes cultured on a stiff substrate with and without Latrunculin A (1 μ M) and found that Latrunculin A inhibited bindings of Pol II and c-MYC at the *Dnmt1* promoter, although active chromatin mark (H3K4M2) was not significantly changed (**Figure 6C** shown below). This

finding suggests that that the epigenetic modifier, DNMT1, is directly regulated by mechanical signals. Latrunculin A also decreased binding of DNMT1 at *Klotho* promoter (**Figure 6D** shown below). Taken together, these data support our hypothesis that increased DNMT1 and DNMT1-mediated epigenetic repression of *Klotho* in chondrocytes exposed to a stiff substrate are driven, at least in part, by mechanotransduction. We have added this new data in the RESULTS section (**Page 13-14, Lines 301-311; Fig 6C-D**).

Figure 6C-D in the revised manuscript

[Comment #6]

6. How the mechanotransductive force of lamin A/C is measured is not clearly stated.

[Author action #6]

We apologize for the confusion. We did not measure the mechanotransductive force of lamin A/C. Instead, similar to previous reports¹, we utilized cellular morphology using Phalloidin-stained cells as a surrogate marker of altered mechanotransductive forces². We confirmed that chondrocytes cultured on a stiff substrate displayed decreased formfactor (i.e., less roundness) with increased stress fiber formation (**Figure S10**). A stiff substrate also induced increased lamin A/C expression, as shown in **Figure S9**. These data indicate that increased mechanotransductive forces via stress fiber formation is accompanied by increased lamin A/C expression, as has been previously characterized in the literature^{3,4,5}. We have clarified this point in the main text as follows:

Page 10 / Line 217-220 in RESULTS: “Aged-like (stiff) substrates also increased stress fiber formation and altered chondrocyte cellular morphology (Figure S10), thereby further implicating mechanotransductive pathways in the stiffness-dependent regulation of cellular phenotype.”

Reference

1. Kim JH, et al. Matrix cross-linking-mediated mechanotransduction promotes posttraumatic osteoarthritis. *Proc Natl Acad Sci U S A* 112, 9424-9429 (2015).
2. Kumar S, et al. Viscoelastic retraction of single living stress fibers and its impact on cell shape, cytoskeletal organization, and extracellular matrix mechanics. *Biophysical journal* 90, 3762-3773 (2006).
3. Swift J, et al. Nuclear lamin-A scales with tissue stiffness and enhances matrix-directed differentiation. *Science* 341, 1240104 (2013).
4. Buxboim A, et al. Matrix elasticity regulates lamin-A,C phosphorylation and turnover with feedback to actomyosin. *Curr Biol* 24, 1909-1917 (2014).
5. Buxboim A, et al. Coordinated increase of nuclear tension and lamin-A with matrix stiffness outcompetes lamin-B receptor that favors soft tissue phenotypes. *Molecular biology of the cell* 28, 3333-3348 (2017).

[Comment #7]

7. The authors should investigate whether chromosome remodeling alters DNMT1 expression and *Klotho* promoter methylation. It was reported that histone methylation regulates *Klotho* gene transcription and expression.

[Author action #7]

Great suggestion. In this revision, we provided evidence showing the epigenetic impact of ECM stiffness on transcription of *Dnmt1* and *Klotho*. *In silico* analysis (UCSC Genome Browser) revealed the potential binding of epigenetic modifiers (including histone modifications) and transcriptional factors at the *Dnmt1* and *Klotho* promoters. Previously, we demonstrated the epigenetic regulation of these promoters in injured skeletal muscles¹ and C2C12 myoblasts².

Using ChIP analysis, we found that increased ECM stiffness increased binding of DNMT1 at the *Klotho* promoter (**Figure 5C** shown below), indicating that increased *Klotho* promoter methylation under stiff microenvironment is attributed, at least in part, to the increased binding of DNMT1 at *Klotho* promoter. Of note, increasing ECM stiffness also showed a decreased binding of RNA Polymerase II (Pol II), active chromatin mark (H3K4M2), as well as increased repressive chromatin mark (H3K9M2) at *Klotho* promoter (**Figure S13** shown below). Elucidation of the mechanisms underlying stiffness-dependent *Klotho* transcriptional regulation by histone modifications is of interest for future studies.

An additional series of ChIP analyses also revealed that stiff substrates induced recruitment of Pol II and H3K4M2, but not H3K9M2, at the *Dnmt1* promoter (**Figure 5D** shown below). We also observed a significant increase in binding of c-MYC, a transcription factor known to be stimulated and/or regulated by kinase signaling, at the *Dnmt1* promoter in response to increased ECM stiffness (**Figure 5D** shown below). Taken together, these findings indicate that *Dnmt1* is directly regulated by substrate stiffness.

We also performed ChIP-qPCR on chondrocytes cultured on a stiff substrate with and without Latrunculin A to determine whether mechanotransductive signaling directly modulates *Dnmt1* and *Klotho* transcription (binding of PolII and other transcription factors at *Dnmt1* and *Klotho* promoter). Interestingly, the increased Pol II and c-MYC bindings at the *Dnmt1* promoter induced by stiff substrates were abolished by lat A treatment (**Figure 6C**). Lat A treatment also abolished the change in binding of DNMT1 at the *Klotho* promoter induced by a stiff substrate (**Figure 6D**). In summary, these data suggest that age-related matrix stiffness drives DNMT1 recruitment at the *Klotho* promoter via cytoskeleton-mediated mechanotransductive pathway. We have revised RESULTS section accordingly (see **Page 12-14 / Line 263-280; Line 301-311**).

Figure 5C-D in the revised manuscript

Figure S13 in the revised manuscript

Figure 6C-D in the revised manuscript

Reference

1. Sahu A, et al. Age-related declines in α -Klotho drive progenitor cell mitochondrial dysfunction and impaired muscle regeneration. *Nat Commun* 19, 4859 (2018).
2. Cheikhi A, et al. Mitochondria are a substrate of cellular memory. *Free Radic Biol Med* 130, 528-541 (2018).

[Comment #8]

8. There is a concern on the systemic delivery of Klotho protein which complicates interpretation of the direct effect of the exogenous Klotho in chondrocytes in cartilage. The changes in cartilage may be secondary to the systemic effects of Klotho.

[Author action #8]

We fully agree, and we received a similar comment from Reviewers #2 and #3 (see **Comments #13** and **#19-20**, respectively). To address this comment, we first evaluated whether systemic α -Klotho administration may affect other tissues in the joint, thereby potentially confounding our findings. To do this, we performed histological analysis of the synovium in animals receiving osmotic pump delivery of α -Klotho versus vehicle controls. We found that supplementation with α -Klotho suppressed synovial inflammation in both aged and *Klotho*^{+/-} mice (**Reference Figure 1A-E**).

In view of these findings and given that experiments utilizing systemic delivery of α -Klotho deviate from our main hypothesis focused on the ECM- α -Klotho axis, we have removed that data from the manuscript. Instead, we have focused on more comprehensively establishing the link between matrix stiffening and epigenetic regulation of α -Klotho in chondrocytes, as per the reviewers' recommendations.

Reference Figure 1

Reference

1. Hui H, et al. Klotho suppresses the inflammatory responses and ameliorates cardiac dysfunction in aging endotoxemic mice. *Oncotarget* 8, 15663-15676 (2017).

[Comment #9]

9. It is not convincingly addressed whether downregulation of Klotho is the cause or consequence of cartilage stiffness and remodeling. It is known that a decrease in Klotho leads to aging phenotype (tissue remodeling, calcification and stiffening). Thus, aging-related downregulation of Klotho may be the cause of cartilage stiffness.

[Author action #9]

This is an interesting point. Our data suggest that a stiff substrate causes *Klotho* promoter hypermethylation

(Figure 5A). As a more stringent test of whether downregulation of α -Klotho is a direct consequence of cartilage stiffness, we added new data demonstrating that inhibition of actin polymerization by Latrunculin A inhibited *Klotho* promoter hypermethylation in chondrocytes cultured on a stiff substrate (Figure 6B shown below). Although we cannot exclude the potential effect of DNA methylation changes on other gene promoters, these findings indicate that *Klotho* expression is regulated by substrate stiffness. We have revised the text as follows:

Page 13 / Line 301-303 in RESULTS: “Increased α -Klotho levels following lat A treatment were also accompanied by a decrease in methylation level of the *Klotho* promoter, while global DNA methylation was not drastically altered (Figure 6B).”

Additionally, to further support this finding *in vivo*, we systemically injected BAPN into aged mice and found that BAPN treatment reduced cartilage stiffness (Figure 7E-F) and enhanced α -Klotho expression in cartilage (Figure 7I). Despite these supporting data, we cannot rule out the possibility that α -Klotho is also an upstream regulator of matrix remodeling. Indeed, PI3K/Akt signaling, which is negatively regulated by α -Klotho¹⁻⁴, has a role in matrix organization (Figure 1E). We have modified the DISCUSSION to acknowledge the bidirectional interaction between chondrocytes and their ECM, a phenomenon known as “dynamic reciprocity”⁵, as follows:

Page 19 / Line 413-418 in DISCUSSION: “Along a similar vein, it is also worthwhile to note that the improved α -Klotho resulting from reduced ECM stiffness may, in turn, propagate a cascade of matrix remodeling and subsequent improvement in the matrix stiffness⁶⁶. This bidirectional interaction between α -Klotho in chondrocytes and their surrounding ECM, a phenomenon known as dynamic reciprocity⁶⁷, is an area of interest for future *in vivo* studies.”

Figure 6B in the revised manuscript

Reference

1. Lim SW, et al. *Klotho* enhances FoxO3-mediated manganese superoxide dismutase expression by negatively regulating PI3K/AKT pathway during tacrolimus-induced oxidative stress. *Cell Death Dis* 8, e2972 (2017).
2. Zhu Y, et al. *Klotho* suppresses tumor progression via inhibiting PI3K/Akt/GSK3 β /Snail signaling in renal cell carcinoma. *Cancer Sci* 104, 663-671 (2013).
3. Shu G, et al. Restoration of *klotho* expression induces apoptosis and autophagy in hepatocellular carcinoma cells. *Cell Oncol (Dordr)* 36, 121-129 (2013).
4. Yamamoto M, et al. Regulation of oxidative stress by the anti-aging hormone *klotho*. *J Biol Chem* 280, 38029-34 (2005).
5. Bissell MJ, Hall HG, Parry G. How does the extracellular matrix direct gene expression? *J Theor Biol* 99, 31-68 (1982).

[Comment #10] 10. A concern is that the role of *Klotho* in the blood in regulation of cartilage homeostasis is ignored. *Klotho* is primarily expressed in the kidney and released to the circulation.

[Author action #10]

As mentioned above in **Author action #8**, we removed the studies utilizing systemic delivery of α -Klotho since these experiments do not directly address our main hypothesis interrogating the ECM- α -Klotho axis as a driver of age-related KOA.

[Comment #11]

11. There is no convincing data showing that exogenous Klotho is uptaken by chondrocytes in cartilage.

[Author action #11]

We fully agree, and, as such, we have removed this data from the manuscript, as described in detail above.

Reviewer #2

[Comment #12]

The authors present support for a mechanism of age-related osteoarthritis (OA) driven by matrix stiffening, methylation of the Klotho promoter, and over-activity of PI3K/Akt. The series of experiments are logical and rigorously performed, and the results are very clearly presented. The baseline experiments of OA and mass spec on male and female mice are important benchmarks for the other findings and are well done. While some of the individual findings are only modest extensions of published work, the “total package” provides a compelling new paradigm that will be important for the field in terms of understanding the connection between matrix changes and risk for OA with aging. The comments below are intended to improve the impressive body of work already present in the manuscript.

1) Are the methylation changes that occur in response to substrate stiffness specific to Klotho promoter? The authors show that DNMT1 levels do change as well, which makes it difficult to tell whether the effects on Klotho are just due to a general mechanism of increased methylation under stiff substrate conditions. Perhaps a series of control sites could be selected for methylation-specific qPCR to compare to Klotho? Or better yet a genome-wide analysis to normalize the changes occurring at Klotho to rest of genome?

[Author action #12]

Excellent points. We measured the 5-methylcytosine level at the genome to address these questions. We found that, while global DNA methylation levels were significantly increased in response to a stiff matrix, the magnitude of effect was considerably lower than the effect observed on the Klotho promoter (Figure 5B and Figure S12 shown below). These findings indicate that Klotho hypermethylation cannot be solely explained by global increases in 5-methylcytosine level under stiff substrate conditions. The binding of DNMT1 together with specific histone modifications (like H3K4M2) at the Klotho promoter in response to a stiff matrix appeared to have contributed to transcription of Klotho. Of note, identification of gene-specific DNA methylation changes by methylome profiling may require future investigation, but is beyond the scope of current study. We have revised as follows:

Page 11 / Line 256-259 in RESULTS: “Culturing young and aged chondrocytes on stiff substrates also promoted global DNA methylation, though the magnitude of effect was considerably lower than the effect observed on the Klotho promoter (Figure 5A and Figure S12A).”

**Figure 5B in the revised manuscript
(Young chondrocytes)**

**Figure S12A in the revised manuscript
(Aged chondrocytes)**

[Comment #13]

2) Figure 2G/H is impressive, but a bit surprising given the poor regenerative capacity of cartilage in older mice (and humans). What is the explanation from the authors that just 2 weeks of systemic Klotho would actually improve cartilage quality (as opposed to preventing further degradation). More support is needed for such a claim, but it is recognized that these experiments are challenging to perform. The legend indicates n=7/8. Were these males? What age? Were timecourse experiments performed or was the 2-week implantation the only experiment performed? There can be a transient increase in GAG staining during the early phases of OA. It would be important to show that these acute improvements persisted to some extent. Would the Klotho pumps “rescue” the excessive degradation in the Klotho +/- middle-aged mice? That would presumably provide more support that Klotho is sufficient to improve joint health after a moderate level of OA, especially given that in this case the phenotype is in part caused by insufficient Klotho.

[Author action #13]

We received similar comments from Reviewers #1 and #3 (see **Comments #8** and **#19-20**, respectively). In response to this comment, we performed a series of new experiments. First, we sought to determine whether the systemic delivery of α -Klotho protein increase circulating α -Klotho protein level, as suggested by Reviewer #3 (see **Comment #20**). For this purpose, we performed ELISA analysis and compared circulating α -Klotho levels in animals receiving systemic delivery of α -Klotho and vehicle control counterparts. However, we observed no significant difference between groups (**Reference Figure 2A**).

However, we cannot rule out the possibility that cartilage changes after α -Klotho systemic delivery are the secondary effect of suppression of synovial inflammation. Indeed, we found that α -Klotho administration suppressed synovial inflammation which is accompanied by improved cartilage integrity in aged mice (**Reference Figure 2B-C**). In addition, we also confirmed that α -Klotho systemic delivery rescued cartilage degeneration in mice heterozygously deficient for α -Klotho (*Klotho* HET). These mice also displayed a suppressed synovial inflammation with α -Klotho systemic delivery (**Reference Figure 2D-F**). These findings provide early evidence that soluble α -Klotho can restore joint health. Since the potential effects of systemic α -Klotho delivery are outside the scope of this paper and may not directly pertain to the direct effect of matrix stiffness on α -Klotho in chondrocytes, we have removed the data pertaining to systemic α -Klotho delivery.

Reference Figure 2

[Comment #14]

3) One relevant reference seemed to be missing: Martinez-Rodono, Protein Cell, 2020: <https://pubmed.ncbi.nlm.nih.gov/31950348/>.

[Author action #14]

We apologize this oversight. We have added this reference as follows:

Page 17 / Line 380-382 in DISCUSSION: “Another study also demonstrated that overexpression of α -Klotho, combined with soluble TGF- β supplementation, enhanced chondrocyte health and cartilage integrity in a chemically-induced model of KOA⁵¹.”

[Comment #15]

4) In general, the methods were quite detailed, but one exception that should be clarified is the number of cells used to quantify immunofluorescence. This is an important readout for both figures 3 (tissue sections) and 4 (cells) because I did not see any bulk analysis using western blot that would encompass many more cells. Could these type of bulk data (for Klotho, Col2) be pulled and highlighted from the mass spec experiments?

One example – the Fig. 4 legend gives n=5. I assume that is 5 different chondrocyte isolations (from individual or pooled mice). How many cells were quantified? Only 3-4 cells are shown in representative images, but of course having many cells over numerous random fields across the culture would make sure this is a consistent effect.

[Author action #15]

We are grateful for the opportunity to clarify our methods. We received a similar comment from Reviewer #3 (**Comment #21**). In each immunofluorescence-based quantification, we used 5 or 10 randomly selected images for 20x and 63x images, respectively, per biological replicate. Data for the images were then averaged for each independent sample and used for statistical analysis (e.g., for 5 biological replicates, data were shown as n = 5 wells/group). We have provided the number of cells or nuclei used for the quantitative analysis in each figure caption, and we note in the statistical analysis section that “n” represents the number of independent observations of knees or cells from different animals (**Page 27 / Line801-803**).

[Comment #16]

5) The following sentence should probably be re-worded, as the osmotic pump for gain of function is also a systemic approach and not specific to the joints. “Therefore, to more directly evaluate whether α -Klotho attenuates an aged cartilage phenotype, we employed a gain-of-function paradigm.” While serial intra-articular injections or some form of stable Klotho overexpression in the joint would be interesting, that may be beyond the scope of this study.

[Author action #16]

We fully agree. Since it is clear from the collective reviewer comments that systemic delivery of α -Klotho is outside the scope of this study, we have removed these experiments from the manuscript.

Reviewer #3

[Comment #17]

General comments

This paper identified Klotho as a chondroprotective factor and may play a critical role in pathogenesis of age-related chondrocyte degeneration. This finding may promote better understanding of pathophysiology of osteoarthritis to develop a novel therapeutic strategy.

Specific comments

Major points:

The authors’ main findings are:

1. Cartilage degeneration with age was associated with disruption of PI3K/Akt signaling and with decrease in

Klotho expression in chondrocytes in mice.

2. A decrease in *Klotho* expression was associated with accelerated cartilage degeneration in mice.
3. Administration of soluble *Klotho* ameliorated cartilage degeneration in mice.
4. Treatment of cultured chondrocytes with soluble *Klotho* attenuated PI3K/Akt signaling.
5. A decrease in endogenous *Klotho* expression in cultured chondrocytes enhanced PI3K/Akt signaling.
6. Chondrocytes cultured on stiff substrates had lower *Klotho* expression levels than those cultured on soft substrates.

These findings support the authors' conclusion that increase in matrix stiffness with age may accelerate cartilage degeneration through suppressing Klotho expression.

However, the following points are not clear:

1. It is not clear whether soluble *Klotho* ameliorated cartilage degeneration through inhibiting PI3K/Akt signaling. The authors did not provide data in support of the hypothesis that the therapeutic effect of soluble *Klotho* on cartilage degeneration might depend on the ability of soluble *Klotho* to inhibit PI3K/Akt signaling. The findings #4 and #5 were not proof of the hypothesis, because they were consistent with previous reports showing that soluble *Klotho* inhibited PI3K/Akt signaling (Kurosu H, et al. *Science* 309:1829-33, 2005). Besides, soluble *Klotho* was reported to regulate activity of several ion transporters/channels and growth factors (for review; Kuro-o M. *Nat Rev Nephrol* 15:27-44, 2019). The authors should discuss these points (i.e. the meaning of the horizontal arrow in Figure 5I).

[Author action #17]

Thank you for raising this important point. We received a similar comment from Reviewers #1 and #2 (Comments #8 and #13, respectively). As noted above, it is clear from the collective reviewer comments that systemic delivery of α -*Klotho* may have additional mechanisms of action, and we have, thus, removed the experiments utilizing soluble α -*Klotho* administration from the manuscript.

As per recommendations by Reviewer #4 (see Comment #35), we sought to quantify PI3K protein expression and phosphorylation in chondrocytes cultured on pAAm gels (soft, medium, and stiff) using western blotting. Results revealed that chondrocytes cultured on soft substrates had lower levels of PI3K phosphorylation when compared to cells cultured on stiff substrates (see Reference Figure 3), suggesting that a young-like (soft) matrix suppresses PI3K/Akt signaling. However, this suppressive effect on PI3K phosphorylation was blocked when chondrocytes seeded on soft substrates were treated with *Klotho* siRNA (see Reference Figure 3). Given that matrix stiffness regulates chondrogenicity *via* α -*Klotho*, these findings suggest that matrix stiffness regulates chondrocyte health *via* modulation of α -*Klotho* expression with subsequent changes in PI3K/Akt signaling. Despite the fact that these data support our hypothesis and previous literature, we have major concerns about the reliability of the western blot data because of strong background noise when considering the full blot. Although we have tried several different antibodies for PI3K and Akt, western blot band signals were not necessarily consistent. As a result, we have greatly downplayed the role of PI3K/Akt throughout the current manuscript in order to prevent any confusion.

Reference Figure 3

[Comment #18]

2. It is not clear whether Klotho may act on synoviocytes and macrophages in the knee joint. The PI3K/Akt signaling affects not only chondrocyte function but also function of other cells in the knee joint, including synoviocytes and macrophages (Sun K, et al. *Osteoarthritis Cartilage* 28:400-9, 2020). All these cells participate in the development of osteoarthritis. Therefore, it is possible that soluble Klotho may act on all these cells in an autocrine, paracrine, and/or endocrine manner. The authors should discuss why they focused on chondrocytes.

[Author action #18]

We fully agree and, as described above, Reviewers #1 and #2 (**Comments #8 & #13**) raised a similar point. As noted above, we have removed the experiments utilizing soluble α -Klotho administration via osmotic pump from the manuscript.

[Comment #19]

3. It is not clear which soluble Klotho may be of physiological importance for maintenance of chondrocytes, autocrine/paracrine Klotho from chondrocytes or endocrine Klotho secreted from renal tubular cells. The major source of circulating soluble Klotho is the kidney (Hu MC, et al. *J Am Soc Nephrol* 27:79-90, 2016). The reviewer was unable to find many papers reporting that patients with chronic kidney disease, whose serum Klotho levels should be low, were at high risk for osteoarthritis. However, the authors showed that systemic administration of soluble Klotho prevented chondrocyte degeneration. The authors should discuss a mode of action of soluble Klotho in chondrocytes.

[Author action #19]

We thank you for this very thoughtful comment. As described above, given the valid concerns regarding the confounding variables associated with systemic versus local α -Klotho levels, we have removed the data of the systemic delivery of α -Klotho.

[Comment #20]

4. How were the serum levels of soluble Klotho, FGF23, phosphate, calcium, and active vitamin D changed in mice treated with soluble Klotho protein? Both membrane and soluble Klotho form binary complexes with receptors for fibroblast growth factor (FGF) to function as the receptor for FGF23 (Chen G, et al. *Nature* 553:461-6, 2018). FGF23 is a bone-derived endocrine factor that functions as a phosphaturic hormone and a counter-regulatory hormone against active vitamin D and parathyroid hormone. Administration of soluble Klotho may affect these factors and thus pathophysiology of OA.

[Author action #20]

Since systemic delivery of α -Klotho is outside the scope in this study (see **Author action #18**), these additional assays were not performed.

[Comment #21]

Minor points

1. Figure 2B, 2D, 3H, S4b, S4C, S4D, S4F, 5B, 5C, 5G: What does the y-axis represent? Protein levels determined by the mass-spec or mRNA levels determined by qPCR?

[Author action #21]

We apologize the lack of clarity. The y-axis represents fluorescence intensity/cell (Fig.2B, 2D, 3H, S4B, S4C, S4D, S4F, and 5G in the original submission) or protein abundance quantified by mass spectrometry-based proteomics (Fig.5B, 5C in the original submission). To clarify these points, we changed the name of y-axis to “fluorescence intensity (AU)” or “protein abundance”, as appropriate. We have also provided explanation within each figure caption.

[Comment #22]

2. Figure S2, 5C: What does the x-axis represent? Protein levels determined by the mass-spec or mRNA levels determined by qPCR?

[Author action #22]

We again apologize for the confusion. Protein levels described were determined by immunofluorescence. We changed the name of y-axis to “fluorescence intensity (AU)” and have clarified the labels in the figure caption.

[Comment #23]

3. Page 11, Line 248: via via -> via.

[Author action #23]

Corrected. We apologize for the mistake.

Reviewer #4

[Comment #24]

This is an interesting manuscript providing mechanistic insight into the relationship between changes in mechanical property with age and knee osteoarthritis (KOA). The authors have used a combination of in vitro/in vivo biochemical /biophysical assessments to link changes in the stiffness of the ECM in the knee cartilage that could regulate the epigenetic changes in the promoter of alpha-Klotho, a gene well-known to be involved in ageing and longevity. Their conclusion is based on computational, statistical, and functional assessments in consolidating their findings. Much of the pathways and genes identified/selected for assessments are known to be related to cartilage biology and degeneration, and the novelty is in linking these together providing a mechanistic insight. Data provided are clear, but some key associations are of less rigour that need strengthening. Some minor editing needed to correct typos and inconsistencies in terminologies.

Leveraging on the proteomic assessment of the mouse knee joint with age (young, middle aged, and aged), the authors showed significant changes in a number of molecular signals, more so in male than female mice, and the PI3K/Akt pathway is one linking the progression from young to aged mice. PI3K/Akt is one of the major pathways modulating cell growth, proliferation, metabolism, survival, and angiogenesis. The authors then jumped to alpha-Klotho as a candidate gene for further assessment. This link is rather weak as Klotho are known to link with numerous pathways such as IGF, FGF, Wnt, and TGFbeta. Thus, the authors need to demonstrate an unbiased selection for alpha-Klotho amongst other candidate genes from the proteomic data.

[Author action #24]

Thank you for the opportunity to clarify our rationale. Our recent systematic review with meta-analysis identified elevated inflammation, increased senescence, and impaired autophagy denominators associated with aging-induced KOA¹, all of which are downstream targets of α -Klotho²⁻⁴. Moreover, it is well-established that α -Klotho is a regulator of the PI3K/Akt signaling pathway through inhibiting INS and INSR⁵, which we identified in our mass spectrometry-based analysis to be a critical driver of aging-induced progressive cartilage degeneration in male mice (see **Figure 1**). Since our new data analyzed by ingenuity pathway analysis predicted INS and INSR as significantly activated upstream regulators for aging process in male mice cartilage (**Figure S3** shown below), we hypothesized that α -Klotho in cartilage is decreased with aging. It was the culmination of these studies that led to our interest in α -Klotho as a potential driver of age-related KOA. To better justify our focus on α -Klotho, we have revised the text as follows:

Page 7 / Line 143-150 in RESULTS: “Ingenuity pathway analysis¹⁷ using mass spectrometry data predicted activated INS/INSR signaling as significant regulators for age-related change in proteins in male mice cartilage

(Figure S3). *INS/INSR* is upstream of *PI3K/Akt* signaling¹³. Therefore, we next sought to identify factors that inhibit the *INS/INSR/PI3K* axis. One key regulator of this pathway is the longevity protein, α -Klotho (Figure 1F)¹⁸. Accordingly, a recent systematic review identified elevated inflammation, increased senescence, and impaired autophagy as common denominators associated with aging-induced KOA in mice¹⁹, all of which are downstream of α -Klotho²⁰⁻²⁴.”

Figure S3 in the revised manuscript

Reference

1. Iijima H, et al. Meta-analysis integrated with multi-omics data analysis to elucidate pathogenic mechanisms of age-related knee osteoarthritis in mice. *J Gerontol A Biol Sci Med Sci*, (2022).
2. Zhou H, Pu S, Zhou H, Guo Y. Klotho as Potential Autophagy Regulator and Therapeutic Target. *Front Pharmacol* 12, 755366 (2021).
3. Kuro-o M. Klotho as a regulator of oxidative stress and senescence. *Biol Chem* 389, 233-241 (2008).
4. Hui H, et al. Klotho suppresses the inflammatory responses and ameliorates cardiac dysfunction in aging endotoxemic mice. *Oncotarget* 8, 15663-15676 (2017).
5. Yamamoto M, et al. Regulation of oxidative stress by the anti-aging hormone klotho. *J Biol Chem* 280, 38029-34 (2005).

[Comment #25]

The functional assessment of α -Klotho in knee cartilage degeneration in mice is robust, that further substantiated previous observations for a protective role of α -Klotho against KOA. Leveraging on a previous microarray dataset in vitro for a gain- and loss-of-function study of α -Klotho, they made a connection to *PI3K/Akt* pathway. It would be relevant to show and rank *PI3K/Akt* with all pathways with significant changes.

[Author action #25]

Thank you for the suggestion. We have provided the rank of *PI3K/AKT* signaling among all KEGG pathways as shown in Figure S4C shown below.

Figure S4C in the revised manuscript.

[Comment #26]

The progression from alpha-Klotho to the nuclear lamina is also unclear and has another “cheery picking” feel. Further, the assessment of changes in lamin A/C and B was done using the proteomic data that the authors have not presented data on the level of alpha-Klotho changes. The link to changes in nuclear morphology is interesting and the assessments were robust with increased eccentricity to decreased expression of alpha-Klotho.

[Author action #26]

Thank you for this feedback. Studies have shown a link between α -Klotho decline and alterations in mechanotransduction signals within renal fibrosis¹⁻². Supporting this, our mass spectrometry data identified mechanoresponsive pathways changing with aging (Figure 1C). The collective evidence led us to evaluate the link between age-related α -Klotho decline and reorganization of the nuclear envelope as a surrogate marker for altered mechanotransductive signals. We have explained these points as follows:

Page 8 / Line 171-174: “What drives a decline in chondrocyte α -Klotho expression over time? Our proteomic data revealed that mechanotransductive pathways were significantly disrupted in aged murine cartilage (Figure 1C). This led us to question whether age-related declines in α -Klotho are a consequence of altered mechanical signals from the microenvironment.”

As the reviewer pointed out, mass spectrometry analysis did not identify changes in α -Klotho, and this may be a function of low protein abundance. To clarify the relationship between α -Klotho and lamin expression, we performed integrative analysis of proteomics (i.e., lamin A/C and lamin B2 protein abundance) and histology (i.e., α -Klotho expression quantified by immunofluorescence) data. The results revealed an inverse relationship between both lamin A/C and lamin B2 protein abundance and α -Klotho immunofluorescence (see Figure S7B shown below).

Figure S7B in the revised manuscript

References

1. Lim SW, et al. Klotho enhances FoxO3-mediated manganese superoxide dismutase expression by negatively regulating PI3K/AKT pathway during tacrolimus-induced oxidative stress. *Cell Death Dis* 8, e2972 (2017).
2. Zhu Y, et al. Klotho suppresses tumor progression via inhibiting PI3K/Akt/GSK3 β /Snail signaling in renal cell carcinoma. *Cancer Sci* 104, 663-671 (2013).

[Comment #27]

As matrix stiffness is known to alter lamin A/C expression, the authors then attempt to link alpha-Klotho expression to matrix stiffness via the cytoskeleton to the nuclear envelope. They showed increased matrix stiffness is associated with reduced expression of alpha-Klotho and type II collagen, a major extracellular matrix for the function of articular cartilage. There are couple minor issues here. One is the use of immunostaining for type II collagen assessing intracellular proteins. This is problematic as type II collagen is a secreted protein and majority of the type II collagen produced would be outside the cells, and that this could reflect a secretion problem

with age. Second is whether cells are cultured in the presence of ascorbic acid that is needed for the hydroxylation of collagens and elastin. Hydroxylation Proline and lysine residues in collagen is essential for collagen production and secretion. Also, assessment for Aggrecan as a control would further strengthen the argument.

[Author action #27]

Thank you for these very helpful suggestions. We agree that quantifying the secreted type II collagen together with consideration of ascorbic acid is important. To address this, we first quantified markers of chondrogenicity (type II collagen and aggrecan) in chondrocytes cultured on each pAAM gel (soft, medium, and stiff) with and without ascorbic acid (50µg/mL) as shown in **Figure S11A**. The concentration used has been shown to enhance chondrogenic gene expression in murine chondrocytes¹. Consistent with our original data, a stiff substrate reduced type II collagen, but we observed no interaction between substrate stiffness (soft, medium, and stiff) and the presence of ascorbic acid in our linear regression analysis (see **Figure S11B** shown below). We have also added quantification of aggrecan by immunofluorescence, as suggested. The data reveal a similar response to that observed for type II collagen (see **Figure S11B** shown below).

Next, we performed an ELISA for secreted type II collagen in conditioned medium. In these analyses, we concentrated the protein in the conditioned medium to detect the soluble protein by ELISA. Results revealed that chondrocytes cultured on stiff substrate secreted less soluble type II collagen compared to chondrocytes cultured on soft substrates, independent of the presence of ascorbic acid (see **Figure S11C** shown below). We confirmed that the number of chondrocytes was similar across the different conditions to ensure that the data were not skewed by differences in cell number (see **Figure S11D** shown below). These findings support the conclusion that a stiff substrate inhibits secretion of type II collagen². We have revised the RESULTS as follows:

Page 10 / Line 225-237 in RESULTS: “Several studies have shown that ascorbic acid, an essential cofactor for lysyl hydroxylase and prolyl hydroxylase, is required for collagen biosynthesis *in vitro*³⁸. Therefore, we performed a sensitivity analysis to determine whether the stiffness-dependent regulation of chondrogenicity markers (type II collagen and aggrecan expression) was influenced by the addition of ascorbic acid to the culture medium. However, we found that the expression of chondrogenic markers was not significantly influenced by the presence of ascorbic acid (Figure S11A-B). When we assessed the secretion of type II collagen by chondrocytes under each of the experimental conditions (+/- ascorbic acid on soft, medium, or stiff substrates) using Enzyme-linked immuno-sorbent assay (ELISA), we found that chondrocytes cultured on stiff substrates secreted less type II collagen compared to those cultured on soft substrates, an effect that was independent from ascorbic acid supplementation (Figure S11C). We note that the number of chondrocytes was similar across the groups, confirming that the reduced secretion of type II collagen on stiff substrates was not a function of fewer chondrocytes (Figure S11D).”

Figure S11 in the revised manuscript

References

1. Lindsey RC, et al. Vitamin C effects on 5-hydroxymethylcytosine and gene expression in osteoblasts and chondrocytes: Potential involvement of PHD2. *PLoS One* 14, e0220653 (2019).
2. Asopa V, et al. The effects of age and cell isolation on collagen II synthesis by articular chondrocytes: evidence for transcriptional and posttranscriptional regulation. *BioMed Research International* 2020, (2020).

[Comment #28]

Finally, the author looked for ECM and related proteins in the proteomic data associated with increased lamin A/C. This part was a bit unclear to me as to how the authors have correlated ECM proteins with the level of lamin A/C in the samples with ageing. They identified 155 such proteins and selected Lysyl Oxidase (LOX) as the candidate from one of the top 20 genes. Again, there is a feeling of “cherry picking” here. Would be good to show all 155 proteins or at least the top 20 and the ranking of LOX in relation to others. The use of BAPN would inhibit crosslinking of all collagens and elastin fibers. While in vivo assessment is important, the interpretation here needs to be careful as tendons and ligaments will be affected and the mechanical load on the joints could be impaired, not just changes in the stiffness of the cartilage. It would be good if the stiffness of the BAPN-treated cartilage can be measured to show the specific mechanical changes.

[Author action #28]

Great points, thank you. We have now summarized the top 20 proteins contributing to PC2 (see Figure 7C shown below). LOX (highlighted in the heat map) ranked 16th. Regarding changes in cartilage mechanical properties following BAPN administration, we received a similar comment from Reviewer #5 (Comment #36). To address this, we measured the stiffness of the medial tibia using AFM and compared the Young’s modulus between the two groups: aged+saline and aged+BAPN; n = 5 in each group. We found that BAPN significantly reduced the Young’s modulus of aged cartilage (see Figure 7D-F shown below). These data support our hypothesis that reduction in matrix stiffness alters downstream mechanotransductive signaling and increases α-Klotho expression.

Figure 7C in the revised manuscript

Figure 7D in the revised manuscript

[Comment #29]

Overall, a very interesting study. The authors have provided a number of good association and functional studies. While they have proposed a model in Figure 5I to summarize their finding, I am not convinced that they have

clearly demonstrated the link between nuclear stiffness to DNMT1 expression, and as DNMT1 is a global methyltransferase, how this can lead to changes in the PI3K/Akt pathway that are related specifically to the functional level of alpha-Klotho. Perhaps the authors can discuss this more clearly with the limitations of their findings in mind. The difference in gender is also intriguing and if indeed the level of estrogen in female mice has a protective effect on this matrix stiffness and degenerative outcomes, perhaps the authors should consider testing this directly in relation to their model, so that the finding is not mouse specific and strengthens the relevance to human. The authors could also speculate whether their finding is specific to KOA or has more general conceptual implication to other degenerative diseases and cancer biology.

[Author action #29]

Thank you for the helpful suggestions. This study hypothesized that PI3K/Akt was, at least partially, regulated by altered α -Klotho expression in response to matrix stiffness via epigenetic regulation (i.e., DNMT1). However, we fully agree that DNMT1 affects many genes that may play a role in the maintenance of cartilage integrity. To address this question, we evaluated the level of 5-methylcytosine at the genome and found that, while global DNA methylation levels in young chondrocytes did significantly increase with increased matrix stiffness, these changes in global DNA methylation were manifold times smaller than changes in *Klotho* promoter methylation (**Figure 5B** and **Figure S12** shown below). These findings indicate that *Klotho* hypermethylation cannot be solely explained by increased global DNA methylation under stiff substrate conditions. Indeed, we demonstrated that the impact of mechanosignaling on regulation of DNMT1 and the consequent epigenetic modifications at *Klotho* promoter.

To further clarify the direct role of DNMT1 on *Klotho* gene expression, we performed ChIP-qPCR and found that increasing ECM stiffness showed an increased binding of DNMT1 and increased H3K4M2 at the *Klotho* promoter. On the other hand, Latrunculin A treatment abolished the change in binding of DNMT1 at the *Klotho* promoter (**Figure 6D** shown below). The change in DNMT1 binding at the *Klotho* promoter and DNMT1 expression by ECM stiffness is concordant with the change in *Klotho* promoter methylation and gene expression. Additional series of ChIP analyses also found that stiff substrates induced recruitment of Pol II, H3K4M2, but not H3K9M2, at the *Dnmt1* promoter (**Figure 6C** shown below). We observed a significant increase in binding of c-MYC, a transcription factor known to be stimulated and/or regulated by kinase signaling, at the *Dnmt1* promoter in response to increased ECM stiffness. Taken together, these findings indicate that *Dnmt1* is directly regulated by substrate stiffness, resulting in regulation of *Klotho* gene expression. We acknowledge that there may be additional histone modifications and/or transcription factors may contribute to the gene regulation of *Dnmt1* and *Klotho*. However, our data demonstrated that mechanotransductive signaling (ECM stiffness) epigenetically modulates *Klotho* transcription at least through DNMT1 binding.

To directly establish α -Klotho as playing a role in DNMT expression and chondrocyte health, we have also added new experiments in which DNMT activity in chondrocytes was inhibited using 5-Aza-2'-deoxycytidine, with and without siRNA *Klotho*. The results revealed that *Klotho* inhibition abrogated the beneficial effect of pharmacologically-inhibited DNMT (see **Figure 5G-H** shown below). Our data suggest that increased DNMT under stiff substrates disrupts chondrocyte health, and that this effect is at least partially mediated by epigenetic repression of *Klotho*. We have revised RESULTS accordingly (see **Page 12-13 / Line 281-291**).

We agree that the differences in sex are important and worthy of future study, and we considered adding studies to directly address this comment. However, we soon realized that to rigorously perform these studies, a post-menopausal model of KOA would be required to adequately model female humans. Such experiments would require extensive troubleshooting and analyses. As such, we felt these studies are outside the current scope of this paper. As an alternative, we have expanded our discussion of sex differences and the generalizability of our findings as follows:

Page 16 / Line 371-374 in DISCUSSION: “Investigating the link between estrogen and matrix-mediated

epigenetic regulation is of particular interest given that estrogen modulates mechanotransductive pathways, such as Hippo signaling⁴⁹, which we found to be associated with cartilage aging (Figure 1C).”

We also agree that a discussion of whether our finding is specific to KOA or has general conceptual implications to other age-related disease is warranted. We have revised the DISCUSSION as follows:

Page 18-19 / Line 419-425 in DISCUSSION: “While this study focused on cartilage as a model, a major conceptual innovation of our work is the demonstration of the novel mechanistic link between age-related increases in ECM stiffness and epigenetic repression of α -Klotho. It is widely recognized that α -Klotho plays a role in the attenuation of an aging phenotype in tissues throughout the body⁶⁸. However, the mechanisms provoking these declines over time remain unknown. As such, we anticipate that these studies may have broader implications in the field of aging research, even beyond cartilage.”

Figure 5B in the revised manuscript
(Young chondrocytes)

Figure S12A in the revised manuscript
(Aged chondrocytes)

Figure 6C-D in the revised manuscript

Figure 5G-H in the revised manuscript

Reviewer #5

[Comment #30]

Summary

The manuscript “Epigenetic regulation of α -Klotho by age-related matrix stiffening drives cartilage degeneration” by Iijima and colleagues presents interesting and important results linking cartilage ECM mechanical changes associated with aging and changes in the aging protein α -Klotho and methylation of its promoter, nuclear lamina, and DNMT1 expression. The manuscript’s findings address a timely topic in understanding how biophysical cues associated with aging/diseased tissue initiate and/or amplify pathogenic signaling to contribute to tissue

degeneration. In particular, the connection between ECM mechanosignaling and epigenetic remodeling is an emerging area of interest, with several notable recent reports elucidating novel pathways in this regard. Despite the enthusiasm for the topic and interesting results, further experiments are needed to fully establish the proposed mechanism and provide casual evidence for the pathway presented. In the current state, the causal links between different constituents of the molecular mechanism are not sufficiently supported by the data presented.

Major Comments

1) One premise of the proposed pathway is that α -Klotho inhibits PI3K/Akt signaling. Can the authors provide evidence that differences in α -Klotho expression are actually altering PI3K/Akt signaling? In Fig. 2I-K, there is some evidence from an α -Klotho knockdown/overexpression gene expression dataset that PI3KCA and AKT2 gene expression is modestly up and down regulated respectively. However, this does not necessarily result in a change in PI3K/Akt activity in response to α -Klotho levels, and the authors suggest a more direct inhibitory mechanism. Differences in Akt phosphorylation with respect to α -Klotho would provide much more compelling evidence.

[Author action #30]

Thank you for your suggestion. To address this, we sought to quantify PI3K protein expression and phosphorylation in chondrocytes cultured on pAAm gels (soft, medium, and stiff) using western blotting. Results revealed that chondrocytes cultured on soft substrates had lower levels of PI3K phosphorylation when compared to cells cultured on stiff substrates (see **Reference Figure 4**). These data suggest that a young-like (soft) matrix suppresses PI3K/Akt signaling. However, this suppressive effect on PI3K phosphorylation was blocked when chondrocytes seeded on soft substrates were treated with *Klotho* siRNA (see **Reference Figure 4**). Given that matrix stiffness regulates chondrogenicity *via* α -Klotho, these findings suggest that matrix stiffness regulates chondrocyte health via modulation of α -Klotho expression with subsequent changes in PI3K/Akt signaling. Despite the fact that these data support our hypothesis and are consistent with previous literature, we have concerns about the reliability of the western blot data because of strong background noise when considering the full blot. Although we tried several different antibodies for both PI3K and Akt, the western blot band signals were not consistent. As a result, we have not included these data and, instead, have downplayed the role of PI3K/Akt throughout the current manuscript, including removing PI3K from the graphical abstract in order to prevent any confusion.

Reference Figure 4

[Comment #31]

2) Much of the expression data relies on quantification of immunofluorescent images, which may not be the most reliable method to quantify protein abundance. Can the authors validate that these measurements reflect the true protein abundance in the cell through Western blots or other quantitative technique?

[Author action #31]

As per the reviewer’s suggestion, we quantified secreted type II collagen using ELISA in accordance with a comment raised by Reviewer #4 (see **Author action #27**). We also attempted to quantify α -Klotho concentration within conditioned media by ELISA. However, levels were below the level of detection of the assay, even after concentration using centrifugal filter units (Ultra-4, Amicon). We have included data (**Figure S11** for ELISA)

and revised the RESULTS as follows:

Page 10 / Line 231-237 in RESULTS: “When we assessed the secretion of type II collagen by chondrocytes under each of the experimental conditions (+/- ascorbic acid on soft, medium, or stiff substrates) using Enzyme-linked immuno-sorbent assay (ELISA), we found that chondrocytes cultured on stiff substrates secreted less type II collagen compared to those cultured on soft substrates, an effect that was independent from ascorbic acid supplementation (Figure S11C). We note that the number of chondrocytes was similar across the groups, confirming that the reduced secretion of type II collagen on stiff substrates was not a function of fewer chondrocytes (Figure S11D).”

[Comment #32]

3) It is not clear how the changes in nuclear morphology and Lamin A/C levels are related to the DNMT1 expression changes and α -Klotho promoter methylation levels measured. Are these two phenomena linked and if so, can the authors provide evidence that one effects the other? As is, this is presented in a correlative manner, and even in the schematic pathway (Fig. 5I) it is not shown how nuclear stiffness and Lamin A/C are mechanistically connected to the other constituents.

[Author action #32]

Thank you for these very helpful comments. We have now added new data showing that a stiff matrix increased chondrocyte nuclear eccentricity (i.e., making them less round), and that these changes were positively correlated with lamin A/C expression (see **Figure S9C-D** shown below).

We agree that clarifying the causal relationships of altered nuclear dynamics and DNMT1 are informative. However, manipulating nuclear dynamics in a physiological manner is technically difficult. As an alternative, we evaluated DNMT1 expression after inhibition of actin polymerization by Latrunculin A. This experiment was based on the understanding that altered mechanotransduction is tightly linked with changes in nuclear dynamics¹. Indeed, treatment of chondrocytes with Latrunculin A decreased nuclear eccentricity when compared to non-treated controls (i.e., increased roundness; **Figure S15C** shown below). We also performed ChIP-qPCR on chondrocytes cultured on a stiff substrate with and without Latrunculin A to determine whether inhibiting mechanotransduction and subsequent reorganization of the nuclear envelope (i.e., altered nuclear morphology and lamin A/C expression) directly modulates *Dnmt1* (binding of PolII and other transcription factors at *Dnmt1* promoter). Interestingly, the binding of Pol II, H3K4M2, and c-MYC binding at the *Dnmt1* promoter was abolished by Latrunculin A (**Figure 6C** shown below). These results indicate that increasing ECM stiffness can modulate gene transcription of *Dnmt1*, at least mediated by active chromatin marks and c-MYC.

Taken together, these results suggest that changes in mechanotransductive signaling result from matrix stiffening and a subsequent increase in nuclear deformation, ultimately leading to increased *Dnmt1* expression. Of note, siRNA for *Dnmt1* did not significantly alter the nuclear eccentricity of chondrocytes cultured on a stiff substrate (see **Figure S14A-C** shown below), indicating that increased nuclear deformation and lamin A/C under stiff matrix conditions are not a result of changes in *Dnmt1* expression.

Despite this, we recognize that our findings are correlative, and we cannot conclude a causal relationship between changes in lamin A/C and DNMT1. To prevent any confusion, we have modified the graphical abstract (see **Figure 8** shown below).

Figure S9C-D in the revised manuscript

Figure S15C in the revised manuscript

Figure 6C in the revised manuscript

Figure S14A-C in the revised manuscript

Figure 8 in the revised manuscript

Reference

1. Swift J, et al. Nuclear lamin-A scales with tissue stiffness and enhances matrix-directed differentiation. *Science* 341, 1240104 (2013).

[Comment #33]

4) Can the authors show that inhibition of DNMT1 results in more α -Klotho and less degeneration to support their molecular model? This would provide more direct evidence that DNMT1 is mediating α -Klotho expression and leading to phenotypic changes.

[Author action #33]

Thank you for this suggestion. As suggested, we inhibited *Dnmt1* in young chondrocytes cultured on a stiff

substrate using siRNA. Results revealed that inhibition of *Dnmt1* increased α -Klotho expression as well as markers of chondrogenicity (i.e., type II collagen and Aggrecan; **Figure 5E-F** shown below). Still, given our observation that a stiff matrix induces increased global methylation changes, we considered the possibility that the effect of DNMT1 on chondrocyte health is independent of α -Klotho levels. Therefore, to more definitively implicate α -Klotho as a mediator of the regulatory function of *Dnmt1* on chondrocyte health, we performed simultaneous inhibition of α -Klotho by siRNA *Klotho* and DNMT by 5-Aza-2'-deoxycytidine (5 Aza). Results revealed that intercepting the response of the *Klotho* gene to the pharmacological inhibition of DNMT abrogated the beneficial effect of the DNMT inhibition alone (**Figure 5G-H** shown below). These results suggest that the stiff substrate decreased chondrogenicity through increased *Dnmt1* with suppression of *Klotho*.

Figure 5E-H in the revised manuscript

[Comment #34]

5) *Is enhanced PI3K/Akt signaling the driver of increased DNMT1 expression and activity to reduce α -Klotho expression through promoter methylation? This is one potential mechanism, but it is not clear what is driving the increase in DNMT1 activity or why it is methylating the α -Klotho promoter specifically.*

[Author action #34]

We apologize for the confusion. We received a similar comment from **Reviewer #1 (Comment #5)**. Our mechanism suggests that PI3k/Akt signaling is downstream of DNMT expression and the subsequent α -Klotho changes. We propose that an age-related increase in ECM stiffness increases DNMT1 activity via mechanotransductive signals based on our findings demonstrating that: (1) an aged-like (stiff) substrate increased DNMT1 expression and (2) pharmacological inhibition of cytoskeletal actin polymerization by Latrunculin A reversed the effect of the aged-like (stiff) substrate on DNMT1 expression. In turn, DNMT1 inhibits *Klotho* expression, driving downstream cellular responses.

To further support our hypothesis, we performed ChIP-qPCR on chondrocytes cultured on a stiff substrate with and without Latrunculin A to determine whether mechanotransductive signaling directly modulates *Dnmt1* and *Klotho* transcription (binding of PolII and other transcription factors at *Dnmt1* and *Klotho* promoter). Interestingly, the increased Pol II and c-MYC bindings at the *Dnmt1* promoter induced by stiff substrates (**Figure 5D** shown below) were abolished by lat A treatment (**Figure 6C** shown below). Lat A treatment also abolished the change in binding of DNMT1 at the *Klotho* promoter induced by a stiff substrate (**Figure 5C** and **Figure 6D** shown below). In summary, these data suggest that age-related matrix stiffness drives DNMT1 recruitment at the *Klotho* promoter via cytoskeleton-mediated mechanotransductive pathway.

Figure 5C-D in the revised manuscript

Figure 6C-D in the revised manuscript

[Comment #35]

6) Similarly, is PI3K/Akt more activated on or in stiff matrices compared to soft (as reported in many other papers with a variety of cell types)? If so, is this due to a reduction in α -Klotho and subsequent loss of inhibition, or due to more direct integrin-mediated signaling?

[Author action #35]

To address this comment, we quantified phosphorylated PI3K and total PI3K for chondrocytes cultured on pAAm gel (soft, medium, and stiff) using western blotting. The results revealed that a soft substrate decreased pPI3K protein expression (see Reference Figure 5 shown below), indicating that young-like (soft) matrix suppresses PI3K/Akt signaling. However, the inhibitory effect of a soft matrix on PI3K/Akt was blocked when chondrocytes were seeded on a soft substrate with siRNA *Klotho* (see Reference Figure 5 shown below). This data indicates that activation of PI3K/Akt signaling is attributed to the loss of inhibition by α -Klotho.

Despite the fact that these data support our hypothesis and are consistent with previous literature, we have concerns about the reliability of the western blot data because of strong background noise when considering the full blot. Although we tried several different antibodies for both PI3K and Akt, the western blot band signals were not consistent. As a result, we have not included these data and, instead, have downplayed the role of PI3K/Akt throughout the current manuscript, including removing PI3K from the graphical abstract to prevent any confusion.

Reference Figure 5

[Comment #36]

7) Does the BAPN treatment actually reduce the stiffness of cartilage? Similarly, does it reduce DNMT1 expression/activity, or PI3K/Akt signaling, or other components of the proposed pathway?

[Author action #36]

We received a similar comment from Reviewer #4 (Comment #28). In response, we measured the stiffness of

the medial tibia using AFM and compared the Young's modulus across the two groups: aged+saline and aged+BAPN; n = 5 per group. Results revealed that BAPN significantly reduced the Young's modulus of aged cartilage (see **Figure 7D** shown below).

To further evaluate the downstream effect of the reduced cartilage stiffness by BAPN, we also added new data demonstrating that BAPN treatment did not significantly improve cartilage integrity when we use male middle-aged *Klotho* HET mice (**Figure 7K** shown below). These findings indicate that the improved cartilage integrity after BAPN treatment is driven, at least in part, by increased α -Klotho.

Figure 7D in the revised manuscript

Figure 7K in the revised manuscript

[Comment #37]

Minor Comments

1) Can the authors show images of the Lamin A/C staining in Fig. 3 that were used to quantify expression? These images would demonstrate differences in nuclear morphology and the nuclear lamina that would be of interest to the nuclear mechanotransduction audience.

[Author action #37]

We apologize the lack of clarity. Lamin A/C values in **Figure 3B** were derived from mass spec data. We have clarified this point in the figure caption.

We agree that showing the relationship between nuclear morphology and nuclear lamina is important. To address this comment, we first tried to quantify lamin A/C expression in cartilage tissue. Unfortunately, the fluorescence signal for lamin A/C was weak in cartilage tissue. In addition, we performed immunofluorescence for lamin A/C in triplicate, but we found that the signal intensity was not consistent across the different experimental cohorts. As an alternative, we integrated proteomic and histological data, which revealed that age-related alterations in nuclear eccentricity (assessed by histology) were significantly associated with lamin A/C and lamin B2 protein abundance (assessed by proteomics) *in vivo* (see **Figure S7A** shown below). We also assessed the same relationship in chondrocytes cultured on pAAm (soft, medium, and stiff) *in vitro* and found that lamin A/C fluorescence intensity in chondrocytes cultured on a stiff substrate was significantly positively associated with changes in nuclear deformation and nuclear eccentricity (see **Figure S9A-D** shown below). These findings suggest that increased lamin A/C expression in chondrocytes within an aged-like (stiff) microenvironment is accompanied by increased nuclear deformation both *in vivo* and *in vitro*.

Figure S7A in the revised manuscript

Figure S9A-D in the revised manuscript

[Comment #38]

2) It is not clear what the shape metrics presented in Fig. 3F represent, how they are measured, or why they were selected. The authors should provide more context surrounding this experimental design and interpretation.

[Author action #38]

We quantified chondrocyte nuclear morphology in cartilage tissue across the three groups (young, middle-age, and aged mice) according to 53 morphological variables detected by Cell Profiler software¹. To clarify, we have presented the calculation method for each morphological variable (see **Table S1** shown below). Principal component analysis revealed a clear segregation in nuclear morphology between young and aged chondrocytes in the native mouse cartilage (**Figure 3E**). To identify the specific factors segregating the nuclear morphology of the three groups, we extracted the top 10 variables contributing to the first principal component (**Figure 3F**). We have also revised the text as follows:

Page 8 / Line 185-189 in RESULTS: “CellProfiler measures the size, shape, intensity, and texture of a variety of nuclear/cell types in a high throughput manner. Our study focused on 53 morphological variables representing the shape and geometry of nuclei. Principal component analysis (PCA) for 53 morphological variable revealed clear segregation of nuclei from young and aged chondrocytes in mice (**Figure 3E**).”

Page 9 / Line 191-192 in RESULTS: “Detailed description for the top 10 variables is shown in Table S1.”

Table S1 in the revised manuscript

Variable	Description	Illustration
Eccentricity	Ratio of the distance between the foci of the ellipse and its major axis length. Ratio approaching 1 indicates a more elliptical shape.	Solidity	Measure of how many holes or concave boundaries in each object. Measured as area/convex hull area. The convex hull area is fitted onto the object by drawing a straight line across straight or concave areas of the cell but expanding outward along object protrusions. Ratio of 1 indicates an object without any concavities or indentations, while a ratio less than 1 indicates an object with holes or an irregular boundary. Ratio of Area to convex hull area.	MinorAxisLength	Length of the minor axis of the ellipse that bounds the nucleus/cell.	MedianRadius	Median distance of any pixel in the object to the closest pixel outside of the object.	MeanRadius	Mean distance of any pixel in the object to the closest pixel outside of the object.	
MaximumRadius	The greatest distance between any pixel inside the object to the nearest pixel outside of the object.	
Zernike shape features	Set of polynomial coefficients used to describe cell shape with increasing detail. The smallest circle that encloses the object is used to calculate Zernike features. Cells that are more closely related to the Zernike polynomial in question are reflected in a higher value.	
References

1. Carpenter AE, et al. CellProfiler: image analysis software for identifying and quantifying cell phenotypes. *Genome Biol* 7, R100 (2006).
2. Kametsky L, et al. Improved structure, function and compatibility for CellProfiler: modular high-throughput image analysis software. *Bioinformatics* 27, 1179-1180 (2011).
3. Cutiongco MF, Jensen BS, Reynolds PM, Gadegaard N. Predicting gene expression using morphological cell responses to nanotopography. *Nature communications* 11, 1-13 (2020).

[Comment #39]

3) As mentioned in the summary above, several prominent articles demonstrating connections between mechanotransduction and epigenetic remodeling have been published very recently. Including a brief discussion of these findings in the context of these other recent works would be beneficial. Examples include:

- Killaars, Walker, Anseth. *PNAS*, 2020. <https://doi.org/10.1073/pnas.2006765117>
- Walker et al, *Nature Biomedical Engineering*, 2021. <https://doi.org/10.1038/s41551-021-00748-3>
- Stowers et al, *Nature Biomedical Engineering*, 2019. <https://doi.org/10.1038/s41551-019-0420-5>
- Jang et al. *Nature Biomedical Engineering*, 2020. <https://doi.org/10.1038/s41551-020-00657-x>
- Jones et al. *Journal of Cell Biology*, 2021. <https://doi.org/10.1083/jcb.202007152>

[Author action #39]

Thank you for these suggestions. We have added the suggested references as follows:

Page 13 / Line 292-294 in RESULTS: “*Since extrinsic mechanical signals from the microenvironment are transmitted to the nucleus via the actin cytoskeleton⁴⁴, we next tested whether disruption of the cytoskeleton inhibits the impact of a stiff matrix on downstream chondrocyte responses.*”

Page 17 / Line 393-395 in DISCUSSION: “*Additionally, biophysical cues influence gene expression through recruitment of epigenetic modifiers or by inducing reorganization of laminin-associated chromatin^{37, 40, 59-63}.*”

[Comment #40]

4) *It is not clear why the phrase “PI3K/Akt signaling” is italicized in every instance in the paper.*

[Author action #40]

We have removed the italics for PI3K/Akt signaling throughout.

[Comment #41]

5) *More details regarding how fluorescence intensity was quantified should be provided in the Methods section.*

[Author action #41]

We received a similar comment from Reviewer #2 (**Comment #15**). We have provided the number of cells or nuclei used for the quantitative analysis in each figure caption. We also provided details in the METHODS as follows:

Page 25 / Line 651-656 in METHODS: “*Fluorescence intensity was quantified using Image J. More specifically, integrated density for each channel was divided by number of cells in each image (i.e., fluorescence intensity per cell). In each immunofluorescence-based quantification, we used 5 or 10 randomly selected images for 20x and 63x images, respectively, per individual sample. Data for the random images were then averaged in each independent sample and used for statistical analysis (e.g., if we analyzed 5 independent samples in total, data were shown as n = 5 wells/group).*”

[Comment #42]

6) *The acronyms pOAR and OARSI should be defined in the manuscript.*

[Author action #42]

We apologize for the lack of clarity. We have defined these words in the figure caption as outlined below and where they appear in the manuscript.

Figure 1 Caption:

“...Osteoarthritis Research Society International (OARSI) score is provided (0-24 points; higher value indicates more severe cartilage degeneration)...”

“**C.** Pathway analysis for young vs. aged in male and female mice. pAcc: the Boolean value of total bootstrap permutations accumulation; pORA: the Boolean value of over-representation p-value.”

[Comment #43]

7) *In figure panels with gradients (for example, Fig. 2A,B), the text overlapping the gradient is somewhat difficult to read. Perhaps these can be modified to readability.*

[Author action #43]

We have revised figures according to the reviewer suggestion.

[Comment #44]

8) In Fig. 3A, how were these proteins quantified? It may be more informative to present the data in a dot plot, similar to other data in this manuscript. If that is what is presented in Fig. 3B, that was not immediately clear to me, especially given the significant difference in Lamin A/C but not Lamin B2, despite similar trends in the dot plot.

[Author action #44]

Figure 3A shows $-\log_{10}$ (p-value) of nuclear envelope proteins, which were extracted from our mass spec data. It should be noted that both lamin A/C and B2 in cartilage were significantly changed (i.e., $-\log_{10}$ p-value >1.30) during aging in mice. The heat map for Figure 3A was presented in order to emphasize that only lamin A/C and B2 were significantly changed among the nuclear envelope proteins. To clarify, we have revised the figure caption as follows:

Figure 3 caption: “A. Age-related change in nuclear envelope proteins in male mice quantified by mass spectrometry-based proteomics. The heat map represents \log_{10} (p-value) of each protein, with >1.3 considered significant.”

REVIEWER COMMENTS

Reviewer #1 (Remarks to the Author):

In this revised manuscript, the authors reported that age-related alterations in ECM biophysical properties initiate pathogenic mechanotransductive cascades that promote Klotho promoter methylation and impair cartilage integrity. The authors are responsive to previous comments by adding new data and revising the manuscript.

Overall, this is an improved report which characterizes the association between age-related changes in ECM stiffness and Klotho expression. However, some concerns remain to be addressed.

Major

1. There is still a lack of data that support the major conclusion that “age-related alterations in ECM biophysical properties initiate pathogenic mechanotransductive cascades that promote Klotho promoter methylation and impair cartilage integrity”. The current data at most provide an association between the age-related alterations in ECM biophysical property and Klotho promoter methylation.
2. A major weakness is that there is virtually no data supporting the hypothesis that mechanotransductive force directly drive the recruitment of DNMT1/transcription factor to the Klotho gene promoter region. How the ECM stiffness is translated to the biochemical signaling remains insufficiently addressed.
3. The mechanistic data that directly link the mechanotransductive signaling and DNMT activities are apparently missing. It is highly possible that stiffer substrates directly or indirectly regulate DNMT1 expression and activity which is independent of the mechanotransductive signaling. In this revision, the authors inhibited actin polymerization of chondrocytes using Latrunculin A which reduced Pol II and c-MYC at the Dnmt1 promoter, decreased DNMT1 expression, and reduced Klotho promoter methylation and Klotho expression. These changes may be directly promoted by Latrunculin A and/or reduced actin polymerization rather than the reduced mechanotransductive signaling.
4. Similarly, LOX inhibitors and reduced collagen crosslinking may regulate DNMT1 expression and activity and Klotho promoter methylation which is independent of the mecnanotransductive signaling.
5. To prove that Klotho deficiency indeed drives aging-related cartilage degeneration, it is important to investigate whether in vivo expression of Klotho in chondrocytes prevents aging-associated impairment in cartilage. It is also helpful to determine whether expression of Klotho in chondrocytes affects stiff ECM-induced chondrocyte senescence and other degenerative changes (in vitro).
6. Does the RNA-seq analysis show decreased Klotho mRNA levels in aged chondrocytes?
7. Klotho deficiency causes ECM remodeling in other organs. This should be discussed in the context of Klotho in cartilage degeneration.

Reviewer #2 (Remarks to the Author):

I commend the authors on a thorough response to reviewer comments. The revised version is much improved and is likely to make an important contribution to the field.

Reviewer #3 (Remarks to the Author):

The authors responded the reviewer's comments appropriately. Although the authors removed the data on the systemic administration of soluble Klotho protein in the revised manuscript, the reviewer believes that the authors' conclusions are still well supported. If the authors clarify the precise mechanism by which the systemic administration of soluble Klotho improves cartilage degeneration in the future, it may worth another publication. Minor points: Figure S4: The panel labels are not matched with the legends.

Reviewer #4 (Remarks to the Author):

The authors have substantially addressed all my comments and I am satisfied with the rebuttal and the revision in the manuscript. I have no further comments to make.

Reviewer #5 (Remarks to the Author):

The authors have submitted a revised manuscript that contains extensive additional data to address most of the major concerns that were raised in the initial review. I have only two minor comments that should be addressed below, but I support the publication of this article.

- 1) The change in cartilage stiffness from BAPN treatment seems relatively minor (around 10% reduction) compared to the effect on the cells. This change is also modest compared to the stiffness differences evaluated in in vitro studies in this manuscript. Can the authors comments on the magnitude of the mechanical change and the cellular response?
- 2) The figure caption for figure 8 should contain more detail to guide the reader through the schematic.

We are grateful to the reviewers for providing additional feedback, which we believe has made the work even stronger. The point-to-point reply to the remaining reviewers' comments is found below.

Reviewer #1

[Comment #1]

In this revised manuscript, the authors reported that age-related alterations in ECM biophysical properties initiate pathogenic mechanotransductive cascades that promote Klotho promoter methylation and impair cartilage integrity. The authors are responsive to previous comments by adding new data and revising the manuscript. Overall, this is an improved report which characterizes the association between age-related changes in ECM stiffness and Klotho expression. However, some concerns remain to be addressed.

Major

- 1. There is still a lack of data that support the major conclusion that “age-related alterations in ECM biophysical properties initiate pathogenic mechanotransductive cascades that promote Klotho promoter methylation and impair cartilage integrity”. The current data at most provide an association between the age-related alterations in ECM biophysical property and Klotho promoter methylation.*
- 2. A major weakness is that there is virtually no data supporting the hypothesis that mechanotransductive force directly drive the recruitment of DNMT1/transcription factor to the Klotho gene promoter region. How the ECM stiffness is translated to the biochemical signaling remains insufficiently addressed.*
- 3. The mechanistic data that directly link the mechanotransductive signaling and DNMT activities are apparently missing. It is highly possible that stiffer substrates directly or indirectly regulate DNMT1 expression and activity which is independent of the mechanotransductive signaling. In this revision, the authors inhibited actin polymerization of chondrocytes using Latrunculin A which reduced Pol II and c-MYC at the Dnmt1 promoter, decreased DNMT1 expression, and reduced Klotho promoter methylation and Klotho expression. These changes may be directly promoted by Latrunculin A and/or reduced actin polymerization rather than the reduced mechanotransductive signaling.*

[Author action #1]

As the reviewer noted, in the previous data presented, we demonstrated that a stiff substrate, mimicking the biophysical properties of aged cartilage, methylates the Klotho promoter and compromises chondrocyte health (see **Figure 4-5**, **Figure S12**). In a loss-of-function paradigm, we showed that administration of Latrunculin A, an inhibitor of actin polymerization, counteracted stiffness-induced increases in DNMT1 level, increased DNMT1 binding to the Klotho promoter, reduced α -Klotho protein level, and reduced Klotho hypermethylation (see **Figure 6**, **Figure S15**).

In addition, another series of loss-of-function paradigm demonstrated that inhibition of *Dnmt1* by siRNA abrogated the deleterious effect of a stiff substrate on α -Klotho and chondrogenicity (see **Figure 5**). However, when both *Dnmt1* and *Klotho* were inhibited, the beneficial effect of *Dnmt1* inhibition on chondrocyte health was eliminated (**Figure 5**). These results further support the hypothesis that a stiff ECM drives DNMT1 inhibition of *Klotho*, resulting in decreased chondrogenicity.

To further evaluate the effect of mechanically mediated changes in biochemical signaling within chondrocytes (points 1 & 2) and to exclude the possibility that Latrunculin A had a direct biological impact on DNMT1 or α -Klotho (point 3), we have now conducted additional experiments to inhibit mechanotransductive signaling via alternative means. Specifically, we have inhibited cytoskeleton- and adhesion-mediated mechanotransductive pathways in chondrocytes cultured on a stiff substrate using a different inhibitor of actin fiber formation, Y-27632, as well as an inhibitor of focal adhesion kinase, PF-562271. We choose these drugs because they are a well-validated and a consistently used means to inhibit cellular mechanotransduction.¹⁻³ Moreover, the proteomics data

shown in **Figure 1** revealed that the focal adhesion pathway significantly changed with aging. This approach is consistent with previous reports in this journal that have similarly used Latrunculin A, Y27632, and PF-562271 to implicate mechanotransductive signaling.⁴⁻⁶

For our studies, DNMT1 and α -Klotho protein levels were assessed using immunofluorescence in a blinded manner. Similar to the effect observed with Latrunculin A, we found that both Y27632 and PF-562271 abolished the effect of a stiff substrate on DNMT1 and α -Klotho levels in chondrocytes (see **Figure S16A-B** below). Taken together, these data further support our hypothesis that age-related matrix stiffness impairs chondrocyte health through mechanotransductive signaling. We have revised RESULTS as follows:

Page 13-14 / Line 310-312 in RESULTS: “Treatments with Y27632, an inhibitor of actin fiber formation, and PF-562271, an inhibitor of focal adhesion kinase, similarly abolished the observed increase in DNMT1 and decrease in α -Klotho when chondrocytes were seeded on a stiff substrate (**Figure S16A-B**).”

— Inhibiting mechanotransduction abolished stiff substrates-induced increased DNMT1 and decreased α -Klotho —

Figure S16 in the revised manuscript

Reference

1. Kim JH, et al. Matrix cross-linking-mediated mechanotransduction promotes posttraumatic osteoarthritis. *Proc Natl Acad Sci U S A* 112, 9424-9429 (2015).
2. Dupont S, et al. Role of YAP/TAZ in mechanotransduction. *Nature* 474, 179-183 (2011).
3. Sato T, et al. A FAK/HDAC5 signaling axis controls osteocyte mechanotransduction. *Nat Commun* 11, 3282 (2020)
4. Junqueira Alves C, et al. Plexin-B2 orchestrates collective stem cell dynamics via actomyosin contractility, cytoskeletal tension and adhesion. *Nat Commun* 12, 6019 (2021)
5. Chen K, et al. Disrupting biological sensors of force promotes tissue regeneration in large organisms. *Nat Commun* 12, 5256 (2021)
6. Atcha H, et al. Mechanically activated ion channel Piezo1 modulates macrophage polarization and stiffness sensing. *Nat Commun* 12, 3256 (2021)

[Comment #2]

4. Similarly, LOX inhibitors and reduced collagen crosslinking may regulate DNMT1 expression and activity and Klotho promoter methylation which is independent of the mechanotransductive signaling.

[Author action #2]

We agree that it is difficult to entirely dissect the mechanical effects of BAPN from the biological effects. In our studies, we used BAPN, a LOX inhibitor, as a physiological tool to counteract the observed age-related increase in LOX protein abundance that we found in our proteomics data set (see **Figure 7**), which has been previously correlated with tissue stiffness¹. We confirmed the decreased cartilage stiffness of BAPN-treated aged mice (see **Figure 7**).

To address the reviewer's concern, we performed a new experiment to directly assess the effect of BAPN on α -Klotho expression in chondrocytes. Specifically, aged chondrocytes cultured on a stiff substrate were treated with BAPN (200 μ M) for 48 hours, after which time α -Klotho levels were quantified by an investigator blinded to the treatment group. We found that BAPN supplementation did not produce any direct effect on α -Klotho levels in aged chondrocytes (see **Figure S17** below), suggesting that the observed findings *in vivo* are likely due to the effects of BAPN on the matrix rather than on the chondrocytes themselves. These findings are consistent with previous studies showing that the effects of BAPN on chondrocyte function are primarily mediated by mechanotransductive mechanisms². We have revised the RESULTS as follows:

Page 15 / Line 341-346 in RESULTS: “It is noteworthy that BAPN supplementation did not influence α -Klotho levels in aged chondrocytes cultured on aged-like stiff substrates (**Figure S17**), indicating that the increased α -Klotho levels after BAPN administration *in vivo* is not likely to be explained by a direct effect of BAPN on α -Klotho. These findings are consistent with a previous study showing that the effects of BAPN on chondrocyte function are primarily mediated by mechanotransductive mechanisms⁴⁶.”

Figure S17 in the revised manuscript

We have revised the DISCUSSION as follows:

Page 18-19 / Line 416-422 in DISCUSSION: “...it is possible that LOX inhibition may have exerted direct biological effects, such as epigenetic modifications, on chondrocytes⁶⁵. To test this possibility, we showed that BAPN administration did not alter α -Klotho levels in aged chondrocytes *in vitro*, suggesting that the chondroprotective effects of BAPN administration *in vivo* is not likely to be attributed to direct effects of BAPN on α -Klotho. Rather, our data suggest that the beneficial effect of BAPN on α -Klotho expression *in vivo* is likely attributed to modification of matrix biophysical properties.”

Reference

1. Swift J, et al. Nuclear lamin-A scales with tissue stiffness and enhances matrix-directed differentiation. *Science* 341, 1240104 (2013).
2. Kim JH, et al. Matrix cross-linking-mediated mechanotransduction promotes posttraumatic osteoarthritis. *Proc Natl Acad Sci U S A* 112, 9424-9429 (2015).

[Comment #3]

5. To prove that *Klotho* deficiency indeed drives aging-related cartilage degeneration, it is important to investigate whether *in vivo* expression of *Klotho* in chondrocytes prevents aging-associated impairment in cartilage. It is also helpful to determine whether expression of *Klotho* in chondrocytes affects stiff ECM-induced chondrocyte senescence and other degenerative changes (*in vitro*).

[Author action #3]

We agree that a chondrocyte-specific overexpression of α -*Klotho* model is of interest, and we carefully considered the feasibility of these experiments. However, we soon realized that to rigorously perform these studies, the time required to make and validate the transgenic mice model would be extensive and the experiments risky. We previously worked with a company to knockout in muscle stem cells and, after more than one year, we were unsuccessful. We therefore view such studies for chondrocytes to be outside the scope of the current work.

In terms of the second comment, we revisited the archived microarray data of primary chondrocytes isolated from patients with knee osteoarthritis cultured on stiff plastic culture dish with and without siRNA *Klotho* (GSE 80285). Single gene set enrichment analysis revealed that *Klotho* deficiency did not significantly influence cellular senescence (see **Figure S4C** below). These data indicate that age-related α -*Klotho* deficiency may not be a primary driver of senescence in chondrocytes. We have added this new data (see **Figure S4C**).

Figure S4C in the revised manuscript

[Comment #4]

6. Does the RNA-seq analysis show decreased *Klotho* mRNA levels in aged chondrocytes?

[Author action #4]

Revisiting the RNA-seq and proteomics data, we were unable to detect *Klotho* transcripts and protein, respectively. This is not surprising given the relatively low mRNA/protein expression of *Klotho*, making it difficult to detect through -omics analysis. As an alternative, we compared *Klotho* gene expression between young and aged murine chondrocytes cultured on pAAm gel using the data presented in **Figure 5** and **Figure S12**. For these analyses we combined *Klotho* gene expression across the different stiffness groups (i.e., soft, medium, and stiff). The results revealed that aged murine chondrocytes displayed significantly lower *Klotho* gene expression than young counterparts (see **Reference Figure 1** below).

[Comment #5]

7. *Klotho* deficiency causes ECM remodeling in other organs. This should be discussed in the context of *Klotho* in cartilage degeneration.

[Author action #5]

Thank you for the suggestion. We have revised the text as follows:

Page 19 / Line 425-429 in DISCUSSION: “This possibility is supported by growing evidence that α -*Klotho* exerts anti-fibrotic effects in several tissues⁶⁷. Accordingly, α -*Klotho* overexpression or supplementation protects against fibrosis in renal and cardiac diseases⁶⁷. These anti-fibrotic effects of *Klotho* are mediated by the direct inhibitory effects on TGF- β 1, Wnt, and FGF2 signaling⁶⁷.”

Reviewer #3

[Comment #6]

The authors responded the reviewer’s comments appropriately.

Although the authors removed the data on the systemic administration of soluble *Klotho* protein in the revised manuscript, the reviewer believes that the authors’ conclusions are still well supported. If the authors clarify the precise mechanism by which the systemic administration of soluble *Klotho* improves cartilage degeneration in the future, it may worth another publication.

Minor points: Figure S4: The panel labels are not matched with the legends.

[Author action #6]

We apologize for this oversight. We have revised the legend in Figure S4 and checked for errors throughout.

Reviewer #5

[Comment #7]

The authors have submitted a revised manuscript that contains extensive additional data to address most of the major concerns that were raised in the initial review. I have only two minor comments that should be addressed below, but I support the publication of this article.

*1) The change in cartilage stiffness from BAPN treatment seems relatively minor (around 10% reduction) compared to the effect on the cells. This change is also modest compared to the stiffness differences evaluated in *in vitro* studies in this manuscript. Can the authors comments on the magnitude of the mechanical change and the cellular response?*

[Author action #7]

We apologize for the confusion. In Figure 7F, cartilage stiffness data is presented as “normalized Young’s modulus” and was calculated by: (1) transformation from MPa to Log MPa and (2) normalization based on cartilage stiffness in naïve young group. We performed the log transformation because the Young’s modulus data were not normally distributed. When we revisited the raw cartilage stiffness values, BAPN injection showed ~70% reduction compared to saline injection, which corresponds to our *in vitro* experimental design (4kPa, 21kPa, 100kPa). We have acknowledged these in the legend for Figure 7 as follows:

Page 42 / Figure 7 caption: “*Young’s modulus data was transformed into Log MPa and normalized by the Young’s modulus value in naïve young group.*”

We recognize that the reduced cartilage stiffness in BAPN group is still stiffer than young counterparts. Nevertheless, the OARSI score was significantly improved in the BAPN group when compared to age-matched controls, reaching values close to naïve young. This may speak to the sensitivity of chondrocytes to even small-scale changes in the stiffness of the microenvironment.

[Comment #8]

2) The figure caption for figure 8 should contain more detail to guide the reader through the schematic.

[Author action #8]

Thank you for the suggestion. We have elaborated the caption for Figure 8 as follows:

Page 43 / Figure 8 caption: “*Figure 8. Graphical abstract. Age-related matrix stiffening in articular cartilage initiates pathogenic mechanotransduction, driving chondrocyte dysfunction as well as disrupted cartilage integrity through Klotho promoter hypermethylation.*”

REVIEWERS' COMMENTS

Reviewer #1 (Remarks to the Author):

The authors addressed some questions raised by the previous reviewer. Overall, this is an improved report which characterizes the association between age-related changes in ECM stiffness and Klotho expression. However, some concerns remain to be addressed.

1. Because the mechanotransductive force was not measured in this study, the authors should modify their hypothesis or statement that "... mechanotransductive force directly drive the recruitment of DNMT1/transcription factor to the Klotho gene promoter region". The term "mechanotransductive signaling molecules" seems to be more appropriate.

2. The authors concluded that "...an aged-related loss of α -Klotho drives progressive cartilage degeneration.". Unfortunately, the authors did not provide data supporting this important conclusion in this revision. In [Author action #3], the authors indicated that "... the time required to make and validate... would be extensive and the experiments risky" [Author action #3]. It is not understood why the authors "view such studies for chondrocytes to be outside the scope of the current work" [Author action #3]. Actually, this is the major topic of the study (see title of the manuscript). If the transgenic mouse model is unavailable, in vivo cell-specific expression of Klotho may be considered.

3. The concern is that Klotho deficiency may not drive aging-associated cartilage degeneration because the authors concluded in the revision that age-related α -Klotho deficiency may not be a primary driver of senescence in chondrocytes (Fig. S4C) [Author action #3].

Reviewer #1

[Comment #1]

1. Because the mechanotransductive force was not measured in this study, the authors should modify their hypothesis or statement that "... mechanotransductive force directly drive the recruitment of DNMT1/transcription factor to the Klotho gene promoter region". The term "mechanotransductive signaling molecules" seems to be more appropriate.

[Author action #1]

We have revised the **DISCUSSION** as follows:

Page 18 / Line 414-420 in DISCUSSION: "...inhibition of actin polymerization rescued the stiff matrix-induced increases in binding of DNMT1 at the Klotho promoter, suggesting that an age-related increase in matrix stiffness increases DNMT1 recruitment at the Klotho promoter, which is attributed, at least in part, to cytoskeleton-mediated mechanotransduction. Nevertheless, whether mechanotransductive force directly drives the recruitment of DNMT1/transcription factors to the α -Klotho gene promoter region was not directly evaluated in this work and remains an interesting area for future investigation."

[Comment #2]

2. The authors concluded that "...an aged-related loss of α -Klotho drives progressive cartilage degeneration.". Unfortunately, the authors did not provide data supporting this important conclusion in this revision. In [Author action #3], the authors indicated that "... the time required to make and validate... would be extensive and the experiments risky" [Author action #3]. It is not understood why the authors "view such studies for chondrocytes to be outside the scope of the current work" [Author action #3]. Actually, this is the major topic of the study (see title of the manuscript). If the transgenic mouse model is unavailable, *in vivo* cell-specific expression of Klotho may be considered.

[Author action #2]

We apologize for the confusion. We did not mean to imply that the studies suggested by the reviewer were not relevant to the study but, rather, that the time required to perform the studies in a rigorous and robust way were infeasible. To address this comment, we have removed any causal claims of epigenetic regulation of α -Klotho driving cartilage degeneration in OA *in vivo* throughout the manuscript including title and abstract. For example, title was changed as follows:

Title: "Age-related matrix stiffening epigenetically regulates α -Klotho expression and compromises chondrocyte integrity"

We further acknowledged that the lack of data from chondrocyte-specific genetic manipulation on Klotho limited providing strong conclusion regarding the causal relationship among age-related increased matrix stiffness, Klotho, and cartilage degeneration *in vivo* as follows:

Page 17 / Line 385-389 in DISCUSSION: "In search of novel upstream candidates that regulate PI3K/Akt signaling, which we found to be significantly changed over time in male mice, a series of genetic and pharmacologic manipulations revealed declines in α -Klotho as a possible driver of cartilage degeneration. Future studies would benefit from chondrocyte-specific genetic manipulation of α -Klotho to further evaluate the specificity of this effect."

[Comment #3]

3. The concern is that Klotho deficiency may not drive aging-associated cartilage degeneration because the authors concluded in the revision that age-related α -Klotho deficiency may not be a primary driver of senescence

in chondrocytes (Fig. S4C) [Author action #3].

[Author action #3]

As pointed out, microarray data (GSE80285) from *Klotho* knockdown indicates that α -Klotho deficiency may not be a primary driver of cellular senescence. We posit that Klotho may act through other age-related mechanisms, such as elevated inflammation and/or impaired autophagy¹. Indeed, cellular senescence is just one of the nine hallmarks of aging². Studies have shown that α -Klotho inhibits insulin growth factor receptor-mediated PI3K/Akt signaling and subsequently enhances FoxO³⁻⁴. FoxO, which protects chondrocytes from oxidative stress and prevents spontaneous cartilage degeneration⁵⁻⁶, is markedly reduced with aging in both murine and human cartilage⁵⁵. We have included these points in the **DISCUSSION**.

Page 17 / Line 389-394 in DISCUSSION: “Our results are consistent with previous *in vitro* and *in vivo* studies showing that α -Klotho overexpression counteracts chondrocyte dysfunction and cartilage degeneration in a PTOA model^{25, 51}. Another study also demonstrated that overexpression of α -Klotho, combined with soluble TGF- β supplementation, enhanced chondrocyte health and cartilage integrity in a chemically-induced model of KOA51. α -Klotho inhibits insulin growth factor receptor-mediated PI3K/Akt signaling and subsequently enhances FoxO^{21, 53}. FoxO, which protects chondrocytes from oxidative stress and prevents spontaneous cartilage degeneration^{54, 54}, is markedly reduced with aging in both murine and human cartilage⁵⁶. Our results suggest that the interaction of α -Klotho and FoxO expression to mitigate oxidative stress is an area worthy of future studies.”

References

1. Iijima H, et al. Meta-analysis Integrated With Multi-omics Data Analysis to Elucidate Pathogenic Mechanisms of Age-Related Knee Osteoarthritis in Mice. *J Gerontol A Biol Sci Med Sci* 77, 1321-1334 (2022).
2. López-Otín, et al. The hallmarks of aging. *Cell* 153, 1194-217 (2013).
3. Martínez-Redondo P, et al. α KLOTHO and sTGF β R2 treatment counteract the osteoarthritic phenotype developed in a rat model. *Protein Cell* 11, 219-226 (2020).
4. Yamamoto M, et al. Regulation of oxidative stress by the anti-aging hormone klotho. *J Biol Chem* 280, 38029-38034 (2005).
5. Akasaki Y, et al. FoxO transcription factors support oxidative stress resistance in human chondrocytes. *Arthritis Rheumatol* 66, 3349-3358 (2014).
6. Matsuzaki T, et al. FoxO transcription factors modulate autophagy and proteoglycan 4 in cartilage homeostasis and osteoarthritis. *Science translational medicine* 10, (2018).
7. Akasaki Y, et al. Dysregulated FOXO transcription factors in articular cartilage in aging and osteoarthritis. *Osteoarthritis Cartilage* 22, 162-170 (2014).